# DRAGO: Primal-Dual Coupled Variance Reduction for Faster Distributionally Robust Optimization

Ronak Mehta[1]     Jelena Diakonikolas[2]     Zaid Harchaoui[1]

[1]University of Washington, Seattle     [2]University of Wisconsin, Madison

## Abstract

We consider the penalized distributionally robust optimization (DRO) problem with a closed, convex uncertainty set, a setting that encompasses learning using $f$-DRO and spectral/$L$-risk minimization. We present DRAGO, a stochastic primal-dual algorithm that combines cyclic and randomized components with a carefully regularized primal update to achieve dual variance reduction. Owing to its design, DRAGO enjoys a state-of-the-art linear convergence rate on strongly convex-strongly concave DRO problems with a fine-grained dependency on primal and dual condition numbers. The theoretical results are supported by numerical benchmarks on regression and classification tasks.

## 1 Introduction

Contemporary machine learning research is increasingly exploring the phenomenon of distribution shift, in which predictive models encounter different data-generating distributions in training versus deployment [Wiles et al., 2022]. A popular approach to learn under potential distribution shift is *distributionally robust optimization* (DRO) of an empirical risk-type objective

$$\min_{w \in \mathcal{W}} \max_{q \in \mathcal{Q}} \left[ \mathcal{L}_0(w, q) := \sum_{i=1}^n q_i \ell_i(w) \right], \tag{1}$$

where $\ell_i : \mathbb{R}^d \to \mathbb{R}$ denotes the loss on training instance $i \in [n] := \{1, \ldots, n\}$, and $q = (q_1, \ldots, q_n) \in \mathcal{Q}$ is a vector of $n$ weights for each example. The feasible set $\mathcal{Q}$, often called the *uncertainty set*, is a collection of possible instance-level reweightings arising from distributional shifts between train and evaluation data, and is often chosen as a ball about the uniform vector $\mathbf{1}/n = (1/n, \ldots, 1/n)$ in $f$-divergence [Namkoong and Duchi, 2016, Carmon and Hausler, 2022, Levy et al., 2020] or a spectral/$L$-risk-based uncertainty set [Mehta et al., 2023].

We consider here the penalized version of (1), stated as

$$\mathcal{L}(w, q) := \sum_{i=1}^n q_i \ell_i(w) - \nu D(q \| \mathbf{1}/n) + \frac{\mu}{2} \|w\|_2^2, \tag{2}$$

where $\mu, \nu \geq 0$ are regularization parameters and $D(q \| \mathbf{1}/n)$ denotes some statistical divergence (such as the Kullback-Leibler (KL) or $\chi^2$-divergence) between the original weights $\mathbf{1}/n$ and shifted weights $q$. For clarity, we focus on the cases of $\mu, \nu > 0$, but also describe the modifications to the methods, results, and proofs for cases in which $\mu = 0$ or $\nu = 0$, in Appx. C.4. See Fig. 1 for intuition on the relationship between the uncertainty set, divergence $D$, and hyperparameter $\nu$.

Standard (1) and penalized (2) DRO objectives have seen an outpour of recent use in reinforcement learning and control [Lotidis et al., 2023, Yang et al., 2023, Wang et al., 2023a, Yu et al., 2023, Kallus et al., 2022, Liu et al., 2022] as well as creative applications in robotics [Sharma et al., 2020], language modeling [Liu et al., 2021], sparse neural network training [Sapkota et al., 2023]

CVaR Uncertainty Set with $\chi^2$ Penalty    $\chi^2$ Uncertainty Set with $\chi^2$ Penalty    CVaR Uncertainty Set with KL Penalty

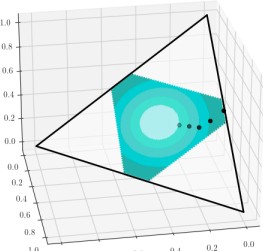 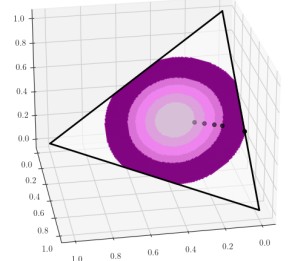 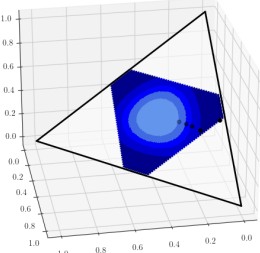

Figure 1: **Visualization of Uncertainty Sets and Penalties.** Each plot is a probability simplex in $n = 3$ dimensions with the uncertainty set as the colored portion. The black dots are optimal dual variables $q_\nu^\star := \arg\max_{q \in \mathcal{Q}} \sum_{i=1}^n q_i \ell_i(w) - \nu D(q \| \mathbf{1}/n)$ for a fixed $w \in \mathcal{W}$. As $\nu$ decreases, $q_\nu^\star$ may shift toward the boundary of the uncertainty set. The combination of $\nu$ and $D$ determines an "effective" uncertainty set, whose shape is given by the level sets of $D$. Our methods apply to both.

and defense against model extraction [Wang et al., 2023b]. However, even in a classical supervised learning setup, current optimization algorithms for DRO have limitations in both theory and practice.

For context, we consider the large-scale setting in which the sample size $n$ is high, and the training loss in each example is accessed through a collection of $n$ *primal first-order oracles* $\{(\ell_i, \nabla \ell_i)\}_{i=1}^n$. Quantitatively, we measure the performance of algorithms by runtime or global complexity of elementary operations to reach within $\varepsilon$ of the minimum of $\bar{\mathcal{L}}(w) = \max_{q \in \mathcal{Q}} \mathcal{L}(w, q)$, while qualitatively, we consider the types of uncertainty sets that can be handled by the algorithm and convergence analysis. Under standard assumptions, $\bar{\mathcal{L}}$ is differentiable with gradient computed via

$$q^\star(w) = \arg\max_{q \in \mathcal{Q}} \mathcal{L}(w, q), \text{ followed by } \nabla \bar{\mathcal{L}}(w) = \sum_{i=1}^n q_i^\star(w) \nabla \ell_i(w) + \mu w. \tag{3}$$

In the learning setting, we are interested in stochastic algorithms that can approximate this gradient with $b < n$ calls to the oracles. For uniformly randomly sampled $i \in [n]$, $n q_i^\star(w) \nabla \ell_i(w) + \mu w$ is an unbiased estimator of $\nabla \bar{\mathcal{L}}(w)$. However, computing $q_i^\star(w)$ depends on the first step in (3) which itself requires calling all $n$ oracles (see (2)), i.e., it is no different than the cost of full batch gradient descent. A direct minibatch stochastic gradient descent approach would approximate $\mathcal{L}(w, \cdot)$ in the first step of (3) with only $b$ calls to generate approximate weights $\hat{q}(w)$. Because $\hat{q}(w) \neq q^\star(w)$ in general for $b < n$, these methods have non-vanishing bias, i.e., do *not* converge [Levy et al., 2020].

This motivated research on DRO-specific stochastic algorithms with theoretical convergence guarantees under particular assumptions (see Tab. 1 and Appx. B for details) [Namkoong and Duchi, 2016, Levy et al., 2020, Carmon and Hausler, 2022]. Although we highlight the dependence on sample size $n$ and suboptimality $\varepsilon$, the dependence on all constants is given in Tab. 1. For $f$-divergence-based uncertainty sets in the standard oracle framework, several methods achieve a $O(\varepsilon^{-2})$ complexity. Levy et al. [2020] do so by proving uniform bias bounds, so that if $b$ scales as $O(\varepsilon^{-2})$, the convergence guarantee is achieved. However, if the required batch size $b$ exceeds the training set size $n$, then the method reduces to the sub-gradient method, as we can see in Tab. 1. These sublinear rates typically stem from two causes. The first is the adoption of a "fully stochastic" perspective on the oracles, wherein each oracle's output is treated as an independent random sample drawn from a probability distribution. The second is the non-smoothness of the objective, as we shall see below.

Variance reduction techniques, on the other hand, exploit the fact that the optimization algorithm takes multiple passes through the same dataset, and achieve *linear* rates of the form $O((n + \kappa_\ell) \log(\varepsilon^{-1}))$ in empirical risk minimization when the objective is both smooth (i.e., has Lipschitz continuous gradient) and strongly convex and $\kappa_\ell$ is an associated condition number [Johnson and Zhang, 2013, Defazio et al., 2014]. Assuming access to stronger oracles involving constrained minimization and applying a variance reduction scheme, Carmon and Hausler [2022] achieve $O(n\varepsilon^{-2/3} + n^{3/4}\varepsilon^{-1})$ for $f$-divergences as well, but do not obtain linear convergence due to the second type of cause: the objective $\nabla \bar{\mathcal{L}}$ is *non-smooth* when $\nu = 0$.

Recently, Mehta et al. [2024] handled the $\nu > 0$ case for spectral risk uncertainty sets, and their variance-reduced algorithm achieves a linear $O((n + \kappa_{\mathcal{Q}}\kappa_\ell)\ln(1/\varepsilon))$ convergence guarantee (where $\kappa_{\mathcal{Q}} \geq 1$ measures the "size" of the uncertainty set), but only with a lower bound of order $\Omega(n)$ on the problem parameter $\nu$. The challenge of this problem, considered from a general optimization viewpoint beyond DRO, stems from the non-bilinearity of the coupled term $\sum_{i=1}^n q_i \ell_i(w)$ and the constraint that $\sum_{i=1}^n q_i = 1$ for probability vectors. If the coupled term was bilinear (i.e., of the form $q^\top A w$ for $A \in \mathbb{R}^{n \times d}$) and the constraints applied separately to each $q_i$, then dual decomposition techniques could be used. Qualitatively, algorithms and analyses often rely on particular uncertainty sets; for example, Kumar et al. [2024] use duality arguments specific to the Kullback-Leibler uncertainty set to create a primal-only minimization problem. See Appx. B for a detailed discussion of related work from the ML and the optimization lenses. Given the interest from both communities, we address whether a stochastic DRO algorithm can simultaneously 1) achieve a linear convergence rate for any $\nu > 0$ and 2) apply to many common uncertainty sets.

**Contributions** We propose DRAGO, a minibatch primal-dual algorithm for the penalized DRO problem (2) that achieves $\varepsilon$-suboptimality in

$$O\left(\left[\frac{n}{b} + \frac{\kappa_{\mathcal{Q}} L}{\mu} + \frac{n}{b}\sqrt{\frac{nG^2}{\mu\nu}}\right]\ln\left(\frac{1}{\varepsilon}\right)\right) \tag{4}$$

iterations, where $b \in \{1, \ldots, n\}$ is the minibatch size, $\kappa_{\mathcal{Q}} = nq_{\max} := n\max_{q \in \mathcal{Q}, i \in [n]} q_i$ measures the size of the uncertainty set, and $G$ and $L$ are the Lipschitz continuity parameters of $\ell_i$ and $\nabla\ell_i$, respectively. For commonly used parameters of uncertainty sets, $nq_{\max}$ is bounded above by an absolute constant independent in $n$ (see Prop. 3), so for $d < n$ and $b = n/d$, we maintain an $O(n)$ per-iteration complexity (the dual dimensionality) while reducing the number of iterations to $O((d + \kappa_{\mathcal{Q}} L/\mu + d\sqrt{nG^2/(\mu\nu)}))\ln(1/\varepsilon))$. Theoretically, the complexity bound we achieve in (4) is the best one among current penalized DRO algorithms, delineating a clear dependence on smoothness constants of the coupled term and strong convexity constants of the individual terms in (2). Practically, DRAGO has a single hyperparameter and operates on any closed, convex uncertainty set for which the map $l \mapsto \arg\max_{q \in \mathcal{Q}}\{q^\top l - \nu D(q\|\mathbf{1}/n)\}$ is efficiently computable. DRAGO is also of general conceptual interest as a stochastic variance-reduced primal-dual algorithm for min-max problems. It delicately combines randomized and cyclic updates, which effectively address the varying dimensions of the two problems (see Sec. 2). The theoretical guarantees of the algorithm are explained in Sec. 3. Numerical performance benchmarks are shown in Sec. 4.

## 2 The DRAGO Algorithm

We present here the **D**istributionally **R**obust **A**nnular **G**radient **O**ptimizer (DRAGO). While similar in spirit to a primal-dual proximal gradient method with a stochastic flavor, there are several innovations that allow the algorithm to achieve its superior complexity guarantee. These include using 1) minibatch stochastic gradient estimates to improve the trade-off between the per-iteration complexity and required number of iterations (especially when $n \gg d$), 2) a combination of randomized and cyclically updated components in the primal and dual gradient estimates, and 3) a novel regularization term in the primal update which reduces variance in the gradient estimate (i.e., *coupled variance reduction*). Here, we describe the algorithm in a manner that helps elucidate the upcoming theoretical analysis (Sec. 3). On the other hand, in Appx. D, we present an alternate description of DRAGO that is amenable to direct implementation in code.

**Notation & Terminology** Let $\psi : \mathbb{R}^n \to \mathbb{R} \cup \{+\infty\}$ be a proper, convex function such that $\mathcal{Q} \subseteq \mathrm{dom}(\psi) := \{q \in \mathbb{R}^n : \psi(q) < +\infty\}$. Let $\psi$ have a non-empty subdifferential for each $q \in \mathcal{Q}$, and denote by $\nabla\psi$ a map from $q \in \mathcal{Q}$ to an arbitrary but consistently chosen subgradient in $\partial\psi(q)$. We denote the *Bregman divergence* generated by $\psi$ as $\Delta_\psi(q, \bar{q}) = \psi(q) - \psi(\bar{q}) - \langle\nabla\psi(\bar{q}), q - \bar{q}\rangle$. We employ the Bregman divergence in this way for purely technical reasons, and for common cases this version will not be invoked. Finding a minimizer of (2) is equivalent to finding a saddle-point $(w_\star, q_\star) \in \mathcal{W} \times \mathcal{Q}$ which satisfies

$$\max_{q \in \mathcal{Q}} \mathcal{L}(w_\star, q) = \mathcal{L}(w_\star, q_\star) = \min_{w \in \mathcal{W}} \mathcal{L}(w, q_\star).$$

| Method | Assumptions | Uncertainty Set | Runtime / Global Complexity (Big-$\tilde{O}$) |
|---|---|---|---|
| Sub-Gradient Method | $\ell_i$ is $G$-Lipschitz
$\|w-w'\|_2 \leq R$ (if included)
$\mu > 0$ (if included)
$\nabla \ell_i$ is $L$-Lipschitz and $\nu > 0$ (if included) | Support Constrained
Support Constrained
Support Constrained | $nd \cdot (GR)^2 \varepsilon^{-2}$
$nd \cdot G^2 \mu^{-1} \varepsilon^{-1}$
$nd \cdot \mu^{-1}\left(L+\mu+nG^2/\nu\right)\log(1/\varepsilon)$ |
| $\mathcal{L}_{\mathrm{CVaR}}$-SGD[†]
$\mathcal{L}_{\chi^2}$-SGD[†]
$\mathcal{L}_{\chi^2-\mathrm{pen}}$-SGD[†]
[Levy et al., 2020] | $\ell_i$ is $G$-Lipschitz and in $[0, B]$
$\|w-w'\|_2 \leq R$ for all $w, w' \in \mathcal{W}$
$\nu > 0$ (if included) | $\theta$-CVaR
$\rho$-ball in $\chi^2$-divergence
$\chi^2$-divergence penalty | $\min\left\{n, B^2\theta^{-1}\varepsilon^{-2}\right\}d \cdot (GR)^2 \varepsilon^{-2}$
$\min\left\{n, (1+\rho)B^2\varepsilon^{-2}\right\}d \cdot (GR)^2 \varepsilon^{-2}$
$\min\left\{n, B^2\nu^{-1}\varepsilon^{-1}\right\}d \cdot (GR)^2 \varepsilon^{-2}$ |
| BROO*
BROO*
[Carmon and Hausler, 2022] | $\ell_i$ is $G$-Lipschitz
$\|w-w'\|_2 \leq R$ for all $w, w' \in \mathcal{W}$
$\nabla \ell_i$ is $L$-Lipschitz (if included) | 1-ball in $f$-divergence
1-ball in $f$-divergence | $nd \cdot (GR)^{2/3}\varepsilon^{-2/3}+d(GR)^2\varepsilon^{-2}$
$nd \cdot (GR)^{2/3}\varepsilon^{-2/3}+n^{3/4}d\left(GR\varepsilon^{-1}+L^{1/2}R\varepsilon^{-1/2}\right)$ |
| LSVRG
LSVRG
[Mehta et al., 2023] | $\ell_i$ is $G$-Lipschitz
$\nabla \ell_i$ is $L$-Lipschitz
$\mu > 0, \nu > 0, \kappa := (L+\mu)/\mu$ | Spectral Risk Measures ($\nu$ small)
Spectral Risk Measures ($\nu \geq \Omega(nG^2/\mu)$) | None
$(n + \kappa_{\mathcal{Q}}\kappa)\,d\log(1/\varepsilon)$ |
| Prospect
Prospect
[Mehta et al., 2024] | $\ell_i$ is $G$-Lipschitz
$\nabla \ell_i$ is $L$-Lipschitz
$\mu > 0, \nu > 0, \kappa := (L+\mu)/\mu, \delta = G^2/(\mu\nu)$ | Spectral Risk Measures ($\nu$ small)
Spectral Risk Measures ($\nu \geq \Omega(nG^2/\mu)$) | $n(n+d)\max\left\{n\delta + \kappa n q_{\max}, n^3\delta^2\kappa^2, n^3\delta^3\right\}\log(1/\varepsilon)$
$(n + \kappa_{\mathcal{Q}}\kappa)(n+d)\log(1/\varepsilon)$ |
| DRAGO (Ours) | $\ell_i$ is $G$-Lipschitz
$\nabla \ell_i$ is $L$-Lipschitz
$\mu > 0, \nu > 0, b := $ Batch Size | Support Constrained | $n\left(d + \kappa_{\mathcal{Q}}L/\mu + d\sqrt{nG^2/(\mu\nu)}\right)\log(1/\varepsilon)$ |

Table 1: **Complexity Bounds of DRO Methods.** Runtime or global complexity (i.e., the total number of elementary operations required to compute $w$ satisfying $\max_{q \in \mathcal{Q}} \mathcal{L}(w, q) - \mathcal{L}(w_\star, q_\star) \leq \varepsilon$. Throughout, we assume that each $\ell_i$ is convex and $\mu, \nu \geq 0$. The "Support Constrained" uncertainty set refers to all closed, convex sets of probability mass vectors and any 1-strongly convex penalty. This includes $f$-divergences and spectral risk measures, but not general Wasserstein balls. *Bounds hold in high probability. †Complexity is measured in our framework; see Appx. B for details.

We denote the Jacobian of $\ell := (\ell_1, \ldots, \ell_n)$ at $w$ as $\nabla \ell(w) \in \mathbb{R}^{n \times d}$. We refer to the gradient of the coupled term $q^\top \ell(w)$ of (2) with respect to $w$ and $q$ (that is, $\nabla \ell(w)^\top q$ and $\ell(w)$) as the primal and dual gradients, respectively. In both the definition of the algorithm as well as the analysis, we will consider a sequence of positive constants $(a_t)_{t \geq 1}$ with the additional values $a_0 = 0$. The partial sums of this sequence will be denoted $A_t = \sum_{\tau=0}^t a_\tau$. Here, $t$ represents the iteration counter while the convergence rate will be proportional to $A_t^{-1}$ (see Sec. 3), so we wish for the sequence to grow as fast as possible. As mentioned in Sec. 1, $d$ is the primal dimension, $n$ is the dual dimension as well as the sample size, and $b$ denotes a batch size that divides $n$ for ease of presentation.

**Algorithm Description** We specify the algorithm by recursively defining a sequence of primal-dual iterates $\{(w_t, q_t)_{t \geq 1}\}$ that achieve a particular convergence guarantee. First, we may assume that $\mathbf{0} \in \mathcal{W}$ without loss of generality, so fix $w_0 = \mathbf{0}$ and $q_0 = \mathbf{1}/n$. For any $t \geq 1$, we introduce $v_{t-1}^{\mathrm{P}}$ and $v_t^{\mathrm{D}}$ as to-be-specified stochastic gradient estimates of the quantities $\nabla \ell(w_{t-1})^\top q_{t-1}$ and $\ell(w_t)$, respectively. We choose to update $w_t$ before $q_t$, so that $v_{t-1}^{\mathrm{P}}$ may depend only on the history $\{(w_\tau, q_\tau)\}_{\tau=0}^{t-1}$, whereas $v_t^{\mathrm{D}}$ may depend additionally on $w_t$ as well. Letting $(C_t)_{t \geq 1}$ and $(c_t)_{t \geq 1}$ be to-be-specified sequences of positive constants with $C_0 = c_0 = 0$, we employ primal-dual proximal approach guided by the updates

$$w_t := \arg\min_{w \in \mathcal{W}} \left\{ a_t \langle v_t^{\mathrm{P}}, w \rangle + \frac{a_t \mu}{2} \|w\|_2^2 + \frac{C_{t-1}\mu}{2} \|w - w_{t-1}\|_2^2 \right. \tag{5}$$

$$\left. + \frac{c_{t-1}\mu}{2} \sum_{s=t-n/b}^{t-2} \|w - w_{s \vee 0}\|_2^2 \right\} \tag{6}$$

$$q_t := \arg\max_{q \in \mathcal{Q}} \left\{ a_t \langle v_t^{\mathrm{D}}, q \rangle - a_t \nu D(q \| q_0) - A_{t-1} \nu \Delta_D(q, q_{t-1}) \right\}. \tag{7}$$

For the primal update, despite the non-standard term (6), $w_t$ can be computed easily in closed form when $\mathcal{W} = \mathbb{R}^d$. Otherwise, retrieving $w_t$ relies on computing an $\ell_2$-norm projection onto $\mathcal{W}$. On the other hand, the maximization (7) can often be solved exactly or to high accuracy using methods specific to each uncertainty set; we describe these in detail in Appx. D.2. The randomized and cyclic coordinate-wise components mentioned above are contained within the upcoming formulas for $v_{t-1}^{\mathrm{P}}$ and $v_t^{\mathrm{D}}$. After specifying these vectors, the constants $(C_t, c_t)$ will be chosen in the analysis to recover the final algorithm.

Next, we describe the computation of $v_{t-1}^{\mathrm{P}}$ and $v_t^{\mathrm{D}}$, which will rely on quantities that are stored by the algorithm along its iterations. We store three tables of values $\hat{\ell}_t \in \mathbb{R}^n$, $\hat{q}_t \in \mathbb{R}^n$, and $\hat{g} \in \mathbb{R}^{n \times d}$, which are approximations of $\ell(w_t)$, $q_t$, and $\nabla \ell(w_t)$, respectively. Before explaining how these tables are updated, consider the data indices $\{1, \ldots, n\}$ to be partitioned into $n/b$ blocks, written $(B_1, \ldots, B_{n/b})$ for $B_K = (K - b + 1, \ldots, Kb)$. On each iteration $t$, we randomly sample independent block indices $I_t, J_t \sim \mathrm{Unif}[n/b]$ and define

$$v_{t-1}^{\mathrm{P}} = \hat{g}_{t-1}^\top \hat{q}_{t-1} + \frac{a_{t-1}}{a_t} \cdot \frac{n}{b} \sum_{i \in B_{I_t}} (q_{t-1,i} \nabla \ell_i(w_{t-1}) - \hat{q}_{t-2,i} \hat{g}_{t-2,i}) \tag{8}$$

$$v_t^{\mathrm{D}} = \hat{\ell}_t + \frac{a_{t-1}}{a_t} \cdot \frac{n}{b} \sum_{j \in B_{J_t}} (\ell_j(w_t) - \hat{\ell}_{t-1,j}) e_j \tag{9}$$

As for the tables of approximations, we update them on each iteration without suffering the $O(nd)$ computational cost of querying every first-order oracle $(\ell_1, \nabla \ell_1), \ldots, (\ell_n, \nabla \ell_n)$. We set $(\hat{\ell}_0, \hat{q}_0, \hat{g}_0) = (\ell(w_0), q_0, \nabla \ell(w_0))$, and for iteration $t \geq 1$ update block $K_t := (n/b) \mod t + 1$ via

$$(\hat{\ell}_{t,k}, \hat{q}_{t,k}, \hat{g}_{t,k}) = \begin{cases} (\ell_k(w_t), q_{t,k}, \nabla \ell_k(w_t)) & \text{if } k \in B_{K_t} \\ (\hat{\ell}_{t-1,k}, \hat{q}_{t-1,k}, \hat{g}_{t-1,k}) & \text{if } k \notin B_{K_t} \end{cases}$$

While $q_t$ in particular is always known by the algorithm, we use the approximation $\hat{q}_t$ which is possibly *dual infeasible* for every $t \geq 1$, but will approach $q_t$ as $t$ grows large. Using this approximation is essential to controlling the per-iteration time complexity, as described below. In addition, the design of (8) and (9) is grounded in the long line of work on incremental methods (both deterministic and randomized). The table of past gradients updated cyclically resembles IAG [Blatt et al., 2007], whereas the randomized component resembles methods such as SAGA [Defazio et al., 2014] and stochastic PDHG [Chambolle et al., 2018]. The full algorithm description is given in Algorithm 1. In the next section, we show that $(a_t, C_t, c_t)_{t \geq 0}$ can be determined by a single hyperparameter.

**Computational Complexity** We also discuss the per-iteration time complexity and the global space complexity of Algorithm 1, whereas the number of required iterations for $\varepsilon$-suboptimality is given in the next section. We see that the per-iteration time complexity is $O(n + bd)$, as we query $b$ first-order oracles in both the primal and dual updates and all other operations occur on $n$-length or $d$-length vectors. While we need to employ $O(nd)$ operations to compute $\hat{g}_{t-1}^\top \hat{q}_{t-1}$ in (8) when $t = 1$, this quantity can be maintained with $O(bd)$ operations in every subsequent iteration as only $b$ rows of $\hat{q}_t$ and $\hat{g}_t$ are edited in each iteration. For space complexity, we may consider storing the entire gradient table $\hat{g}_t$ in memory, resulting in an $O(nd)$ complexity. However, due to our time complexity calculation, we may reduce the space complexity to $O(n + bd)$ by storing only $w_{t-1}, \ldots, w_{t-n/b}$ and recomputing the relevant values of $\hat{g}_t$ in (8) in every iteration. Therefore, the use of block-cyclic updates in the historical tables may significantly reduce the space complexity as compared to randomized updates (as in Defazio et al. [2014]).

## 3 Theoretical Analysis

We provide the theoretical convergence rate and global complexity of DRAGO along with technical highlights of the proof which may be of independent interest.

**Convergence Analysis** We measure suboptimality using the *primal-dual gap*:

$$\gamma_t := \mathcal{L}(w_t, q_\star) - \mathcal{L}(w_\star, q_t) - \frac{\mu}{2} \|w_t - w_\star\|_2^2 - \frac{\nu}{2} \|q_t - q_\star\|_2^2,$$

where the saddle point $(w^\star, q^\star)$ exists under Asm. 1. Note that $\gamma_t \geq 0$, as it is the sum of non-negative quantities

$$\mathcal{L}(w_t, q_\star) - \mathcal{L}(w_\star, q_\star) - \frac{\mu}{2} \|w_t - w_\star\|_2^2 \geq 0 \text{ and } \mathcal{L}(w_\star, q_\star) - \mathcal{L}(w_\star, q_t) - \frac{\nu}{2} \|q_t - q_\star\|_2^2 \geq 0.$$

To state the main result, define $\mathbb{E}_t$ as the conditional expectation over $(I_t, J_t)$ given $(w_{t-1}, q_{t-1})$ and $\mathbb{E}_1$ as the marginal expectation over the optimization trajectory. Consider the following assumptions.

**Algorithm 1** **D**istributionally **R**obust **A**nnular **G**radient **O**ptimizer (DRAGO)

---

**Input:** Sequence of constants $(a_t, C_t, c_t)_{t \geq 0}$, block size $b \in \{1, \ldots, n\}$, number of iterations $T$.

Initialize $w_0 = 0_d$, $q_0 = \mathbf{1}/n$, $\hat{\ell}_0 = \ell(w_0)$, $\hat{g}_0 = \nabla\ell(w_0)$, and $\hat{q}_0 = q_0$.

**for** $t = 1$ **to** $T$ **do**

    Sample blocks $I_t$ and $J_t$ uniformly on $[n/b]$, and define $K_t = t \mod (n/b) + 1$.

    **Primal Update:**

    Set $v_{t-1}^{\mathrm{P}} = \hat{g}_{t-1}^{\top}\hat{q}_{t-1} + \frac{a_{t-1}n}{a_t b}\sum_{i \in B_{I_t}}(q_{t-1,i}\nabla\ell_i(w_{t-1}) - \hat{q}_{t-2,i}\hat{g}_{t-2,i})$, update $w_t$ using (5).

    Update table $(\hat{\ell}_{t,k}, \hat{g}_{t,k})$ to $(\ell_k(w_t), \nabla\ell_k(w_t))$ if $k \in B_{K_t}$ or $(\hat{\ell}_{t-1,k}, \hat{g}_{t-1,k})$ if $k \notin B_{K_t}$.

    **Dual Update:**

    Set $v_t^{\mathrm{D}} = \hat{\ell}_t + \frac{a_{t-1}n}{a_t b}\sum_{j \in B_{J_t}}(\ell_j(w_t) - \hat{\ell}_{t-1,j})e_j$, update $q_t$ using (7).

    Update table $\hat{q}_{t,k}$ to $q_{t,k}$ if $k \in B_{K_t}$ or $\hat{q}_{t-1,k}$ if $k \notin B_{K_t}$.

**end for**

**return** $(w_T, q_T)$.

---

**Assumption 1.** *Let $\ell_1, \ldots, \ell_n$ be G-Lipschitz continuous and L-smooth, in that for all $i \in [n]$,*

$$\|\ell_i(w) - \ell_i(w')\|_2 \leq G\|w - w'\|_2 \text{ and } \|\nabla\ell_i(w) - \nabla\ell_i(w')\|_2 \leq L\|w - w'\|_2,$$

*i.e., each $\ell_i$ is G-Lipschitz continuous and L-smooth with respect to $\|\cdot\|_2$. Let $\mu > 0$ and $\nu > 0$, and let $q \mapsto D(q\|\mathbf{1}_n/n)$ be 1-strongly convex with respect to $\|\cdot\|_2$. Finally, $\mathcal{Q}$ is closed, convex, and contains $\mathbf{1}/n$.*

Regarding Asm. 1, an example of a loss function satisfying both Lipschitzness and smoothness is given by the Huber loss used in robust statistics [Huber, 1981]. Another setting in which both assumptions are satisfied is when the domain $\mathcal{W}$ is compact, as smoothness will imply Lipschitz continuity. Compactness is a common assumption when pursuing statistical guarantees such as uniform convergence. As for the assumption of strong convexity, we describe modifications of the algorithm when $\mu = 0$ or $\nu = 0$ in Appx. C.4 along with corresponding changes in the analysis.

**Theorem 2.** *For a constant $\alpha > 0$, define the sequence*

$$a_1 = 1, a_2 = 4\alpha, \text{ and } a_t = (1 + \alpha)a_{t-1} \text{ for } t > 2,$$

*along with its partial sum $A_t = \sum_{\tau=1}^{t} a_\tau$. Under Asm. 1, there is an absolute constant $C$ such that using the parameter*

$$\alpha = C\min\left\{\frac{b}{n}, \frac{\mu}{L\kappa_{\mathcal{Q}}}, \frac{b}{n}\sqrt{\frac{\mu\nu}{nG^2}}\right\},$$

*the iterates of Algorithm 1 satisfy:*

$$\sum_{t=1}^{T} a_t \mathbb{E}_1[\gamma_t] + \frac{A_T\mu}{4}\mathbb{E}_1\|w_T - w_\star\|_2^2 + \frac{A_T\nu}{4}\mathbb{E}_1\|q_T - q_\star\|_2^2 \leq \frac{nG^2}{\nu}\|w_0 - w_1\|_2^2.$$

*We can compute a point $(w_T, q_T)$ achieving an expected gap no more than $\varepsilon$ with big-O complexity*

$$(n + bd) \cdot \left(\frac{n}{b} + \frac{L\kappa_{\mathcal{Q}}}{\mu} + \frac{n}{b}\sqrt{\frac{nG^2}{\mu\nu}}\right) \cdot \ln\left(\frac{1}{\varepsilon}\right). \tag{10}$$

By dividing the result of Thm. 2 by $A_T$, we see that both the expected gap and expected distance-to-optimum in the primal and dual sequences decay geometrically in $T$. By plugging in $b = n/d$ we get the following runtime for Algorithm 1:

$$O\left(nd + \frac{ndL\kappa_{\mathcal{Q}}}{\mu} + n^{3/2}d\sqrt{\frac{G^2}{\mu\nu}}\right)\ln\left(\frac{1}{\varepsilon}\right). \tag{11}$$

Note in particular the individual dependence on the condition numbers $L/\mu$ and $nG^2/(\mu\nu)$, as opposed to the max-over-min-type condition numbers such as those achievable by generic primal-dual or variational inequality methods (see Appx. B.3). The proof of Thm. 2 is provided in Appx. C, along with a high-level overview in Appx. C.1. Our analysis relies on controlling $a_t\gamma_t$ by bounding $a_t\mathcal{L}(w_\star, q_t)$ above and bounding $a_t\mathcal{L}(w_t, q_\star)$ below. Key technical steps occur in the lower bound. We first apply that $w \mapsto a_t q_t^\top \ell(w)$ is convex to produce a linear underestimator of the function supported at $w_t$, and use that $w_t$ is the minimizer of a strongly convex function, so

$$a_t\mathcal{L}(w_\star, q_t) \geq a_t q_t^\top \ell(w_t) + \text{telescoping and non-positive terms}$$
$$+ a_t \left\langle \nabla\ell(w_t)^\top q_t - v_t^{\mathrm{P}}, w_\star - w_t \right\rangle \tag{12}$$
$$+ \frac{c_t\mu}{2} \sum_{\tau=t-n/b}^{t-2} \|w_t - w_{\tau\vee 0}\|_2^2 \tag{13}$$

Note that the terms above will be negated when combining the upper and lower bounds. The labeled terms are highly relevant in the analysis, with the second being non-standard. By expanding the definition of $v_t^{\mathrm{P}}$ and adding and subtracting $w_{t-1}$, we write

$$a_t \left\langle \nabla\ell(w_t)^\top q_t - v_t^{\mathrm{P}}, w_\star - w_t \right\rangle = a_t \left\langle \nabla\ell(w_t)^\top q_t - \hat{g}_{t-1}^\top \hat{q}_{t-1}, w_\star - w_t \right\rangle$$
$$- \frac{na_{t-1}}{b(1+\eta)} \sum_{i\in I_t} \left\langle \nabla\ell_i(w_{t-1})q_{t-1} - \hat{g}_{t-2,i}\hat{q}_{t-2,i}, w_\star - w_{t-1} \right\rangle$$
$$- \frac{na_{t-1}}{b(1+\eta)} \sum_{i\in I_t} \left\langle \nabla\ell_i(w_{t-1})q_{t-1} - \hat{g}_{t-2,i}\hat{q}_{t-2,i}, w_{t-1} - w_t \right\rangle.$$

When choosing the learning rate $\eta$ correctly, the first two terms will telescope in expectation (see Lem. 11). The third term, after applying Young's inequality, requires controlling $\frac{1}{b}\sum_{i\in I_t} \|\nabla\ell_i(w_{t-1})q_{t-1} - \hat{g}_{t-2,i}\hat{q}_{t-2,i}\|_2^2$ in expectation, which we dub the "primal noise bound" (Lem. 5). When we combine the upper and lower bounds, we get a similar inner product term $a_t \left\langle q_\star - q_t, \ell(w_t) - v_t^{\mathrm{D}} \right\rangle$, and mirroring the arguments above, we encounter the term $\frac{1}{b}\sum_{j\in J_t}(\ell_j(w_t) - \hat{\ell}_{t-1,j})_2^2$ which also requires a "dual noise bound" (Lem. 6). Without loss of generality, assume that the blocks are ordered such that

$$\hat{\ell}_{t-1,i} = \ell_i(w_{t-1-K\vee 0}) \text{ for } i \in B_K. \tag{14}$$

By computing the conditional expectation $\mathbb{E}_t[\cdot] := \mathbb{E}[\cdot|w_{t-1}]$, we have that

$$\mathbb{E}_t\left[\frac{1}{b}\sum_{j\in J_t}(\ell_j(w_t) - \hat{\ell}_{t-1,j})_2^2\right] = \frac{1}{n}\sum_{i=1}^{n}(\ell_i(w_t) - \hat{\ell}_{t-1,i})_2^2$$
$$= \frac{1}{n}\sum_{K=1}^{n/b}\sum_{i\in B_K}(\ell_i(w_t) - \ell_i(w_{t-1-K\vee 0}))_2^2$$
$$\leq \frac{G^2 b}{n}\sum_{\tau=t-n/b}^{t-2}\|w_{t-1} - w_{\tau\vee 0}\|_2^2,$$

where the second line follows from (14) and the third from $G$-Lipschitzness. This will telescope with the second term introduced in (13), showing the importance of the regularization. The argument follows similarly even when (14) does not hold, as the blocks can simply be "renamed" to achieve the final bound. While the proof is technical, this core idea guides the analysis and the algorithm design.

## 4 Experiments

In this section, we provide numerical benchmarks to measure DRAGO against baselines in terms of evaluations of each component $\{(\ell_i, \nabla\ell_i)\}_{i=1}^n$ and wall clock time. We consider regression and classification tasks. Letting $(x_i, y_i)$ denote a feature-label pair, we have that each $\ell_i$ represents the

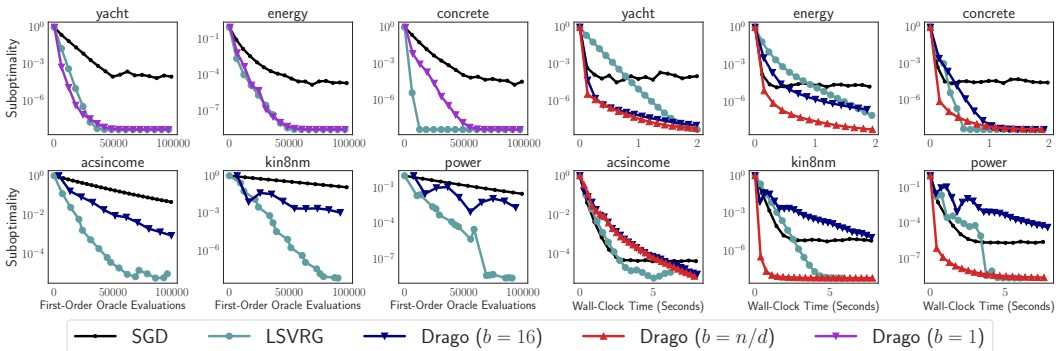

Figure 2: **Regression Benchmarks.** In both panels, the $y$-axis measures the primal suboptimality gap (15). Individual plots correspond to particular datasets. **Left:** The $x$-axis displays the number of individual first-order oracle queries to $\{(\ell_i, \nabla\ell_i)\}_{i=1}^n$. **Right:** The $x$-axis displays wall-clock time.

squared error loss or multinomial cross-entropy loss, given by

$$\ell_i(w) := \frac{1}{2}\left(y_i - x_i^\top w\right)^2 \text{ and } \ell_i(w) := -x_i^\top w_{y_i} + \log \sum_{y \in \mathcal{Y}} \exp\left(x_i^\top w_y\right),$$

respectively. In the latter case, we denote $w = (w_1, \ldots, w_C) \in \mathbb{R}^{C \times d}$, indicating multiclass classification with label set $\mathcal{Y} := \{1, \ldots, C\}$. We show results for the conditional value-at-risk (CVaR) [Rockafellar and Royset, 2013] in this section, exploring the effect of sample size $n$, dimensionality $d$, and regularization parameter $\nu$ on optimization performance. The parameter $\nu$ in particular has interpretations both as a conditioning device (as it is inversely related to the smoothness constant of $w \mapsto \max_{q \in \mathcal{Q}} \mathcal{L}(w, q)$ and as a robustness parameter, as it controls the essential size of the uncertainty set. Detailed experimental settings are contained in Appx. E, including additional experiments with the $\chi^2$-divergence ball uncertainty set. The code to reproduce these experiments can be found at https://github.com/ronakdm/drago.

We compare against baselines that can be used on the CVaR uncertainty set: distributionally robust stochastic gradient descent (SGD) [Levy et al., 2020] and LSVRG [Mehta et al., 2023]. For SGD, we use a batch size of 64 and for LSVRG we use the default epoch length of $n$. For DRAGO, we investigate the variants in which $b$ is set to 1 and $b = n/d$ a priori, as well as cases when $b$ is a tuned hyperparameter. On the $y$-axis, we plot the primal gap

$$\frac{\max_{q \in \mathcal{Q}} \mathcal{L}(w_t, q) - \mathcal{L}(w_\star, q_\star)}{\max_{q \in \mathcal{Q}} \mathcal{L}(w_0, q) - \mathcal{L}(w_\star, q_\star)}, \tag{15}$$

where we approximate $\mathcal{L}(w_\star, q_\star)$ by running LBFGS [Nocedal and Wright, 1999] on the primal objective until convergence. On the $x$-axis, we display either the exact number of calls to the first-order oracles of the form $(\ell_i, \nabla\ell_i)$ or the wall clock time in seconds. We fix $\mu = 1$ but vary $\nu$ to study its role as a conditioning parameter, which is especially important as prior work establishes different convergence rates for different values of $\nu$ (see Tab. 1).

## 4.1 Regression with Large Block Sizes

In this experiment, we consider six regression datasets, named yacht ($n = 244, d = 6$) [Tsanas and Xifara, 2012], energy ($n = 614, d = 8$) [Baressi Segota et al., 2020], concrete ($n = 824, d = 8$) [Yeh, 2006], acsincome ($n = 4000, d = 202$) [Ding et al., 2021], kin8nm ($n = 6553, d = 8$) [Akujuobi and Zhang, 2017], and power ($n = 7654, d = 4$) [Tüfekci, 2014]. In each case, there is a univariate, real-valued output. Notice that most datasets, besides acsincome, are low-dimensional as compared to their sample size. Thus, the default block size $n/d$ becomes relatively large, imposing an expensive per-iteration cost in terms of oracle queries. However, when the block size is high, the stochastic gradient estimates in each iteration have lower variance and the table components are updated more frequently, which could improve convergence in principle. The main question of this

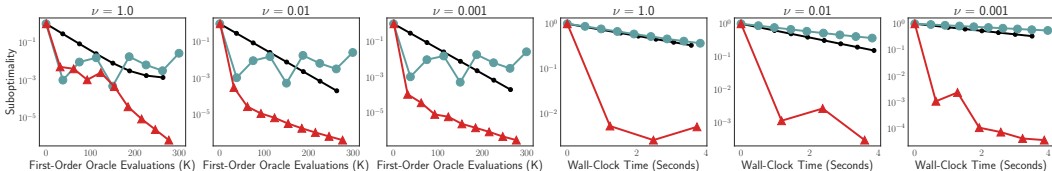

Figure 3: **Text Classification Benchmarks.** In all plots, the $y$-axis measures the normalized primal (i.e., DRO risk) suboptimality gap, defined in (15). Columns represent a varying dual regularization parameter $\nu$. On the first three columns the $x$-axis measures the number of individual first-order oracle queries to $\{(\ell_i, \nabla \ell_i)\}_{i=1}^n$ and the remaining three the $x$-axis displays wall-clock time. The objective becomes ill-conditioned as $\nu$ decreases.

section is whether DRAGO efficiently manages this trade-off via the block size parameter $b$. Results for gradient evaluations and wall clock time are on the left and right panels of Fig. 2, respectively.

**Results** The DRAGO variant for $b = n/d$ is not included on the left plot, as the number of queries (almost 2,000 in the case of power) penalizes its performance heavily. Still, the same variant performs best or near best on all datasets in terms of wall clock time (right plot). Thus, if the computation of the queries is inexpensive enough, DRAGO can achieve the lowest suboptimality within a fixed time budget. This is most striking in the case of kin8nm, in which DRAGO achieves $10^{-7}$ primal gap within 1 second, versus LSVRG which is only able to reach within $10^{-2}$ of the minimum in the same amount of time. We also experiment with tuning $b$ to reach a balance between the cost of queries and distance to optimum in the left plot with the $b = 16$ variant. In the datasets with $n \leq 1,000$, DRAGO can match the performance of baselines with only $b = 1$, whereas in the larger datasets, a batch size of 16 is needed to be comparable.

### 4.2 Text Classification Under Ill-Conditioning

We consider a natural language processing example using the emotion dataset [Saravia et al., 2018], which is a classification task consisting of six sentiment categories: sadness, anger, love, fear, joy, and surprise. To featurize the text, we fine-tune a pre-trained BERT network [Devlin et al., 2019] on a held-out set of 8,000 training examples to learn a vectorial representation. We then use a disjoint subset of 8,000 training points and apply PCA to reduce them to 45-dimensional vectors. Because of the six classes this results in $d = 270$ parameters to learn. To study the effect of dual regularization, we consider $\nu \in \{1.0, 0.01, 0.001\}$. As $\nu$ decreases, the dual solution may shift further from uniformity, and potentially increase the distributional robustness of the learned minimizer. However, the objective can also become poorly conditioned, introducing a key trade-off between optimization and statistical considerations when selecting $\nu$. The results are in Fig. 3.

**Results** The run time required for LSVRG to make 500K gradient evaluations is too large to be considered. We also observe that LSVRG is vulnerable to ill-conditioned objectives, as it is outperformed by SGD for smaller values of $\nu$ in terms of wall clock time. Within 4 seconds, DRAGO can achieve close to a $10^{-5}$ primal suboptimality gap while the gaps of SGD and LSVRG are 2 to 3 orders of magnitude larger in the same amount of time. We hypothesize that because the dual variables in LSVRG are updated once every $n$ iterations, the primal gradient estimates may accrue excessive bias. DRAGO with $b = n/d$, making $\sim 30$ individual first-order queries per iteration, is performant in terms of oracle queries and wall clock time even as $\nu$ drops by 3 orders of magnitude.

## 5 Conclusion

We proposed DRAGO, a stochastic primal-dual algorithm for solving a host of distributionally robust optimization (DRO) problems. The method achieves linear convergence without placing conditions on the dual regularizer, and its empirical performance remains strong across varying settings of the sample size $n$, dimension $d$, and the dual regularization parameter $\nu$. The method combines ideas of variance reduction, minibatching, and cyclic coordinate-style updates even though the dual feasible set (a.k.a. the uncertainty set) is non-separable. Opportunities for future work include extensions to non-convex settings and applications to min-max problems beyond distributional robustness, such as missing data imputation and fully composite optimization.

**Acknowledgements** This work was supported by NSF DMS-2023166, CCF-2019844, DMS-2134012, NIH, IARPA 2022-22072200003, U. S. Office of Naval Research under award number N00014-22-1-2348. Part of this work was done while R. Mehta and Z. Harchaoui were visiting the Simons Institute for the Theory of Computing.

**Broader Impact** Distributionally robust optimization (DRO) within machine learning is heavily motivated by problems in artificial intelligence (AI) safety, such as mitigating catastrophic performance of models on minority groups or end-users. While this work is focused on theoretical and algorithmic aspects of the problem, we intend to increase accessibility and scalability for downstream applications as well.

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

# Appendix

Appx. A contains all notation introduced throughout the paper. Appx. B discusses convergence rate comparisons in detail with contemporary work. The full convergence analysis of DRAGO is given in Appx. C. A description of the algorithm amenable to implementation is given in Appx. D, whereas experimental settings are described in detail within Appx. E.

## Table of Contents

# A    Notation

| Symbol | Description |
|---|---|
| $n$ | Sample size, or number of loss functions. |
| $d$ | Dimensionality of primal variables. |
| $\mathcal{W} \subseteq \mathbb{R}^d$ | Primal feasible set, which is closed and convex. |
| $\mathcal{Q} \in \Delta^n$ | Uncertainty set, or dual feasible set, which is closed and convex. Here, $\Delta^n = \{p \in [0,1]^n : \sum_{i=1}^n p_i = 1\}$. |
| $q_0$ | The uniform vector $q_0 = \mathbf{1}/n = (1/n, \ldots, 1/n)$. Used as a dual initialization. |
| $D(q\|q_0)$ | A statistical divergence between $q \in \mathcal{Q}$ and $q_0$, which is 1-strongly convex in its first argument with respect to $\|\cdot\|_2$. |
| $\ell$ | Loss function $\ell : \mathcal{W} \to \mathbb{R}^n$, which is differentiable in each component on an open set $\mathcal{O} \subseteq \mathbb{R}^d$ such that $\mathcal{W} \subseteq \mathcal{O}$. |
| $\mu \geq 0$ | Primal regularization constant. |
| $\nu \geq 0$ | Dual regularization constant. |
| $\mathcal{L}(w,q)$ | Objective function $\mathcal{L}(w,q) := q^\top \ell(w) - \nu D(q\|q_0) + \frac{\mu}{2}\|w\|_2^2$. |
| $(w_\star, q_\star)$ | Saddle point of $\mathcal{L}$, which satisfies $\mathcal{L}(w_\star, q) \leq \mathcal{L}(w_\star, q_\star) \leq \mathcal{L}(w, q_\star)$ for all $(w, q) \in \mathcal{W} \times \mathcal{Q}$. |
| $[n]$ | Index set $[n] = \{1, \ldots, n\}$. |
| $q_{\max}$ | The value $\max_{q \in \mathcal{Q}} \|q\|_\infty$. |
| $\kappa_{\mathcal{Q}}$ | The condition number $\kappa_{\mathcal{Q}} = nq_{\max} \geq 1$. |
| $G$ | Lipschitz continuity constant of each $\ell_i$ for $i \in [n]$, for which $|\ell_i(w) - \ell_i(w')| \leq G\|w - w'\|_2$. |
| $L$ | Lipschitz continuity constant of each $\nabla \ell_i$ for $i \in [n]$, for which $\|\nabla \ell_i(w) - \nabla \ell_i(w')\|_2 \leq L\|w - w'\|_2$. |
| $\nabla \ell(w)$ | Jacobian matrix of $\ell : \mathbb{R}^d \to \mathbb{R}^n$ at $w$ (shape $= n \times d$). |
| $(a_t)$ | Sequence of positive constants that weigh the average gap criterion. |
| $(A_t)$ | Sequence of partial sums of $(a_t)$, or $A_t = \sum_{s=1}^t a_s$. The convergence rate will be given by $A_t^{-1}$, so we have $a_t$ increase geometrically. |
| $b$ | Batch or block size. |
| $M$ | Number of blocks $n/b$. |
| $(w_t, q_t)_{t \geq 0}$ | Sequence of primal and dual iterates. |
| $e_j$ | The $j$-th standard basis vector $e_j \in \{0,1\}^n$. |
| $\hat{\ell}_t$ | Loss table, which approximates $\ell(w_t) \in \mathbb{R}^n$. |
| $\hat{g}_t$ | Gradient table, which approximates $\nabla \ell(w_t) \in \mathbb{R}^{n \times d}$. |
| $\hat{q}_t$ | Weight table, which approximates $q_t \in \mathcal{Q}$. |
| $\mathbb{E}_t[\cdot]$ | Shorthand for $\mathbb{E}[\cdot \mid \mathcal{H}_{t-1}]$, i.e., expectation conditioned on history $\mathcal{H}_{t-1} = \sigma\left(\{(I_s, J_s)\}_{s=1}^{t-1}\right)$. |

Table 2: Notation used throughout the paper.

# B Comparisons to Existing Literature

In this appendix, we compare our work to existing literature along two axes: 1) distributional robust optimization (DRO), and 2) primal-dual algorithms for saddle-point problems. In the first category, we are primarily concerned with questions of practical and statistical interest, such as which uncertainty sets can be used, how the size of the uncertainty set affects the convergence rate, and what assumptions are needed on the distribution of losses. In the second category, we discuss computational complexity under various assumptions such as smoothness and strong convexity of the objective.

## B.1 Directly Using Gradient Descent

In the penalized case, we add that the objective $w \mapsto \max_{q \in \mathcal{Q}} \mathcal{L}(w, q)$ is $(L + \mu + \frac{nG^2}{\nu})$-smooth when the losses $\ell_i$ are $G$-Lipschitz continuous and $L$-smooth. For this reason, we may consider simply applying full-batch gradient descent to this objective, which is included in our comparisons. To see why this smoothness condition holds, define $h(l) := \max_{q \in \mathcal{Q}} \left\{ q^\top l - \nu D(q \| \mathbf{1}/n) \right\}$, so that when $q \mapsto D(q \| \mathbf{1}/n)$ is 1-strongly convex with respect to $\|\cdot\|_2^2$, it holds that $\nabla h$ is $(1/\nu)$-Lipschitz continuous with respect to $\|\cdot\|_2^2$, and that $\nabla h(l)$ is non-negative and sums to one (i.e. is a probability mass function on $[n]$). Then, by the chain rule, for any $w_1, w_2 \in \mathcal{W}$, we have that

$$
\begin{aligned}
&\left\| \nabla \ell(w_1)^\top \nabla h(\ell(w_1)) - \nabla \ell(w_2)^\top \nabla h(\ell(w_2)) \right\|_2 \\
&\leq \left\| \nabla \ell(w_1)^\top (\nabla h(\ell(w_1)) - \nabla h(\ell(w_2))) \right\|_2 + \left\| (\nabla \ell(w_1) - \nabla \ell(w_2))^\top \nabla h(\ell(w_2)) \right\|_2 ] \\
&\leq \frac{nG}{\nu} \left\| \ell(w_1) - \ell(w_2) \right\|_2 + L \left\| w_1 - w_2 \right\|_2 \\
&\leq \left( \frac{nG^2}{\nu} + L \right) \left\| w_1 - w_2 \right\|_2 .
\end{aligned}
$$

Thus, when referring to the gradient descent on $w \mapsto \max_{q \in \mathcal{Q}} \mathcal{L}(w, q) = h(\ell(w)) + \frac{\mu}{2} \|w\|_2^2$, we reference the smoothness constant $\left( L + \mu + \frac{nG^2}{\nu} \right)$.

## B.2 Distributionally Robust Optimization (DRO)

**Examples of DRO Problems**   Our problem

$$
\min_{w \in \mathcal{W}} \max_{q \in \mathcal{Q}} \left[ \mathcal{L}(w, q) := q^\top \ell(w) - \nu D(q \| \mathbf{1}/n) + \frac{\mu}{2} \|w\|_2^2 \right] \tag{16}
$$

accommodates several settings of interest across machine learning. For example, $f$-DRO with parameter $\rho$ [Namkoong and Duchi, 2016, Carmon and Hausler, 2022] results by defining the uncertainty set and penalty as

$$
\mathcal{Q} = \mathcal{Q}(\rho) := \left\{ q : D_f(q \| \mathbf{1}/n) \leq \frac{\rho}{n} \right\} \text{ and } D(q \| \mathbf{1}/n) = D_f(q \| \mathbf{1}/n),
$$

where $D_f(q \| p) = \sum_{i=1}^n p_i f(q_i / p_i)$ denotes an $f$-divergence generated by $f$ (which is always well-defined when $p_i = \mathbf{1}/n$). Common examples include the Kullback-Leibler (KL) divergence, generated by $f_{\text{KL}}(x) = -x \log x$ and the $\chi^2$-divergence generated by $f_{\chi^2}(x) = (x-1)^2$. Spectral risk measures [Mehta et al., 2023] are parametrized by an $n$-length vector $\sigma = (\sigma_1, \ldots, \sigma_n)$ such that $0 \leq \sigma_1 \leq \ldots \leq \sigma_n$ and $\sum_{i=1}^n \sigma_i = 1$. The penalty in that setting may also be in the form of an $f$-divergence [Mehta et al., 2024], so that

$$
\mathcal{Q} = \mathcal{Q}(\sigma) := \text{conv} \left( \{ \text{permutations of } \sigma) \} \right) \text{ and } D(q \| \mathbf{1}/n) = D_f(q \| \mathbf{1}/n),
$$

where $\text{conv}(\cdot)$ denotes the convex hull. The most notable example of such an uncertainty set is the $\theta$-conditional value-at-risk, or CVaR [Rockafellar and Royset, 2013], wherein the largest $\theta n$ values of $\sigma$ are set to $1/(n\theta)$ and the remaining are set to zero (with a fractional component if $\theta n$ is not an integer). Finally, Wasserstein-DRO [Kuhn et al., 2019, Blanchet et al., 2019, Yu et al., 2022] with parameter $\delta$ typically sets the uncertainty set to be a $\delta$-ball $\{ Q : W_c(Q, P_n) \leq \delta \}$ in Wasserstein distance, where $Q$ is a probability measure, $P_n$ is the empirical measure of the training data, and

$W_c$ is the Wasserstein distance associated to cost function $c$. This differs from $f$-DRO and spectral risk minimization because in the latter settings, the "shifted" distribution $Q$ is assumed to remain on the same $n$ atoms as before, so that it may simply be specified by a probability vector $q \in [0, 1]^n$ (resp. $P_n$ by $\mathbf{1}/n$). However, as shown by Yu et al. [2022], Wasserstein-DRO can be reformulated into a problem of the form (16) if the following conditions are satisfied: 1) the loss is of a generalized linear model $\ell_i(w) = \Psi(\langle w, x_i \rangle, y_i)$ with feature-label pair $(x_i, y_i)$ and discrepancy function $\Psi$, 2) the cost function $c((x, y), (x', y'))$ is of the form $\|x - x'\|_2 + \beta |y - y'|$ for $\beta > 0$, and 3) the function $\Psi$ is Lipschitz continuous with known constant.

**Comparison of DRO Approaches**  The performance of classical and recent algorithms designed for the problems above is detailed in Tab. 3. The rightmost column displays the **oracle complexity**, meaning the number of queries to individual/component loss functions $\{(\ell_i, \nabla \ell_i)\}$ as a function of the desired suboptimality $\varepsilon$. The desiderata in the large-scale setting is to decouple the contribution of the sample size $n$ and the condition number in smooth, strongly convex settings ($\mu > 0$ and $L < \infty$) or the quantity $1/\varepsilon$ in non-smooth or non-strongly convex settings. For readability, we encode dependence on $n$ in red and dependence on $1/\varepsilon$ in blue within Tab. 3. In certain cases, such as in the sub-gradient method, the dependence on $n$ is understood as part of a smoothness constant. Similarly, because $q_{\max}$ is of order $1/n$ in the best case and $1$ in the worst case, we interpret $\kappa_{\mathcal{Q}}$ to play the role of a condition number that measures the size of the uncertainty set. That being said, $\kappa_{\mathcal{Q}}$ is upper bounded by a constant independent of $n$ in common cases. We collect examples of these results below.

**Proposition 3.** *In* (16) *if $\mathcal{Q}$ is chosen to be the CVaR uncertainty set with tail probability $\theta$, then $\kappa_{\mathcal{Q}} \leq \frac{1}{\theta}$. If $\mathcal{Q}$ is chosen to be the $\chi^2$-DRO uncertainty set with radius $\rho$, then $\kappa_{\mathcal{Q}} \leq \sqrt{1 + \rho}$.*

*Proof.* For the $\theta$-CVAR, we have that $q_i \in [0, 1/(n\theta)]$ for all $i \in [n]$. Thus,

$$nq_i \leq \frac{1}{\theta} \text{ for all } i = 1, \ldots, n.$$

For the $\chi^2$-DRO uncertainty set, we have by direct computation that

$$n \|q - \mathbf{1}/n\|_2^2 \leq \frac{\rho}{n} \iff \|q\|_2^2 \leq \frac{1 + \rho}{n^2}.$$

This implies that for any $i \in [n]$, we have that $q_i^2 \leq (1 + \rho)/n^2$ which implies that

$$nq_i \leq \sqrt{1 + \rho},$$

the result as desired. $\qquad \square$

The closest comparison to our setting is that of LSVRG [Mehta et al., 2023] and Prospect [Mehta et al., 2024]. Including DRAGO, these methods all achieve linear convergence on (16), under the assumption of strongly convex regularization in the primal objective. However, both LSVRG and Prospect demand a stringent lower bound of $\Omega(nG^2/\mu)$ on the dual regularization parameter $\nu$ to achieve this rate, which essentially matches ours in this regime (as $\sqrt{nG^2/(\mu\nu)}$ reduces to a constant). When this condition does not hold, however, LSVRG does not have a convergence guarantee while Prospect underperforms against the sub-gradient method. Furthermore, while Propsect has a single hyperparameter, LSVRG must search over both a learning rate and an epoch length. DRAGO, on the other hand, is the only method that achieves unconditional linear convergence and fully captures the dependence on the dual regularization parameter $\nu$, which is a smoothness measure of the objective $\max_q \mathcal{L}(\cdot, q)$.

Moving to methods that are not linearly convergent, Levy et al. [2020] consider a variant of mini-batch SGD that solves the maximization problem within the mini-batch itself. In Tab. 3, we include a term $\min \{n, b(\varepsilon)\}$, where $b(\varepsilon)$ denotes a required batch size. This is because Levy et al. [2020] measures complexities in terms of calls to first-order oracles for a loss summed across an *arbitrary* batch size. Thus, our comparison relies on multiplying the required batch size with the number of iterations required for a desired suboptimality (see Levy et al. [2020, Theorem 2], for example). In the setting in which there is a fixed training set of size $n$, we incorporate this requirement on the batch size in the complexity bound, to account for the case in which the theoretically required batch size grows so large that it exceeds the full batch size itself. Indeed, as commented by Carmon

and Hausler [2022], when the uncertainty set is large (i.e. $\theta$ is small for CVaR and $\rho$ is large for $\chi^2$-divergence), the method may underperform against the sub-gradient method. This is true of methods such as in Mehta et al. [2024] and ours that depend on $\kappa_{\mathcal{Q}} \leq n$. This indicates that across DRO settings, increased uncertainty set sizes can bring the performance of incremental methods arbitrarily close to that of the full-batch sub-gradient method. Finally, notice that DRAGO also has a dependence on the batch size $b$ in Tab. 3. While it would appear that $b = 1$ would minimize oracle complexity, we describe in the next section how $b$ can be chosen to minimize global complexity, which includes the cost of each oracle call as a function of the primal dimension $d$. Because of the diversity of settings, a coarser measure such as first-order oracle evaluations may be sufficient to compare methods along the DRO axis; we use the finer-grained global complexity to compare methods along the axis of saddle-point algorithms.

## B.3 Primal-Dual Saddle Point Algorithms

For context, both (1) and (2) are coupled min-max problems with primal-dual variables $(w, q)$. Observe that the coupled term in the objective (depending on both $w$ and $q$) is not bilinear in general. Such objectives have received much less attention in the optimization literature on *primal-dual* methods than their bilinearly-coupled counterparts. For the bilinear setting, methods such as stochastic Chambolle-Pock-style algorithms [He et al., 2020, Song et al., 2021] use stochastic variance-reduced updates, and in particular, employ coordinate-style updates of a single dual variable per iteration. To illustrate the advantage of these updates, notice that the primal and dual dimensions are $d$ and $n$, respectively. When $n$ is significantly larger than $d$, and assuming that each $(\ell_i, \nabla \ell_i)$ call comes at an $O(d)$ cost, updating the dual variables at $O(n)$ cost every iteration becomes the primary computational bottleneck. Coordinate-style updates can eliminate this dependence, but apply only when the dual feasible set $\mathcal{Q}$ decomposes as $\mathcal{Q}_1 \times \ldots \times \mathcal{Q}_n$, i.e., is *separable*.

There is a long line of work on variance-reduced algorithms for general stochastic optimization problems [Johnson and Zhang, 2013, Defazio et al., 2014, Palaniappan and Bach, 2016, Cai et al., 2023] and structured and/or bilinearly coupled min-max problems [Yang et al., 2020, Song et al., 2021, Du et al., 2022, Kovalev et al., 2022, Alacaoglu et al., 2022]. However, few of these results can be directly applied to (2) due to non-bilinearity and non-separability of the dual feasible set. These issues motivated work on reformulating common DRO problems as bilinearly coupled min-max problems in Song et al. [2022]. However, the guarantees obtained in Song et al. [2022] are of the order $O(\varepsilon^{-1})$.

Another viewpoint is that (2) can be written as a monotone variational inequality (VI) problem for the operator $F(w, q) = (\nabla_w \ell(w)^\top q + \mu w, -\ell(w) + \nu \nabla_q D(q\|\mathbf{1}/n)) \in \mathbb{R}^{d+n}$, where we assumed 1-smoothness of $D$ for ease of presentation. Under our assumptions, this operator is $\min\{\mu, \nu\}$-strongly monotone and $\max\{nG^2, L\}$-Lipschitz continuous, and VI algorithms such as mirror-prox, Popov's method, and variance-reduced methods like Alacaoglu and Malitsky [2022], Cai et al. [2024] can be used. However, the max-over-min dependence on the individual Lipschitz constants and strong convexity constants is unfavorable compared to the individual condition numbers observed in (4). If $F$ is called directly, the global complexity will be $n^2$ because $q$ is $n$-dimensional along with $n$ in the condition number. A finite sum approach could improve dependence on the number of oracle calls, but the global complexity would still be $n^2$ due to the non-separability of the dual feasible set and again the $n$-dimensional dual variables.

To summarize the points above and in order to make comparisons among methods for general min-max problems, we first collect aspects of (16) that make it a highly specialized problem in this regard. The major points include:

1. The objective has a *finite-sum* structure, in that it can be written as $\sum_{i=1}^n f_i(w, q)$, where $f_i(w, q) := q_i \ell_i(w) - \nu D(q\|\mathbf{1}/n) + \frac{\mu}{2} \|w\|_2^2$.

2. The dimension of the dual variable $q$ is equal to $n$, the number of functions in the sum (i.e. the sample size), and could be much larger than $d$.

3. The dual regularizer $q \mapsto D(q\|\mathbf{1}/n)$ is not necessarily smooth. This encompasses common statistical divergences such as the Kullback Leibler (KL).

4. The dual feasible set $\mathcal{Q}$ is non-separable, as any feasible dual iterate must sum to 1.

For discussions in which linear convergence is a pre-requisite, it is typically assumed that $(w, q) \mapsto \mathcal{L}(w, q)$ is a smooth map, and finer-grained results depend on the individual Lipschitz continuity parameters of $w \mapsto \nabla_w \mathcal{L}(w, q)$, $q \mapsto \nabla_q \mathcal{L}(w, q)$, $w \mapsto \nabla_q \mathcal{L}(w, q)$, and $q \mapsto \nabla_w \mathcal{L}(w, q)$. To make the regularized DRO setting of (16) comparable to classical and contemporary saddle-point methods, we make the additional assumption in this section that the (rescaled) $\chi^2$-divergence penalty is used, so that $D(q \| \mathbf{1}/n) = \frac{1}{2} \| q - \mathbf{1}/n \|_2^2$ and we may compute the smoothness constants as

$$\| \nabla_w \mathcal{L}(w, q) - \nabla_w \mathcal{L}(w', q) \|_2 \leq (L + \mu) \| w - w' \|_2$$

$$\| \nabla_w \mathcal{L}(w, q) - \nabla_w \mathcal{L}(w, q') \|_2 = \left\| \nabla \ell(w)^\top (q - q') \right\|_2 \leq \sqrt{n} G \| q - q' \|_2$$

$$\| \nabla_q \mathcal{L}(w, q) - \nabla_q \mathcal{L}(w, q') \|_2 \leq \nu \| q - q' \|_2 \tag{17}$$

$$\| \nabla_q \mathcal{L}(w, q) - \nabla_q \mathcal{L}(w', q) \|_2 = \| \ell(w) - \ell(w') \|_2 \leq \sqrt{n} G \| w - w' \|_2 \tag{18}$$

$$\left\| \nabla_{(w,q)} \mathcal{L}(w, q) - \nabla_{(w,q)} \mathcal{L}(w', q') \right\|_2 \leq ((L + \mu) \vee \nu + \sqrt{n} G) \sqrt{2} \| (w, q) - (w', q') \|_2 . \tag{19}$$

In making this assumption, however, we emphasize that our results *do not depend on* (17), (18), *and* (19) *being true*. We assume here that calls to the oracles $(\ell_i, \nabla_w \ell_i)$ cost $O(d)$ operations while calls to $\nabla_q \mathcal{L}$ cost $O(n)$ operations, making the total $O(n + d)$. This per-iteration complexity is a subtlety of DRO which is essential to recognize when comparing methods. With DRAGO, we also have a batch size parameter $b$, setting the complexity to $O(n + bd)$. By tuning $b$, we may achieve improvements in terms of global complexity over standard primal-dual methods. Results comparing linearly convergent methods in terms of the global complexity of elementary operations are given in Tab. 4. The table contains a "half-life" column, which is the constant $\tau$ multiplied with $\log(1/\varepsilon)$ when describing the number of iterations needed to reach $\varepsilon$-suboptimality as $\tau \log(1/\varepsilon)$. Before comparisons, observe that the optimal batch size for DRAGO is $b = n/d$, in the sense that the global number of arithmetic operations is of order

$$n \frac{L \kappa_\mathcal{Q}}{\mu} + nd \sqrt{\frac{n G^2}{\mu \nu}}$$

This is an improvement over setting $b = 1$, in which case the same number is

$$(n + d) \frac{L \kappa_\mathcal{Q}}{\mu} + n(n + d) \sqrt{\frac{n G^2}{\mu \nu}}$$

Next, the most comparable and recent setting is that of Li et al. [2023]. Notice that DRAGO is able to improve by a factor of $d$ on the $\frac{L}{\mu}$ term, as long as $\kappa_\mathcal{Q} = O(1)$.

While less comparable, we mention known lower bounds for completeness. In terms of lower bounds, since (16) enjoys a particular structure, such as decomposability into so-called *marginal* terms $\nu D(q \| \mathbf{1}/n)$ and $\frac{\mu}{2} \| w \|_2^2$ and a *coupled* term $q^\top \ell(w)$, we are not necessarily constrained by the more general lower bounds of Zhang et al. [2022] and Xie et al. [2020]. For example, the proximal incremental first-order (PIFO) model of Xie et al. [2020] assumes that we observe first-order information from a single component in the sum; in DRAGO, using a batch size of $n/d$ is required to achieve the desired improvement. Zhang et al. [2022], on the other hand, do not treat the finite sum class of problems considered here. Furthermore, we still list the per-iteration cost as $O(n + d)$ because a single PIFO call to the dual gradient is of size $n$.

| Method | Assumptions | Uncertainty Set | Runtime / Global Complexity (Big-$\tilde{O}$) |
|---|---|---|---|
| Sub-Gradient Method | $\ell_i$ is $G$-Lipschitz
$\|w - w'\|_2 \le R$ (if included)
$\mu > 0$ (if included)
$\nabla \ell_i$ is $L$-Lipschitz and $\nu > 0$ (if included) | Support Constrained
Support Constrained
Support Constrained | $nd \cdot (GR)^2 \varepsilon^{-2}$
$nd \cdot G^2 \mu^{-1} \varepsilon^{-1}$
$nd \cdot \mu^{-1} \left(L + \mu + nG^2/\nu\right) \log(1/\varepsilon)$ |
| $\mathcal{L}_{\text{CVaR}}\text{-SGD}^{\dagger}$
$\mathcal{L}_{\chi^2}\text{-SGD}^{\dagger}$
$\mathcal{L}_{\chi^2-\text{pen}}\text{-SGD}^{\dagger}$
[Levy et al., 2020] | $\ell_i$ is $G$-Lipschitz and in $[0, B]$
$\|w - w'\|_2 \le R$ for all $w, w' \in \mathcal{W}$
$\nu > 0$ (if included) | $\theta$-CVaR
$\rho$-ball in $\chi^2$-divergence
$\chi^2$-divergence penalty | $\min\left\{n, B^2\theta^{-1}\varepsilon^{-2}\right\} d \cdot (GR)^2 \varepsilon^{-2}$
$\min\left\{n, (1+\rho)B^2\varepsilon^{-2}\right\} d \cdot (GR)^2 \varepsilon^{-2}$
$\min\left\{n, B^2\nu^{-1}\varepsilon^{-1}\right\} d \cdot (GR)^2 \varepsilon^{-2}$ |
| BROO*
BROO*
[Carmon and Hausler, 2022] | $\ell_i$ is $G$-Lipschitz
$\|w - w'\|_2 \le R$ for all $w, w' \in \mathcal{W}$
$\nabla \ell_i$ is $L$-Lipschitz (if included) | 1-ball in $f$-divergence
1-ball in $f$-divergence | $nd \cdot (GR)^{2/3}\varepsilon^{-2/3} + d(GR)^2 \varepsilon^{-2}$
$nd \cdot (GR)^{2/3}\varepsilon^{-2/3} + n^{3/4}d \left(GR\varepsilon^{-1} + L^{1/2}R\varepsilon^{-1/2}\right)$ |
| LSVRG
LSVRG
[Mehta et al., 2023] | $\ell_i$ is $G$-Lipschitz
$\nabla \ell_i$ is $L$-Lipschitz
$\mu > 0, \nu > 0, \kappa := (L+\mu)/\mu$ | Spectral Risk Measures ($\nu$ small)
Spectral Risk Measures ($\nu \ge \Omega(nG^2/\mu)$) | None
$(n + \kappa_{\mathcal{Q}}\kappa)\, d \log(1/\varepsilon)$ |
| Prospect
Prospect
[Mehta et al., 2024] | $\ell_i$ is $G$-Lipschitz
$\nabla \ell_i$ is $L$-Lipschitz
$\mu > 0, \nu > 0, \kappa := (L+\mu)/\mu, \delta = G^2/(\mu\nu)$ | Spectral Risk Measures ($\nu$ small)
Spectral Risk Measures ($\nu \ge \Omega(nG^2/\mu)$) | $n(n+d) \max\left\{n\delta + \kappa n q_{\max}, n^3\delta^2\kappa^2, n^3\delta^3\right\} \log(1/\varepsilon)$
$(n + \kappa_{\mathcal{Q}}\kappa)\,(n + d) \log(1/\varepsilon)$ |
| DRAGO (Ours) | $\ell_i$ is $G$-Lipschitz
$\nabla \ell_i$ is $L$-Lipschitz
$\mu > 0, \nu > 0, b :=$ Batch Size | Support Constrained | $n\left(d + \kappa_{\mathcal{Q}}L/\mu + d\sqrt{nG^2/(\mu\nu)}\right) \log(1/\varepsilon)$ |

Table 3: Replicate of Tab. 1.

| Method | Assumptions | Half-Life | Per-Iteration Cost |
|---|---|---|---|
| Sub-Gradient Method | | $\frac{L+\mu}{\mu} + \frac{nG^2}{\mu\nu}$ | $nd$ |
| Minimax-APPA [Lin et al., 2020] | | $\frac{(L+\mu)\vee\nu+\sqrt{n}G}{\sqrt{\mu\nu}}$ | $nd$ |
| Lower Bound [Xie et al., 2020] | Finite Sum / Single component oracle / May not be decomposable | $n + \frac{(L+\mu)\vee\nu+\sqrt{n}G}{\min\{\mu,\nu\}}$ | $n+d$ |
| Lower Bound [Zhang et al., 2022] | May not be decomposable / May not be finite sum | $\sqrt{\frac{L+\mu}{\mu} + \frac{nG^2}{\mu\nu}}$ | $nd$ |
| AG-OG with Restarts [Li et al., 2023] | | $\frac{L}{\mu} \vee \sqrt{\frac{nG^2}{\mu\nu}}$ | $nd$ |
| Drago ($b=1$) | | $\frac{L\kappa_{\mathcal{Q}}}{\mu} + n\sqrt{\frac{nG^2}{\mu\nu}}$ | $n+d$ |
| Drago ($b=n/d$) | $D(\cdot\|\mathbf{1}/n)$ need not be smooth | $\frac{L\kappa_{\mathcal{Q}}}{\mu} + d\sqrt{\frac{nG^2}{\mu\nu}}$ | $n$ |
| Drago ($b=n$) | | $\frac{L\kappa_{\mathcal{Q}}}{\mu} + \sqrt{\frac{nG^2}{\mu\nu}}$ | $nd$ |

Table 4: **Complexity of Primal-Dual Saddle Point Methods.** Half-life (defined as $\tau$ such that a linearly convergent method requires $O(\tau\log(1/\varepsilon))$ iterations to achieve $\varepsilon$-suboptimality) and per-iteration cost of linearly convergent optimization algorithms. The global number of arithmetic operations (under the assumption that $(\ell_i, \nabla_w\ell_i)$ costs $O(d)$ operations and $\nabla_q\mathcal{L}$ costs $O(n)$ operations) required to achieve a point $(w,q)$ satisfying $\mathcal{L}(w,q_\star) - \mathcal{L}(w_\star,q) \leq \varepsilon$ can be computed by multiplying the last two columns. The "Assumptions" column contains changes to the assumptions that each $\ell_i$ is $L$-smooth and that $D(\cdot\|\mathbf{1}/n)$ is $\nu$-smooth.

# C  Convergence Analysis of DRAGO

This section provides the convergence analysis for DRAGO. We first recall the quantities of interest and provide an alternate description of the algorithm that is useful for understanding the analysis. A high-level overview is given in Appx. C.1, and the remaining subsections comprise steps of the proof.

## C.1  Overview

See Tab. 2 for a reference on notation used throughout the proof, which will also be introduced as it appears. Define $q_0 = \mathbf{1}/n$ and recall the objective

$$\mathcal{L}(w, q) := q^\top \ell(w) - \nu D(q \| q_0) + \frac{\mu}{2} \|w\|_2^2. \tag{20}$$

By strong convexity of $w \mapsto \max_q \mathcal{L}(w, q)$ with respect to $\|\cdot\|_2$ and strong concavity of $q \mapsto \min_w \mathcal{L}(w, q)$ with respect to $\|\cdot\|_2$, a primal-dual solution pair $(w^\star, q^\star)$ is guaranteed to exist and is unique, and thus we define

$$w_\star = \arg\min_{w \in \mathcal{W}} \max_{q \in \mathcal{Q}} \mathcal{L}(w, q) \text{ and } q_\star = \arg\max_{q \in \mathcal{Q}} \min_{w \in \mathcal{W}} \mathcal{L}(w, q).$$

In other words, we have that $\max_{q \in \mathcal{Q}} \mathcal{L}(w_\star, q) = \mathcal{L}(w_\star, q_\star) = \min_{w \in \mathcal{W}} \mathcal{L}(w, q_\star)$. We proceed to describe the algorithm, the optimality criterion, and the proof outline.

**Algorithm Description**   First, we describe two sequences of parameters that are used to weigh various terms in the primal and dual updates. The parameters are set in the analysis, and the version of the algorithm in Appx. D is a description with these values plugged in. However, we keep them as variables in this section to better describe the logic of the proof.

Specifically, let $(a_t)_{t \geq 1}$ be a sequence of positive numbers and define $a_0 = 0$ in addition. Denote $A_t = \sum_{s=1}^{T} a_s$. These will become the averaging sequence that will aggregate successive gaps $\gamma_t$ (see (27)) into the return value $\sum_{t=1}^{T} a_t \gamma_t$, which will be upper bounded by a constant in $T$ (in expectation). We also define another similar sequence $(c_t)_{t \geq 1}$, and define $C_t = A_t - (n/b - 1)c_t$ for batch size $b$. We assume for simplicity that $b$ divides $n$. When all constants are set, the algorithm will reduce to that given in Algorithm 1. We have one hyperparameter $\alpha > 0$, which may be interpreted as a learning rate.

Using initial values $w_0 = 0$ and $q_0 = \mathbf{1}/n$, initialize the tables $\hat{\ell}_0 = \ell(w_0) \in \mathbb{R}^n$, $\hat{g}_0 = \nabla \ell(w_0) \in \mathbb{R}^{n \times d}$, and $\hat{q}_0 = q_0 \in \mathbb{R}^n$. In addition, partition the $[n]$ sample indices into $M := n/b$ blocks of size $b$, or $B_1, \ldots, B_M$ with $B_K := ((K-1)b + 1, \ldots, Kb)$ for $K \in [M]$. We can set the averaging sequence according to the following scheme:

$$a_1 = 1, a_2 = 4\alpha, \text{ and } a_t = (1 + \alpha) a_{t-1} \text{ for } t > 2.$$

The initial value $a_2 = 4\alpha$ is a slight modification for theoretical convenience, and the algorithm operates exactly as in Appx. D in practice. In order to retrieve the Appx. D version, we simply replace the condition above with $a_t = (1 + \alpha) a_{t-1}$ for $t \geq 2$.

Consider iterate $t \in \{1, \ldots, T\}$. We sample a random block $I_t$ uniformly from $[M]$ and compute the primal update.

$$\delta_t^{\mathrm{P}} := \frac{1}{b} \sum_{i \in B_{I_t}} \left( q_{t-1,i} \nabla \ell_i(w_{t-1}) - \hat{q}_{t-2,i} \hat{g}_{t-2,i} \right) \tag{21}$$

$$v_t^{\mathrm{P}} := \hat{g}_{t-1}^\top \hat{q}_{t-1} + \frac{n a_{t-1}}{a_t} \delta_t^{\mathrm{P}} \tag{22}$$

$$w_t := \arg\min_{w \in \mathcal{W}} a_t \left\langle v_t^{\mathrm{P}}, w \right\rangle + \frac{a_t \mu}{2} \|w\|_2^2 + \frac{C_{t-1} \mu}{2} \|w - w_{t-1}\|_2^2 + \frac{c_{t-1} \mu}{2} \sum_{s=t-n/b}^{t-2} \|w - w_{s \vee 0}\|_2^2 \tag{23}$$

We see that $C_{t-1} + c_{t-1}(n/b - 1) = A_{t-1}$, so the inner objective of the update (23) is $A_t\mu$-strongly convex (as the one in (26) is $A_t\nu$-strongly concave). Note that when $n/b < 1$, we simply treat the method as not including the additional regularization term $\frac{c_{t-1}\mu}{2}\sum_{s=t-n/b}^{t-2}\|w - w_{s\vee 0}\|_2^2$. Proceeding, we then modify the loss and gradient table. The loss update has to occur between the primal and dual updates to achieve control over the variation in the dual update (see Appx. C.2). Define $K_t = t \mod M + 1$ as the (deterministic) block to be updated, and set

$$(\hat{\ell}_{t,k}, \hat{g}_{t,k}) = \begin{cases} (\ell_k(w_t), \nabla \ell_k(w_t)) & \text{if } k \in B_{K_t} \\ (\hat{\ell}_{t-1,k}, \hat{g}_{t-1,k}) & \text{otherwise} \end{cases}.$$

Define $e_j$ to be the $j$-th standard basis vector. Then, sample a random block $J_t$ uniformly from $[M]$ compute

$$\delta_t^{\mathrm{D}} := \frac{1}{b} \sum_{j \in B_{J_t}} (\ell_j(w_t) - \hat{\ell}_{t-1,j}) e_j \tag{24}$$

$$v_t^{\mathrm{D}} := \hat{\ell}_t + \frac{n a_{t-1}}{a_t} \delta_t^{\mathrm{D}} \tag{25}$$

$$q_t := \arg\max_{q \in \mathcal{Q}} a_t \langle v_t^{\mathrm{D}}, q \rangle - a_t \nu D(q\|q_0) - A_{t-1}\nu\Delta_D(q, q_{t-1}). \tag{26}$$

Notice the change in indices between (21) and (25), which accounts for the update in the loss table that occurs in between. Finally, we must update the remaining table. Set

$$\hat{q}_{t,k} = \begin{cases} q_{t,k} & \text{if } k \in B_{K_t} \\ \hat{q}_{t-1,k} & \text{otherwise} \end{cases}.$$

We define the random variable $\mathcal{H}_t := \{(I_s, J_s)\}_{s=1}^t$ as the history of blocks selected at all times up to and including $t$, and define $\mathbb{E}_t[\cdot]$ to be the conditional expectation operator given $\mathcal{H}_{t-1}$. In other words, $\mathbb{E}_t$ integrates the randomness $\{(I_s, J_s)\}_{s=t}^T$. Accordingly $\mathbb{E}_1[\cdot]$ is the marginal expectation of the entire random process. We may now describe the optimality criterion.

**Proof Outline** Construct the gap function

$$\gamma_t = a_t \left( \mathcal{L}(w_t, q_\star) - \mathcal{L}(w_\star, q_t) - \frac{\mu}{2}\|w_t - w_\star\|_2^2 - \frac{\nu}{2}\|q_t - q_\star\|_2^2 \right) \tag{27}$$

and aim to bound $\mathbb{E}_1\left[\sum_{s=1}^t \gamma_s\right]$ by a constant. Throughout the proof, we will use a free parameter $\alpha$, with update rule

$$a_t \leq \min\left\{ \left(1 + \tfrac{\alpha}{4}\right) a_{t-1}, \alpha A_{t-1}, 4\alpha(n/b - 1)^2 C_{t-1} \right\}. \tag{28}$$

Because we will search for $\alpha$ down to an absolute constant, we will often swap $\left(1 + \frac{\alpha}{4}\right)$ for $(1 + \alpha)$ for readability. We assume the right-hand side of (28) holds in **Step 1** and **Step 2**. We then select $\alpha$ to satisfy this condition (and all others) in **Step 3** below. The proof occurs in five steps total.

1. Lower bound the dual suboptimality $a_t \mathcal{L}(w_\star, q_t)$.

2. Upper bound the primal suboptimality $a_t \mathcal{L}(w_t, q_\star)$, and combine both to derive a bound on the gap function for $t \geq 2$.

3. Derive all conditions on the learning rate constant $\alpha$ and batch size $b$.

4. Bound $\gamma_1$ and sum $\gamma_t$ over $t$ for a $T$-step progress bound.

5. Bound the remaining non-telescoping terms to complete the analysis.

We begin with a section of technical lemmas that will not only be useful in various areas of the proof but also capture the main insights that allow the method to achieve the given rate. Given these lemmas, the main proof occurs in Appx. C.3 and otherwise follows standard structure.

## C.2 Technical Lemmas

This section contains a number of lemmas that describe common structures in the analysis of quantities in the primal and dual. Lem. 4 bounds cross terms that arise when there are inner products between the primal-dual iterates and their gradient estimates. Lem. 5 and Lem. 6 are respectively the primal and dual noise bounds, constructed to control the variation of the terms $\delta_t^{\mathrm{P}}$ and $\delta_t^{\mathrm{D}}$ appearing in (21). Finally, Lem. 10 exploits the cyclic style updates of the $\hat{q}_t$ table to bound the term $\|\hat{q}_{t-1} - q_{t-2}\|_2^2$ which is used in the primal noise bound.

**Cross Term Bound** Both of the estimates of the gradient of the coupled term $q^\top \ell(w)$ with respect to the primal and dual variables share a similar structure (see (22) and compare to (29)). They are designed to achieve a particular form of telescoping, with a remaining squared term that can be controlled by case-specific techniques. This can be observed within Lem. 4. In the sequel, we will refer to a sequence of random vectors $(u_t)_{t \geq 1}$ as *adapted to $\mathcal{H}_t$*, where $\mathcal{H}_t = \{(I_s, J_s)\}_{s=1}^t$ is the history of random blocks. This simply means that $u_t$, when conditioned on $\mathcal{H}_{t-1}$, is only a function of the current random block $(I_t, J_t)$. Similarly, conditioned on $\mathcal{H}_{t-1}$, we have that $u_{t-1}$ is not random. In the language of probability theory, $\{\sigma(\mathcal{H}_t)\}_{t \geq 1}$ forms a filtration and $u_t$ is $\sigma(\mathcal{H}_t)$-measurable, but this terminology is not necessary for understanding the results.

**Lemma 4** (Cross Term Bound). *Let $(x_t)_{t \geq 1}$, $(y_t)_{t \geq 1}$, and $(\hat{y}_t)_{t \geq 1}$ denote random sequences of $\mathbb{R}^m$-valued vectors, and let $x_\star \in \mathbb{R}^m$ be a vector. Denote by $I_t$ be the index of a uniform random block $[M]$.*

$$
v_t := \hat{y}_{t-1} + \frac{na_{t-1}}{a_t} \frac{1}{b} \sum_{i \in B_{I_t}} (y_{t-1,i} - \hat{y}_{t-2,i}) \tag{29}
$$

*Finally, let $(x_t, y_t, \hat{y}_t)_{t \geq 1}$ adapted to $\mathcal{H}_t$ (as defined above). Then, for any positive constant $\gamma > 0$,*

$$
a_t \mathbb{E}_t [\langle y_t - v_t, x_\star - x_t \rangle] \leq a_t \mathbb{E}_t [\langle y_t - \hat{y}_{t-1}, x_\star - x_t \rangle] - a_{t-1} \langle y_{t-1} - \hat{y}_{t-2}, x_\star - x_{t-1} \rangle
$$
$$
+ \frac{n^2 a_{t-1}^2}{2\gamma} \mathbb{E}_t \left\| \tfrac{1}{b} \sum_{i \in B_{I_t}} (y_{t-1,i} - \hat{y}_{t-2,i}) \right\|_2^2 + \frac{\gamma}{2} \mathbb{E}_t \|x_t - x_{t-1}\|_2^2 .
$$

*Proof.* By plugging in the value of $v_t$ and using $x_\star - x_t = x_\star - x_{t-1} + x_{t-1} - x_t$, we have that

$$
a_t \mathbb{E}_t [\langle y_t - v_t, x_\star - x_t \rangle] = a_t \mathbb{E}_t [\langle y_t - \hat{y}_{t-1}, x_\star - x_t \rangle] - na_{t-1} \mathbb{E}_t \left[ \left\langle \frac{1}{b} \sum_{i \in B_{I_t}} (y_{t-1} - \hat{y}_{t-2}), x_\star - x_t \right\rangle \right]
$$
$$
= a_t \mathbb{E}_t [\langle y_t - \hat{y}_{t-1}, x_\star - x_t \rangle] - a_{t-1} \langle y_{t-1} - \hat{y}_{t-2}, x_\star - x_{t-1} \rangle
$$
$$
+ na_{t-1} \mathbb{E}_t \left[ \left\langle \frac{1}{b} \sum_{i \in B_{I_t}} (y_{t-1} - \hat{y}_{t-2}), x_t - x_{t-1} \right\rangle \right]
$$
$$
\leq a_t \mathbb{E}_t [\langle y_t - \hat{y}_{t-1}, x_\star - x_t \rangle] - a_{t-1} \langle y_{t-1} - \hat{y}_{t-2}, x_\star - x_{t-1} \rangle
$$
$$
+ \frac{n^2 a_{t-1}^2}{2\gamma} \mathbb{E}_t \left\| \tfrac{1}{b} \sum_{i \in B_{I_t}} (y_{t-1,i} - \hat{y}_{t-2,i}) \right\|_2^2 + \frac{\gamma}{2} \mathbb{E}_t \|x_t - x_{t-1}\|_2^2 ,
$$

where the final step follows from Young's inequality with parameter $\gamma$. $\qquad\square$

In the primal case, we have that $v_t = v_t^{\mathrm{P}}$, $y_t = \nabla \ell(w_t)^\top q_t$, $\hat{y}_t = \hat{g}_t^\top \hat{q}_t$, and $x_t = w_t$. In the dual case, we have that $v_t = v_t^{\mathrm{D}}$, $y_t = \ell(w_t)$, $\hat{y}_t = \hat{\ell}_{t+1}$, and $x_t = q_t$. The next few lemmas control the third term appearing in Lem. 4 for the specific case of the primal and dual sequences.

**Noise Term Bounds** Next, we proceed to control the $\delta_t^{\mathrm{P}}$ and $\delta_t^{\mathrm{D}}$ by way of Lem. 5 and Lem. 6. As discussed in Sec. 3, a key step in the convergence proof is establishing control over these terms. Define $\pi(t, i)$ to satisfy $\hat{q}_{t,i} = q_{\pi(t,i),i}$ and $\hat{g}_{t,i} = \nabla \ell_i(w_{\pi(t,i)})$, that is, the time index of the last update of table element $i$ on or before time $t$. This notation is used to write the table values such as $\hat{q}_t$ in terms of past values of the iterates (e.g., $q_t$).

**Lemma 5** (Primal Noise Bound). *When $t \geq 2$, we have that*

$$\mathbb{E}_t \left\| \delta_t^{\mathrm{P}} \right\|_2^2 \leq \frac{3 q_{\max}}{n} \sum_{i=1}^n q_{t-1,i} \left\| \nabla \ell_i(w_{t-1}) - \nabla \ell_i(w^\star) \right\|_2^2$$

$$+ \frac{3 q_{\max}}{n} \sum_{i=1}^n q_{\pi(t-2,i),i} \left\| \nabla \ell_i(w_{\pi(t-2,i)}) - \nabla \ell_i(w^\star) \right\|_2^2$$

$$+ \frac{3 G^2}{n} \left\| q_{t-1} - \hat{q}_{t-2} \right\|_2^2 \tag{30}$$

*Proof.* By definition, we have that

$$\mathbb{E}_t \left\| \delta_t^{\mathrm{P}} \right\|_2^2 = \mathbb{E}_t \left[ \left\| \frac{1}{b} \sum_{i \in B_{I_t}} \nabla \ell_i(w_{t-1}) q_{t-1,i} - \hat{g}_{t-2,i} \hat{q}_{t-2,i} \right\|_2^2 \right]$$

$$= \frac{1}{b^2} \mathbb{E}_t \left[ \left\| \sum_{i \in B_{I_t}} \nabla \ell_i(w_{t-1}) q_{t-1,i} - \hat{g}_{t-2,i} \hat{q}_{t-2,i} \right\|_2^2 \right]$$

$$\leq \frac{1}{b} \mathbb{E}_t \left[ \sum_{i \in B_{I_t}} \left\| \nabla \ell_i(w_{t-1}) q_{t-1,i} - \hat{g}_{t-2,i} \hat{q}_{t-2,i} \right\|_2^2 \right]$$

$$= \frac{1}{n} \sum_{i=1}^n \left\| \nabla \ell_i(w_{t-1}) q_{t-1,i} - \hat{g}_{t-2,i} \hat{q}_{t-2,i} \right\|_2^2 ,$$

where we use that $I_t$ is drawn uniformly over $n/b$. Continuing again with the term above, we have

$$\frac{1}{n} \sum_{i=1}^n \left\| \nabla \ell_i(w_{t-1}) q_{t-1,i} - \hat{g}_{t-2,i} \hat{q}_{t-2,i} \right\|_2^2$$

$$= \frac{1}{n} \sum_{i=1}^n \left\| (\nabla \ell_i(w_{t-1}) - \nabla \ell_i(w_\star)) q_{t-1,i} - (\hat{g}_{t-2,i} - \nabla \ell_i(w_\star)) \hat{q}_{t-2,i} + (q_{t-1,i} - \hat{q}_{t-2,i}) \nabla \ell_i(w_\star) \right\|_2^2$$

$$\leq \frac{3}{n} \sum_{i=1}^n \left( q_{t-1,i}^2 \left\| \nabla \ell_i(w_{t-1}) - \nabla \ell_i(w_\star) \right\|_2^2 + \hat{q}_{t-2,i}^2 \left\| \hat{g}_{t-2,i} - \nabla \ell_i(w_\star) \right\|_2^2 \right.$$

$$+ \left. (q_{t-1,i} - \hat{q}_{t-2,i})^2 \left\| \nabla \ell_i(w_\star) \right\|_2^2 \right)$$

$$\leq \frac{3}{n} \sum_{i=1}^n \left( q_{\max} q_{t-1,i} \left\| \nabla \ell_i(w_{t-1}) - \nabla \ell_i(w_\star) \right\|_2^2 + q_{\max} \hat{q}_{t-2,i} \left\| \hat{g}_{t-2,i} - \nabla \ell_i(w_\star) \right\|_2^2 \right.$$

$$+ \left. (q_{t-1,i} - \hat{q}_{t-2,i})^2 G^2 \right).$$

where we use that $\|\nabla \ell_i(w_\star)\|_2^2 \leq G^2$ because every $\ell_i$ is $G$-Lipschitz with respect to $\|\cdot\|_2^2$, and that $q_i \leq q_{\max} = \max_{q \in \mathcal{Q}} \|q\|_\infty$. This completes the proof. $\square$

The corresponding dual noise bound in Lem. 6 follows similarly, using cyclic updates in the loss table.

**Lemma 6** (Dual Noise Bound). *For $t \geq 2$,*

$$\mathbb{E}_t \left\| \delta_t^{\mathrm{D}} \right\|_2^2 \leq \frac{G^2}{n} \sum_{\tau=t-n/b}^{t-2} \left\| w_{t-1} - w_{\tau \vee 0} \right\|_2^2 .$$

*Proof.* Then, we may exploit the coordinate structure of the noise term to write

$$
\mathbb{E}_t \left\| \delta_t^{\mathrm{D}} \right\|_2^2 = \mathbb{E}_t \left\| \frac{1}{b} \sum_{j \in B_{J_t}} (\ell_j(w_{t-1}) - \hat{\ell}_{t-1,j}) e_j \right\|_2^2
$$

$$
= \frac{1}{b^2} \left( \mathbb{E}_t \sum_{j \in B_{J_t}} \left\| (\ell_j(w_{t-1}) - \hat{\ell}_{t-1,j}) e_j \right\|_2^2 \right.
$$

$$
\left. + \mathbb{E}_t \left[ \sum_{j \neq k} (\ell_j(w_{t-1}) - \hat{\ell}_{t-1,j})(\ell_k(w_{t-1}) - \hat{\ell}_{t-1,k}) \textcolor{red}{\langle e_j, e_k \rangle} \right] \right)
$$

$$
= \frac{1}{b^2} \mathbb{E}_t \sum_{j \in B_{J_t}} \left\| (\ell_j(w_{t-1}) - \hat{\ell}_{t-1,j}) e_j \right\|_2^2
$$

$$
= \frac{1}{b^2} \mathbb{E}_t \sum_{j \in B_{J_t}} |\ell_j(w_{t-1}) - \hat{\ell}_{t-1,j}|^2
$$

$$
\leq \frac{1}{bn} \sum_{i=1}^{n} |\ell_i(w_{t-1}) - \hat{\ell}_{t-1,i}|^2
$$

$$
\leq \frac{G^2}{n} \sum_{\tau=t-n/b}^{t-2} \| w_{t-1} - w_{\tau \vee 0} \|_2^2 .
$$

The red term is zero because $j \neq k$. The sum in the last line has $n/b - 1$ terms because our order of updates forces one of the blocks of the $\hat{\ell}_{t-1}$ vector to have values equal to $w_{t-1}$ *before* defining $\delta_t^{\mathrm{D}}$. $\qquad\square$

**Controlling the Recency of the Loss Table** In this section, we bound the $\| q_{t-1} - \hat{q}_{t-2} \|_2^2$ term appearing in the primal noise bound Lem. 5. Controlling this term is essential to achieving the correct rate, as we comment toward the end of this section.

Recall that the $[n]$ indices are partitioned into blocks $(B_1, \ldots, B_M)$ for $M := n/b$, where $b$ is assumed to divide $n$. For any $t \geq 1$, we first decompose

$$
\sum_{i=1}^{n} (q_{t,i} - \hat{q}_{t-1,i})^2 = \sum_{K=1}^{M} \sum_{i \in B_K} (q_{t,i} - \hat{q}_{t-1,i})^2,
$$

and analyze block-by-block. Our goal is to be able to count this quantity in terms of $\| q_t - q_{t-1} \|_2^2$ terms. The main result is given in Lem. 10, which is built up in the following lemmas. Consider a block index $K \in [M]$. Define the number $t_K = M \lfloor (t-K)/M \rfloor + K$ when $t - 1 \geq K$ and and $t_K = 0$ otherwise.

**Lemma 7.** *It holds that*

$$
\sum_{i \in B_K} (q_{t,i} - \hat{q}_{t-1,i})^2 \leq \sum_{i \in B_K} (t - t_K) \sum_{s=t_K+1}^{t} (q_{s,i} - q_{s-1,i})^2 .
$$

*Proof.* We define $t_K$ to be the earliest time index $\tau$ on or before $t - 1$ when block $K$ of $q_\tau$ was used to update $\hat{q}_\tau$. When $t - 1 < K$, then $t_K = 0$. When $t - 1 \geq K$, we can compute this number

$t_K = M \lfloor [(t-1) - (K-1)]/M \rfloor + K = M \lfloor (t-K)/M \rfloor + K$. Then, write

$$\sum_{i \in B_K} (q_{t,i} - \hat{q}_{t-1,i})^2 = \sum_{i \in B_K} (q_{t,i} - q_{t_K,i})^2$$

$$= \sum_{i \in B_K} \left( \sum_{s=t_K+1}^{t} q_{s,i} - q_{s-1,i} \right)^2$$

$$\leq \sum_{i \in B_K} (t - t_K) \sum_{s=t_K+1}^{t} (q_{s,i} - q_{s-1,i})^2,$$

where the last line follows by Young's inequality. $\qquad \square$

While we will not be able to cancel these terms on every iterate, we will be able to when aggregating over time and then redistributing them. Recall $(a_t)_{t \geq 1}$ as described in Appx. C.1. Indeed, by summing across iterations, we see that

$$\sum_{t=1}^{T} a_t \sum_{i=1}^{n} (q_{t,i} - \hat{q}_{t-1,i})^2 \leq \sum_{t=1}^{T} a_t \sum_{K=1}^{M} \sum_{i \in B_K} (t - t_K) \sum_{s=t_K+1}^{t} (q_{s,i} - q_{s-1,i})^2.$$

We can start by swapping the first two sums and only considering the values of $t$ that are greater than or equal to $K$.

$$\sum_{t=1}^{T} a_t \sum_{K=1}^{M} \sum_{i \in B_K} (t - t_K) \sum_{s=t_K+1}^{t} (q_{s,i} - q_{s-1,i})^2$$

$$= \sum_{K=1}^{M} \sum_{t=1}^{K-1} a_t \sum_{i \in B_K} (t - t_K) \sum_{s=t_K+1}^{t} (q_{s,i} - q_{s-1,i})^2 + \sum_{K=1}^{M} \sum_{t=K}^{T} a_t \sum_{i \in B_K} (t - t_K) \sum_{s=t_K+1}^{t} (q_{s,i} - q_{s-1,i})^2$$

$$= \underbrace{\sum_{K=1}^{M} \sum_{t=1}^{K-1} a_t \sum_{i \in B_K} t \sum_{s=1}^{t} (q_{s,i} - q_{s-1,i})^2}_{S_0} + \underbrace{\sum_{K=1}^{M} \sum_{t=K}^{T} a_t \sum_{i \in B_K} (t - t_K) \sum_{s=t_K+1}^{t} (q_{s,i} - q_{s-1,i})^2}_{S_1},$$

$$(31)$$

where we use in the last line that $t_K = 0$ when $t < K$. We handle the terms $S_0$ and $S_1$ separately. In either case, we have to match the sums over $K$ and over $i$ in order to create complete vectors, as opposed to differences between coordinates. We also maintain the update rules of the sequence $(a_t)_{t \geq 1}$ that will be used in the proof.

**Lemma 8.** *Assume that $\alpha \leq \frac{1}{M}$ and $a_t \leq (1 + \alpha)a_{t-1}$. It holds that $S_0$ as defined in (31) satisfies*

$$S_0 \leq \frac{eM(M-1)}{2} \sum_{t=1}^{M-1} a_t \|q_t - q_{t-1}\|_2^2.$$

*Proof.* Write

$$S_0 = \sum_{K=1}^{M} \sum_{t=1}^{K-1} a_t \sum_{i \in B_K} t \sum_{s=1}^{t} (q_{s,i} - q_{s-1,i})^2 \qquad \text{by definition}$$

$$= \sum_{t=1}^{M-1} t a_t \sum_{K=t+1}^{M} \sum_{i \in B_K} \sum_{s=1}^{t} (q_{s,i} - q_{s-1,i})^2 \qquad \text{swap sums over } K \text{ and } t$$

$$= \sum_{t=1}^{M-1} t a_t \sum_{s=1}^{t} \sum_{K=t+1}^{M} \sum_{i \in B_K} (q_{s,i} - q_{s-1,i})^2 \qquad \text{move sum over } s$$

$$\leq \sum_{t=1}^{M-1} t a_t \sum_{s=1}^{t} \sum_{K=1}^{M} \sum_{i \in B_K} (q_{s,i} - q_{s-1,i})^2 \qquad \sum_{K=t+1}^{M} (\cdot) \leq \sum_{K=1}^{M} (\cdot)$$

$$= \sum_{t=1}^{M-1} t a_t \sum_{s=1}^{t} \|q_s - q_{s-1}\|_2^2 \qquad \sum_{K=1}^{M} \sum_{i \in B_K} (\cdot) = \sum_{i=1}^{n} (\cdot)$$

$$= \sum_{s=1}^{M-1} \left( \sum_{t=s}^{M-1} t a_t \right) \|q_s - q_{s-1}\|_2^2 \qquad \text{swap sums over } s \text{ and } t$$

$$\leq \sum_{s=1}^{M-1} \left( \sum_{t=s}^{M-1} t (1+\alpha)^{t-s} a_s \right) \|q_s - q_{s-1}\|_2^2 \qquad a_t \leq (1+\alpha) a_{t-1}$$

$$\leq \sum_{s=1}^{M-1} a_s (1+\alpha)^M \left( \sum_{t=s}^{M-1} t \right) \|q_s - q_{s-1}\|_2^2 \qquad t - s \leq M$$

$$\leq \sum_{s=1}^{M-1} e a_s \left( \sum_{t=s}^{M-1} t \right) \|q_s - q_{s-1}\|_2^2 \qquad \alpha \leq \frac{1}{M} \implies (1+\alpha)^M \leq e$$

$$\leq \sum_{s=1}^{M-1} e a_s \left( \sum_{t=1}^{M-1} t \right) \|q_s - q_{s-1}\|_2^2 \qquad \sum_{t=s}^{M-1} (\cdot) \leq \sum_{t=1}^{M-1} (\cdot)$$

$$= \frac{e M (M-1)}{2} \sum_{s=1}^{M-1} a_s \|q_s - q_{s-1}\|_2^2,$$

the result as desired. $\qquad \square$

Thus, $S_0$ contributes about $M^3$ of such terms over the entire trajectory, which can be viewed as an initialization cost. Next, we control $S_1$.

**Lemma 9.** *Assume that $\alpha \leq \frac{1}{M}$ and $a_t \leq (1+\alpha) a_{t-1}$. It holds that $S_1$ as defined in (31) satisfies*

$$S_1 \leq 2e^2 M^2 \sum_{t=1}^{T} a_t \|q_t - q_{t-1}\|_2^2.$$

*Proof.* We can reparametrize $t$ in terms of $r = t - K$, which will help in reasoning with $t_K$. Define $r_K = M \lfloor r/M \rfloor + K$, and let

$$S_1 = \sum_{K=1}^{M} \sum_{r=0}^{T-K} a_{r+K} \sum_{i \in B_K} (r + K - r_K) \sum_{s=r_K+1}^{r+K} (q_{s,i} - q_{s-1,i})^2 \quad \text{by definition and } r = t - K$$

$$\leq M \sum_{K=1}^{M} \sum_{r=0}^{T-K} a_{r+K} \sum_{i \in B_K} \sum_{s=r_K+1}^{r+K} (q_{s,i} - q_{s-1,i})^2 \qquad (r - M\lfloor r/M \rfloor) \leq M$$

$$\leq M \sum_{K=1}^{M} \sum_{r=0}^{T-K} (1+\alpha)^K a_r \sum_{i \in B_K} \sum_{s=r_K+1}^{r+K} (q_{s,i} - q_{s-1,i})^2 \qquad a_t \leq (1+\alpha) a_{t-1}$$

$$\leq eM \sum_{K=1}^{M} \sum_{r=0}^{T-K} a_r \sum_{i \in B_K} \sum_{s=r_K+1}^{r+K} (q_{s,i} - q_{s-1,i})^2 \qquad K \leq M \text{ and } \alpha \leq \frac{1}{M}$$

$$= eM \sum_{r=0}^{T-2} a_r \sum_{K=1}^{\min\{T-r,M\}} \sum_{i \in B_K} \sum_{s=r_K+1}^{r+K} (q_{s,i} - q_{s-1,i})^2 \qquad \text{swap sums over } r \text{ and } K$$

$$\leq eM \sum_{r=0}^{T-2} a_r \sum_{K=1}^{\min\{T-r,M\}} \sum_{i \in B_K} \sum_{s=M\lfloor r/M \rfloor+1}^{\min\{r+M,T\}} (q_{s,i} - q_{s-1,i})^2 \qquad \sum_{s=r_K+1}^{r+K} (\cdot) \leq \sum_{s=M\lfloor r/M \rfloor+1}^{\min\{r+M,T\}} (\cdot)$$

$$= eM \sum_{r=0}^{T-2} a_r \sum_{s=M\lfloor r/M \rfloor+1}^{\min\{r+M,T\}} \sum_{K=1}^{\min\{T-r,M\}} \sum_{i \in B_K} (q_{s,i} - q_{s-1,i})^2 \qquad \text{move sum over } s$$

$$\leq eM \sum_{r=0}^{T-2} a_r \sum_{s=M\lfloor r/M \rfloor+1}^{\min\{r+M,T\}} \sum_{K=1}^{M} \sum_{i \in B_K} (q_{s,i} - q_{s-1,i})^2 \qquad \sum_{K=1}^{\min\{T-r,M\}} (\cdot) \leq \sum_{K=1}^{M} (\cdot)$$

$$= eM \sum_{r=0}^{T-2} a_r \sum_{s=M\lfloor r/M \rfloor+1}^{\min\{r+M,T\}} \|q_s - q_{s-1}\|_2^2 \qquad \sum_{K=1}^{M} \sum_{i \in B_K} (\cdot) = \sum_{i=1}^{n} (\cdot)$$

$$\leq eM \sum_{r=0}^{T-2} \sum_{s=M\lfloor r/M \rfloor}^{\min\{r+M,T\}} (1+\alpha)^{r-s} a_s \|q_s - q_{s-1}\|_2^2 \qquad a_t \leq (1+\alpha) a_{t-1}$$

$$\leq e^2 M \sum_{r=0}^{T-2} \sum_{s=M\lfloor r/M \rfloor}^{\min\{r+M,T\}} a_s \|q_s - q_{s-1}\|_2^2 \qquad r - s \leq M \text{ and } \alpha \leq \frac{1}{M}$$

$$\leq 2e^2 M^2 \sum_{s=1}^{T-1} a_s \|q_s - q_{s-1}\|_2^2 .$$

The last line follows because the inner sum $\sum_{s=M\lfloor r/M \rfloor}^{\min\{r+M,T\}} a_s \|q_s - q_{s-1}\|_2^2$ can be thought of as a sliding window of size at most $2M$ centered at $r$, which essentially repeats each element in the sum $2M$ times at most. $\qquad \square$

In either case, the contribution over the entire trajectory is order $M^2 = (n/b)^2$ terms. Combining the bounds for $S_0$ and $S_1$ yields the following.

**Lemma 10.** *Letting $M = n/b$ be the number of blocks. Assume that $\alpha \leq \frac{1}{M}$ and $a_t \leq (1+\alpha) a_{t-1}$. We have that*

$$\sum_{t=1}^{T} a_t \sum_{i=1}^{n} (q_{t,i} - \hat{q}_{t-1,i})^2 \leq 3e^2 M^2 \sum_{t=1}^{T} a_t \|q_t - q_{t-1}\|_2^2 .$$

*Proof.* Use Lem. 7 to achieve (31) and apply Lem. 8 and Lem. 9 on each term to get

$$\sum_{t=1}^{T} a_t \sum_{i=1}^{n} (q_{t,i} - \hat{q}_{t-1,i})^2 \leq \frac{eM(M-1)}{2} \sum_{t=1}^{M-1} a_t \|q_t - q_{t-1}\|_2^2 + 2e^2 M^2 \sum_{t=1}^{T} a_t \|q_t - q_{t-1}\|_2^2$$

$$\leq 3e^2 M^2 \sum_{t=1}^{T} a_t \|q_t - q_{t-1}\|_2^2,$$

completing the proof. □

We close this subsection with comments on how Lem. 10 can be used. The term $\mathbb{E}_t \left\| \delta_t^{\mathrm{D}} \right\|_2^2$ is multiplied in (36) by a factor $\frac{na_{t-1}^2 G^2}{\mu C_{t-1}}$. Thus, if we apply Lem. 10 when redistributing over time a term $\|q_{t-1} - q_{t-2}\|_2^2$ which will be multiplied by a factor (ignoring absolute constants) of

$$\frac{na_{t-1}^2 G^2}{\mu C_{t-1}} \cdot M^2 = \frac{n^3 a_{t-1}^2 G^2}{b^2 \mu C_{t-1}}.$$

In order to cancel such a term, we require the use of $-\frac{A_{t-1}\nu}{2} \|q_t - q_{t-1}\|_2^2$. We can reserve half to be used up by (49), and be left with the condition

$$\frac{n^3 a_{t-1}^2 G^2}{b^2 \mu C_{t-1}} \leq A_{t-1}\nu \iff \alpha^2 \leq \frac{b^2}{n^2} \frac{\mu\nu}{2nG^2},$$

as we have applied that $a_{t-1} \leq 2\alpha C_{t-1}$ and $a_{t-1} \leq \alpha A_{t-1}$. Thus, this introduces a dependence of order $\frac{n}{b} \sqrt{\frac{nG^2}{\mu\nu}}$ on $\alpha$, which propagates to the learning rate $\alpha$. We now proceed to the main logic of the convergence analysis.

## C.3 Proof of Main Result

### C.3.1 Lower Bound on Dual Objective

We first quantify the gap between $\mathcal{L}(w_\star, q_t)$ and $\mathcal{L}(w_\star, q_\star)$ by providing a lower bound in expectation on $\mathcal{L}(w_\star, q_t)$, given in Lem. 11. As in Lem. 5, recall the notation $\pi(t, i)$ to satisfy $\hat{q}_{t,i} = q_{\pi(t,i),i}$ and $g_{t,i} = \nabla\ell_i(w_{\pi(t,i)})$, that is, the time index of the last update of table element $i$ on or before time $t$, with $\pi(t, i) = 0$ for $t \leq 0$.

**Lemma 11.** *Assume that $\alpha \leq \mu/(24eL\kappa_{\mathcal{Q}})$. Then, for $t \geq 2$, we have that:*

$$- \mathbb{E}_t[a_t \mathcal{L}(w_\star, q_t)]$$

$$\leq -a_t \mathbb{E}_t[q_t^\top \ell(w_t)] - \frac{a_t}{2L} \sum_{i=1}^{n} q_{t,i} \mathbb{E}_t \|\nabla\ell_i(w_t) - \nabla\ell_i(w_\star)\|_2^2$$

$$+ \frac{a_{t-1}}{4L} \sum_{i=1}^{n} q_{t-1,i} \|\nabla\ell_i(w_{t-1}) - \nabla\ell_i(w_\star)\|_2^2 + \sum_{i=1}^{n} \frac{a_{\pi(t-2,i)}}{4L} q_{\pi(t-2,i),i} \left\| \nabla\ell_i(w_{\pi(t-2,i)}) - \nabla\ell_i(w_\star) \right\|_2^2$$

$$+ \frac{6n\alpha a_{t-1} G^2}{\mu} \|q_{t-1} - \hat{q}_{t-2}\|_2^2$$

$$- a_t \mathbb{E}_t \left[ \left\langle \nabla\ell(w_t)^\top q_t - \hat{g}_{t-1}^\top \hat{q}_{t-1}, w_\star - w_t \right\rangle \right] + a_{t-1} \left\langle \nabla\ell(w_{t-1})^\top q_{t-1} - \hat{g}_{t-2}^\top \hat{q}_{t-2}, w_\star - w_{t-1} \right\rangle$$

$$- \frac{a_t\mu}{2} \mathbb{E}_t \|w_t\|_2^2 + a_t\nu \mathbb{E}_t[D(q_t \| q_0)] - \frac{\mu A_t}{2} \mathbb{E}_t \|w_\star - w_t\|_2^2$$

$$- \frac{\mu C_{t-1}}{4} \|w_t - w_{t-1}\|_2^2 + \frac{\mu C_{t-1}}{2} \|w_\star - w_{t-1}\|_2^2$$

$$- \frac{c_{t-1}\mu}{2} \sum_{\tau=t-n/b}^{t-2} \mathbb{E}_t \|w_t - w_{\tau\vee 0}\|_2^2 + \frac{c_{t-1}\mu}{2} \sum_{\tau=t-n/b}^{t-2} \|w_\star - w_{\tau\vee 0}\|_2^2 .$$

*Proof.* We use convexity and smoothness (1), then add and subtract (2) elements from the primal update, and finally use the definition of the proximal operator (3) with the optimality of $w_t$ for the problem that defines it.

$$a_t \mathcal{L}(w_\star, q_t) := a_t q_t^\top \ell(w_\star) + a_t \frac{\mu}{2} \|w_\star\|_2^2 - a_t \nu D(q_t \| q_0)$$

$$\stackrel{(1)}{\geq} a_t q_t^\top \ell(w_t) + a_t \langle \nabla \ell(w_t)^\top q_t, w_\star - w_t \rangle + \frac{a_t}{2L} \sum_{i=1}^n q_{t,i} \|\nabla \ell_i(w_t) - \nabla \ell_i(w_\star)\|_2^2$$

$$+ \frac{a_t \mu}{2} \|w_\star\|_2^2 - \nu D(q_t \| q_0)$$

$$\stackrel{(2)}{=} a_t q_t^\top \ell(w_t) + a_t \langle \nabla \ell(w_t)^\top q_t - v_t^{\mathrm{P}}, w_\star - w_t \rangle + a_t \langle v_t^{\mathrm{P}}, w_\star - w_t \rangle$$

$$+ \frac{a_t \mu}{2} \|w_\star\|_2^2 - a_t \nu D(q_t \| q_0) + \frac{a_t}{2L} \sum_{i=1}^n q_{t,i} \|\nabla \ell_i(w_t) - \nabla \ell_i(w_\star)\|_2^2$$

$$+ \frac{\mu C_{t-1}}{2} \|w_\star - w_{t-1}\|_2^2 - \frac{\mu C_{t-1}}{2} \|w_\star - w_{t-1}\|_2^2 \tag{32}$$

$$+ \frac{c_{t-1} \mu}{2} \sum_{\tau = t-n/b}^{t-2} \|w_\star - w_{\tau \vee 0}\|_2^2 - \frac{c_{t-1} \mu}{2} \sum_{\tau = t-n/b}^{t-2} \|w_\star - w_{\tau \vee 0}\|_2^2$$

$$\stackrel{(3)}{\geq} a_t q_t^\top \ell(w_t) + a_t \underbrace{\langle \nabla \ell(w_t)^\top q_t - v_t^{\mathrm{P}}, w_\star - w_t \rangle}_{\text{cross term}} + a_t \underbrace{\langle v_t^{\mathrm{P}}, w_t - w_t \rangle}_{=0}$$

$$+ \frac{a_t \mu}{2} \|w_t\|_2^2 - a_t \nu D(q_t \| q_0) + \frac{a_t}{2L} \sum_{i=1}^n q_{t,i} \|\nabla \ell_i(w_t) - \nabla \ell_i(w_\star)\|_2^2$$

$$+ \frac{\mu C_{t-1}}{2} \|w_t - w_{t-1}\|_2^2 - \frac{\mu C_{t-1}}{2} \|w_\star - w_{t-1}\|_2^2 \tag{33}$$

$$+ \frac{c_{t-1} \mu}{2} \sum_{\tau = t-n/b}^{t-2} \|w_t - w_{\tau \vee 0}\|_2^2 - \frac{c_{t-1} \mu}{2} \sum_{\tau = t-n/b}^{t-2} \|w_\star - w_{\tau \vee 0}\|_2^2$$

$$+ \frac{\mu A_t}{2} \|w_\star - w_t\|_2^2. \tag{34}$$

Next, we are able to use Lem. 4 with $v_t = v_t^{\mathrm{P}}$, $y_t = \nabla \ell(w_t)^\top q_t$, $\hat{y}_t = \hat{g}_t^\top \hat{q}_t$, $x_t = w_t$, $x_\star = w_\star$, and $\gamma = \mu C_{t-1}/2$ which yields that

$$a_t \mathbb{E}_t \left[ \langle \nabla \ell(w_t)^\top q_t - v_t^{\mathrm{P}}, w_\star - w_t \rangle \right]$$

$$\leq a_t \mathbb{E}_t \left[ \langle \nabla \ell(w_t)^\top q_t - \hat{g}_{t-1}^\top \hat{q}_{t-1}, w_\star - w_t \rangle \right] - a_{t-1} \langle \nabla \ell(w_{t-1})^\top q_{t-1} - \hat{g}_{t-2}^\top \hat{q}_{t-2}, w_\star - w_{t-1} \rangle$$

$$+ \frac{n^2 a_{t-1}^2}{\mu C_{t-1}} \underbrace{\mathbb{E}_t \left[ \left\| \frac{1}{b} \sum_{i \in I_t} \nabla \ell_i(w_{t-1}) q_{t-1,i} - g_{t-2,i} \hat{q}_{t-2,i} \right\|_2^2 \right]}_{\mathbb{E}_t \| \delta_t^{\mathrm{P}} \|_2^2} + \frac{\mu C_{t-1}}{4} \mathbb{E}_t \|w_t - w_{t-1}\|_2^2. \tag{35}$$

Then, apply Lem. 5 to achieve

$$\mathbb{E}_t \left\| \delta_t^{\mathrm{P}} \right\|_2^2 \leq \frac{3 q_{\max}}{n} \sum_{i=1}^n q_{t-1,i} \|\nabla \ell_i(w_{t-1}) - \nabla \ell_i(w^\star)\|_2^2$$

$$+ \frac{3 q_{\max}}{n} \sum_{i=1}^n q_{\pi(t-2,i),i} \left\| \nabla \ell_i(w_{\pi(t-2,i)}) - \nabla \ell_i(w^\star) \right\|_2^2$$

$$+ \frac{3 G^2}{n} \|q_{t-1} - \hat{q}_{t-2}\|_2^2 \tag{36}$$

In the following, the blue terms indicate what changes from line to line. Combine the previous two steps to collect all terms for the lower bound. That is, apply (34) to write

$$\mathbb{E}_t[a_t \mathcal{L}(w_\star, q_t)]$$

$$:= a_t q_t^\top \ell(w_\star) + a_t \frac{\mu}{2} \|w_\star\|_2^2 - a_t \nu D(q_t \| q_0)$$

$$\geq a_t q_t^\top \ell(w_t) + \mathbb{E}_t[a_t \langle \nabla \ell(w_t)^\top q_t - v_t^{\mathrm{P}}, w_\star - w_t \rangle] + \frac{a_t}{2L} \sum_{i=1}^n q_{t,i} \|\nabla \ell_i(w_t) - \nabla \ell_i(w_\star)\|_2^2$$

$$+ \frac{a_t \mu}{2} \|w_t\|_2^2 - a_t \nu D(q_t \| q_0) + \frac{\mu A_t}{2} \|w_\star - w_t\|_2^2$$

$$+ \frac{\mu C_{t-1}}{2} \|w_t - w_{t-1}\|_2^2 - \frac{\mu C_{t-1}}{2} \|w_\star - w_{t-1}\|_2^2$$

$$+ \frac{c_{t-1} \mu}{2} \sum_{\tau=t-n/b}^{t-2} \|w_t - w_{\tau \vee 0}\|_2^2 - \frac{c_{t-1} \mu}{2} \sum_{\tau=t-n/b}^{t-2} \|w_\star - w_{\tau \vee 0}\|_2^2,$$

then apply (35) to the blue term above to write

$$\mathbb{E}_t[a_t \mathcal{L}(w_\star, q_t)]$$

$$\geq a_t q_t^\top \ell(w_t) + \frac{a_t}{2L} \sum_{i=1}^n q_{t,i} \|\nabla \ell_i(w_t) - \nabla \ell_i(w_\star)\|_2^2$$

$$- \frac{n^2 a_{t-1}^2}{\mu C_{t-1}} \mathbb{E}_t \left\| \nabla \ell_{i_{t-1}}(w_{t-1}) q_{t-1, i_{t-1}} - g_{t-2, i_{t-1}} \hat{q}_{t-2, i_{t-1}} \right\|_2^2 \tag{37}$$

$$+ a_t \mathbb{E}_t \left[ \langle \nabla \ell(w_t)^\top q_t - \hat{g}_{t-1}^\top \hat{q}_{t-1}, w_\star - w_t \rangle \right] - a_{t-1} \langle \nabla \ell(w_{t-1})^\top q_{t-1} - \hat{g}_{t-2}^\top \hat{q}_{t-2}, w_\star - w_{t-1} \rangle$$

$$+ \frac{a_t \mu}{2} \|w_t\|_2^2 - a_t \nu D(q_t \| q_0) + \frac{\mu A_t}{2} \|w_\star - w_t\|_2^2$$

$$+ \frac{\mu C_{t-1}}{4} \|w_t - w_{t-1}\|_2^2 - \frac{\mu C_{t-1}}{2} \|w_\star - w_{t-1}\|_2^2$$

$$+ \frac{c_{t-1} \mu}{2} \sum_{\tau=t-n/b}^{t-2} \|w_t - w_{\tau \vee 0}\|_2^2 - \frac{c_{t-1} \mu}{2} \sum_{\tau=t-n/b}^{t-2} \|w_\star - w_{\tau \vee 0}\|_2^2.$$

Finally, apply (36) to the term (37) to achieve

$$\geq a_t q_t^\top \ell(w_t) + \frac{a_t}{2L} \sum_{i=1}^n q_{t,i} \|\nabla \ell_i(w_t) - \nabla \ell_i(w_\star)\|_2^2$$

$$- \frac{3 \kappa_\mathcal{Q} a_{t-1}^2}{\mu C_{t-1}} \sum_{i=1}^n q_{t-1,i} \|\nabla \ell_i(w_{t-1}) - \nabla \ell_i(w_\star)\|_2^2 - \frac{3 \kappa_\mathcal{Q} a_{t-1}^2}{\mu C_{t-1}} \sum_{i=1}^n q_{\pi(t-2,i),i} \|\nabla \ell_i(w_{\pi(t-2,i)} - \nabla \ell_i(w_\star))\|_2^2$$

$$- \frac{3n G^2 a_{t-1}^2}{\mu C_{t-1}} \|q_{t-1} - \hat{q}_{t-2}\|_2^2$$

$$+ a_t \mathbb{E}_t \left[ \langle \nabla \ell(w_t)^\top q_t - \hat{g}_{t-1}^\top \hat{q}_{t-1}, w_\star - w_t \rangle \right] - a_{t-1} \langle \nabla \ell(w_{t-1})^\top q_{t-1} - \hat{g}_{t-2}^\top \hat{q}_{t-2}, w_\star - w_{t-1} \rangle$$

$$+ \frac{a_t \mu}{2} \|w_t\|_2^2 - a_t \nu D(q_t \| q_0) + \frac{\mu A_t}{2} \|w_\star - w_t\|_2^2$$

$$+ \frac{\mu C_{t-1}}{4} \|w_t - w_{t-1}\|_2^2 - \frac{\mu C_{t-1}}{2} \|w_\star - w_{t-1}\|_2^2$$

$$+ \frac{c_{t-1} \mu}{2} \sum_{\tau=t-n/b}^{t-2} \|w_t - w_{\tau \vee 0}\|_2^2 - \frac{c_{t-1} \mu}{2} \sum_{\tau=t-n/b}^{t-2} \|w_\star - w_{\tau \vee 0}\|_2^2.$$

Next, we apply $a_{t-1} \leq 2\alpha C_{t-1}$ and $\alpha \leq \frac{\mu}{24eL\kappa_\mathcal{Q}}$ to achieve:

$$\frac{3nG^2 a_{t-1}^2}{\mu C_{t-1}} \leq \frac{6nG^2\alpha a_{t-1}}{\mu},$$

$$\frac{3\kappa_\mathcal{Q} a_{t-1}^2}{\mu C_{t-1}} \leq \frac{a_{t-1}}{4L},$$

$$\frac{3\kappa_\mathcal{Q} a_{t-1}^2}{\mu C_{t-1}} \leq \frac{a_{\pi(t-2,i)}}{4L}, \ \forall i \in [n]$$

For terms that contain $\pi(t-2,i)$, we recall that $\pi(t-2,i)$ can be at most $n/b$ timesteps behind $t-2$, so we have that

$$a_{t-1} \leq \left(1 + \frac{1}{n/b}\right)^{n/b} a_{t-n/b} \leq e a_{t-n/b} \leq e a_{\pi(t-2,i)}.$$

We use here that $a_t \leq (1 + \frac{\alpha}{4})a_{t-1}$ and impose the condition $\frac{\alpha}{4} \leq \frac{b}{n}$ in the rate. Substituting this back into the lower bound achieves the desired claim. $\quad\square$

### C.3.2 Upper Bound on Gap Criterion

Next, we quantify the gap between $\mathcal{L}(w_t, q_\star)$ and $\mathcal{L}(w_\star, q_\star)$ by upper bounding $a_t\mathcal{L}(w_t, q_\star)$, as given in Lem. 12. When combined with the lower bound Lem. 11, we may then control the gap.

**Lemma 12.** *For $t \geq 2$, we have that:*

$$a_t\mathcal{L}(w_t, q_\star) \leq a_t q_\star^\top (\ell(w_t) - v_t^D) + \frac{a_t\mu}{2}\|w_t\|_2^2 + A_{t-1}\nu\Delta_D(q_\star, q_{t-1})$$
$$+ a_t q_t^\top v_t^D - a_t\nu D(q_t\|q_0) - A_{t-1}\nu\Delta_D(q_t, q_{t-1}) - A_t\nu\Delta_D(q_\star, q_t).$$

*Proof.* We add and subtract (1) terms in the dual update step and apply the definition of the proximal operator (2) with Bregman divergence, and the optimality of $q_t$ for the maximization problem that defines it.

$$a_t\mathcal{L}(w_t, q_\star) := a_t q_\star^\top \ell(w_t) - a_t\nu D(q_\star\|q_0) + \frac{a_t\mu}{2}\|w_t\|_2^2$$
$$\overset{(1)}{=} a_t q_\star^\top (\ell(w_t) - v_t^D) + \frac{a_t\mu}{2}\|w_t\|_2^2 + A_{t-1}\nu\Delta_D(q_\star, q_{t-1}))$$
$$+ a_t q_\star^\top v_t^D - a_t\nu D(q_\star\|q_0) - A_{t-1}\nu\Delta_D(q_\star, q_{t-1}))$$
$$\overset{(2)}{\leq} a_t q_\star^\top (\ell(w_t) - v_t^D) + \frac{a_t\mu}{2}\|w_t\|_2^2 + A_{t-1}\nu\Delta_D(q_\star, q_{t-1}))$$
$$+ a_t q_t^\top v_t^D - a_t\nu D(q_t\|q_0) - A_{t-1}\nu\Delta_D(q_t, q_{t-1}) - A_t\nu\Delta_D(q_\star, q_t).$$
$$\square$$

We can combine the upper bound from Lem. 12 and lower bound from Lem. 11 in Lem. 13. We identify telescoping terms in blue and non-positive terms in red. The green term is canceled after aggregation across time $t$. This bound, like before applies for $t \geq 2$.

**Lemma 13.** *Assume that $\alpha \leq \mu/(24eL\kappa_{\mathcal{Q}})$. For $t > 2$, we have that:*

$$\mathbb{E}_t\left[\gamma_t\right] = \mathbb{E}_t\left[a_t(\mathcal{L}(w_t, q_\star) - \mathcal{L}(w_\star, q_t) - \frac{a_t\mu}{2}\left\|w_t - w_\star\right\|_2^2 - \frac{a_t\nu}{2}\left\|q_t - q_\star\right\|_2^2\right]$$

$$\leq a_t\mathbb{E}_t\left[(q_\star - q_t)^\top(\ell(w_t) - \hat{\ell}_t)\right] - a_{t-1}(q_\star - q_{t-1})^\top(\ell(w_{t-1}) - \hat{\ell}_{t-1}) \tag{38}$$

$$+ A_{t-1}\nu\Delta_D(q_\star, q_{t-1}) - A_t\nu\mathbb{E}_t\left[\Delta_D(q_\star, q_t)\right] \tag{39}$$

$$- \frac{a_t}{2L}\mathbb{E}_t\left[\sum_{i=1}^n q_{t,i}\left\|\nabla\ell_i(w_t) - \nabla\ell_i(w_\star)\right\|_2^2\right] \tag{40}$$

$$+ \frac{a_{t-1}}{4L}\sum_{i=1}^n q_{t-1,i}\left\|\nabla\ell_i(w_{t-1}) - \nabla\ell_i(w_\star)\right\|_2^2 + \sum_{i=1}^n \frac{a_{\pi(t-2,i)}}{4L}q_{\pi(t-2,i),i}\left\|\nabla\ell_i(w_{\pi(t-2,i)} - \nabla\ell_i(w_\star))\right\|_2^2 \tag{41}$$

$$- a_t\mathbb{E}_t\left[\left\langle\nabla\ell(w_t)^\top q_t - \hat{g}_{t-1}^\top\hat{q}_{t-1}, w_\star - w_t\right\rangle\right] + a_{t-1}\left\langle\nabla\ell(w_{t-1})^\top q_{t-1} - \hat{g}_{t-2}^\top\hat{q}_{t-2}, w_\star - w_{t-1}\right\rangle \tag{42}$$

$$+ \frac{6nG^2\alpha a_{t-1}}{\mu}\left\|q_{t-1} - \hat{q}_{t-2}\right\|_2^2 \tag{43}$$

$$- \frac{\mu A_t}{2}\mathbb{E}_t\left\|w_\star - w_t\right\|_2^2 + \frac{\mu C_{t-1}}{2}\left\|w_\star - w_{t-1}\right\|_2^2 + \frac{c_{t-1}\mu}{2}\sum_{\tau=t-n/b}^{t-2}\left\|w_\star - w_{\tau\vee 0}\right\|_2^2 \tag{44}$$

$$+ \frac{na_{t-1}\alpha G^2}{\nu}\sum_{\tau=t-n/b}^{t-3}\left\|w_{t-1} - w_{\tau\vee 0}\right\|_2^2 - \frac{c_{t-1}\mu}{2}\sum_{\tau=t-n/b}^{t-2}\mathbb{E}_t\left\|w_t - w_{\tau\vee 0}\right\|_2^2 \tag{45}$$

$$+ \frac{na_{t-1}\alpha G^2}{\nu}\left\|w_{t-1} - w_{t-2}\right\|_2^2 - \frac{\mu C_{t-1}}{4}\mathbb{E}_t\left\|w_t - w_{t-1}\right\|_2^2 \tag{46}$$

$$- \frac{a_t\mu}{2}\mathbb{E}_t\left[\left\|w_t - w_\star\right\|_2^2\right] - \frac{a_t\nu}{2}\mathbb{E}_t\left\|q_t - q_\star\right\|_2^2 - \frac{A_{t-1}\nu}{2}\mathbb{E}_t\left[\Delta_D(q_t, q_{t-1})\right]. \tag{47}$$

*For $t = 2$, the above holds with the addition of the term $\frac{\nu}{4}\left\|q_1 - q_0\right\|_2^2 + \frac{nG^2}{\nu}\left\|w_0 - w_1\right\|_2^2$.*

*Proof.* First, combine Lem. 11 and Lem. 12 to write:

$$\mathbb{E}_t\left[\gamma_t\right] = \mathbb{E}_t\left[a_t(\mathcal{L}(w_t, q_\star) - \mathcal{L}(w_\star, q_t) - \frac{a_t\mu}{2}\|w_t - w_\star\|_2^2 - \frac{a_t\nu}{2}\|q_t - q_\star\|_2^2\right]$$

$$\leq \underbrace{\mathbb{E}_t\left[a_t(q_\star - q_t)^\top(\ell(w_t) - v_t^D)\right]}_{\text{cross term}}$$

$$+\mathbb{E}_t\left[A_{t-1}\nu\Delta_D(q_\star, q_{t-1}) - A_t\nu\Delta_D(q_\star, q_t)\right]$$

$$-\mathbb{E}_t\left[\frac{a_t}{2L}\sum_{i=1}^n q_{t,i}\|\nabla\ell_i(w_t) - \nabla\ell_i(w_\star)\|_2^2\right]$$

$$+\frac{a_{t-1}}{4L}\sum_{i=1}^n q_{t-1,i}\|\nabla\ell_i(w_{t-1}) - \nabla\ell_i(w_\star)\|_2^2 + \sum_{i=1}^n \frac{a_{\pi(t-2,i)}}{4L}q_{\pi(t-2,i),i}\left\|\nabla\ell_i(w_{\pi(t-2,i)}) - \nabla\ell_i(w_\star)\right\|_2^2$$

$$+\frac{6nG^2\alpha a_{t-1}}{\mu}\|q_{t-1} - \hat{q}_{t-2}\|_2^2$$

$$-a_t\mathbb{E}_t\left[\left\langle\nabla\ell(w_t)^\top q_t - \hat{g}_{t-1}^\top\hat{q}_{t-1}, w_\star - w_t\right\rangle\right] + a_{t-1}\left\langle\nabla\ell(w_{t-1})^\top q_{t-1} - \hat{g}_{t-2}^\top\hat{q}_{t-2}, w_\star - w_{t-1}\right\rangle$$

$$-\frac{\mu A_t}{2}\mathbb{E}_t\|w_\star - w_t\|_2^2 + \frac{\mu C_{t-1}}{2}\|w_\star - w_{t-1}\|_2^2 + \frac{c_{t-1}\mu}{2}\sum_{\tau=t-n/b}^{t-2}\|w_\star - w_{\tau\vee 0}\|_2^2$$

$$-\frac{\mu C_{t-1}}{4}\mathbb{E}_t\|w_t - w_{t-1}\|_2^2 - A_{t-1}\nu\mathbb{E}_t[\Delta_D(q_t, q_{t-1})]$$

$$-\frac{a_t\mu}{2}\mathbb{E}_t\left[\|w_t - w_\star\|_2^2\right] - \frac{c_{t-1}\mu}{2}\sum_{\tau=t-n/b}^{t-2}\mathbb{E}_t\|w_t - w_{\tau\vee 0}\|_2^2 - \frac{a_t\nu}{2}\mathbb{E}_{i_t}\left[\|q_t - q_\star\|_2^2\right].$$

Bound the cross term identified above. In the case that $t = 2$, use Lem. 4 with $v_t = v_t^D$, $y_t = \ell(w_t)$, $\hat{y}_{t+1} = \hat{\ell}_t$, $x_t = q_t$, $x_\star = q_\star$, and $\gamma = \nu A_{t-1} = \nu$ which yields that

$$a_t\mathbb{E}_t\left[(q_\star - q_2)^\top(\ell(w_2) - v_2^D)\right]$$

$$\leq a_2\mathbb{E}_2\left[(q_\star - q_2)^\top(\ell(w_t) - \hat{\ell}_2)\right] - a_1(q_\star - q_1)^\top(\ell(w_1) - \hat{\ell}_1)$$

$$+\frac{\nu}{4}\mathbb{E}_2\left[\|q_1 - q_0\|_2^2\right] + \frac{n^2}{\nu}\mathbb{E}_2\left\|\frac{1}{b}\sum_{j\in J_t}(\ell_j(w_1) - \hat{\ell}_{1,j})e_j\right\|_2^2, \tag{48}$$

where the fourth term above can be bounded as

$$\frac{n^2}{\nu}\mathbb{E}_2\left\|\frac{1}{b}\sum_{j\in J_t}(\ell_j(w_1) - \hat{\ell}_{1,j})e_j\right\|_2^2 \leq \frac{nb}{\nu}\left\|\frac{1}{b}\sum_{j=1}^b(\ell_j(w_1) - \hat{\ell}_{1,j})e_j\right\|_2^2 \leq \frac{nG^2}{\nu}\|w_1 - w_0\|_2^2.$$

Using the definition of the update, we have that $\|w_1 - w_0\|_2^2 = (1/\mu^2)\left\|\nabla\ell(w_0)^\top q_0\right\|_2^2$. In the case that $t > 2$, use Lem. 4 as above but instead with $\gamma = \nu A_{t-2}$ which yields that

$$a_t\mathbb{E}_t\left[(q_\star - q_t)^\top(\ell(w_t) - v_t^D)\right]$$

$$\leq a_t\mathbb{E}_t\left[(q_\star - q_t)^\top(\ell(w_t) - \hat{\ell}_t)\right] - a_{t-1}(q_\star - q_{t-1})^\top(\ell(w_{t-1}) - \hat{\ell}_{t-1})$$

$$+\frac{A_{t-2}\nu}{4}\mathbb{E}_t\left[\|q_t - q_{t-1}\|_2^2\right] + \frac{n^2 a_{t-1}^2}{A_{t-2}\nu}\underbrace{\mathbb{E}_t\left\|\frac{1}{b}\sum_{j\in J_t}(\ell_j(w_{t-1}) - \hat{\ell}_{t-1,j})e_j\right\|_2^2}_{\mathbb{E}_t\|\delta_t^D\|_2^2}. \tag{49}$$

We may then apply Lem. 6 and $a_{t-1} \leq \alpha A_{t-2}$ to get that

$$\frac{n^2 a_{t-1}^2}{A_{t-2}\nu}\mathbb{E}_t\left\|\delta_t^D\right\|_2^2 \leq \frac{na_{t-1}\alpha G^2}{\nu}\sum_{\tau=t-n/b}^{t-2}\|w_{t-1} - w_{\tau\vee 0}\|_2^2. \tag{50}$$

We may use strong convexity to get the $\frac{A_{t-2}\nu}{4}\mathbb{E}_t\left[\|q_t - q_{t-1}\|_2^2\right]$ term to cancel with $-\frac{A_{t-1}\nu}{2}\Delta_D(q_t, q_{t-1})$ and that $A_{t-2} \leq A_{t-1}$ to complete the proof. $\qquad\square$

### C.3.3 Determining Constants

In this section, we provide derivations that determine the values of the constant $c_t$ that allow for cancellation of errors. We slightly adjust the notation in this subsection, in that we assume that for some $\eta > 0$

$$a_t \leq (1 + \eta)a_{t-1}$$

and determine $\eta$ such that (28) is satisfied. We will see that $\eta$ is simply a constant factor away from $\alpha$, so the resulting condition we actually be on $\alpha$. The latter is given formally in Lem. 14. We assume here that $n/b \geq 2$, which is taken as an assumption of Thm. 2.

In the statement of Lem. 13, the lines above (43) will telescope without additional conditions. For (44), we set $c_t = a_t/m$ for some parameter $m$. Note that this condition does not need to be checked when $n/b < 1$, as the additional sum term over $\tau$ will not be included in the update. Counting all the terms that will appear when matched on the index $t - 1$, we have the condition that

$$-\frac{a_{t-1}}{4} - A_{t-1} + C_{t-1} + \sum_{s=t+1}^{t+n/b-1} a_s/m \leq 0.$$

The first term result from the "good term" $-\frac{a_{t-1}\mu}{4} \|w_{t-1} - w_\star\|_2^2$ from the bottom. The rightmost term above results because $a_s \|w_\star - w_{\tau \vee 0}\|_2^2$ will have $\tau = t-1$ when $s \in \{t+1, \ldots, t+n/b-1\}$. We will begin by requiring that require that $a_t \leq (1 + \beta)a_{t-1}$ for all $t$ and some $\beta > 0$, and then determine $\beta$ below. The condition reads as

$$\frac{4n/b - 4 + m}{4m} a_{t-1} \geq \frac{(1+\beta)^2}{m} \sum_{s=0}^{n/b-2} (1+\beta)^s a_{t-1},$$

which can be summed and represented as

$$\frac{(4n/b - 4 + m)}{4} \geq (1+\beta)^2 \frac{(1+\beta)^{n/b-1} - 1}{\beta}.$$

Rearranging and taking a logarithm on both sides, this is the same as

$$\ln\left(\frac{\beta(4n/b - 4 + m)}{4(1+\beta)^2} + 1\right) \geq (n/b - 1)\log(1 + \beta). \tag{51}$$

Next, using the inequality $\frac{2x}{2+x} \leq \ln(1 + x)$ with $x = \frac{\beta(4n/b-4+m)}{4(1+\beta)^2}$ which holds for all $x \geq 0$, we have

$$\ln\left(\frac{\beta(4n/b - 4 + m)}{4(1+\beta)^2} + 1\right) \geq \frac{2\frac{\beta(4n/b-4+m)}{4(1+\beta)^2}}{2 + \frac{\beta(4n/b-4+m)}{4(1+\beta)^2}} = \frac{2\beta(4n/b - 4 + m)}{8(1+\beta)^2 + \beta(4n/b - 4 + m)} \tag{52}$$

We can also apply the upper bound $\ln(x + 1) \leq x$ with $x = \beta$ (which also holds for any $x \geq 0$) to write

$$(n/b - 1)\log(1 + \beta) \leq (n/b - 1)\beta,$$

which means that (51) will be satisfied (using (52)) if

$$\frac{2\beta(4n/b - 4 + m)}{8(1+\beta)^2 + \beta(4n/b - 4 + m)} \geq (n/b - 1)\beta \tag{53}$$

$$\iff \frac{n/b - 1 + m/4}{n/b - 1} \geq (1+\beta)^2 + (\beta/2)(n/b - 1 + m/4). \tag{54}$$

In order to satisfy the inequality, substitute $m = 4c\beta(n/b - 1)^2$ for some $c > 0$ to be determined and assume that $\beta \leq \frac{1}{n/b-1}$, so that $\beta(n/b - 1) \leq 1$. The LHS reads as

$$\frac{n/b - 1 + m/2}{n/b - 1} = 1 + c\beta(n/b - 1).$$

The RHS can be upper-bounded as

$$(1 + \beta)^2 + (\beta/2)(n/b - 1 + m/4) = (1 + \beta)^2 + (\beta/2)(n/b - 1 + c\beta(n/b - 1)^2)$$
$$\leq 1 + \beta(2 + \beta) + \beta(n/b - 1)\frac{1 + c}{2},$$

which makes the inequality satisfied when

$$c \geq 2\left(\frac{4 + 2\beta}{n/b - 1} + 1\right),$$

so we can set $c = 16$. We now have the flexibility to control $\beta$, and the telescoping of (44) is achieved. For (45), set $\beta = \frac{\alpha}{4}$ and pass the condition of $\beta \leq 1/(n/b - 1)$ onto $\alpha$, which maintains the rate (and is already satisfied when $\alpha \leq \frac{b}{n}$ and $n \geq 2$). Then, we can achieve

$$\frac{na_t\alpha G^2}{\nu} \leq \frac{\mu a_{t-1}}{m} = \frac{\mu a_{t-1}}{4\alpha(n/b - 1)^2}$$

by requiring that $\alpha \leq \frac{b\sqrt{\mu\nu}}{4n^{3/2}G}$, which achieves the telescoping of (45). Finally, to address (46), we may satisfy it if

$$\frac{na_t\alpha G^2}{\nu} \leq \frac{\mu C_{t-1}}{4}$$

which we can achieve by incorporating the condition $a_t \leq 4\alpha(n/b - 1)^2 C_{t-1}$ into the update of $a_t$, because $\frac{na_t\alpha G^2}{\nu} \leq \frac{\mu a_t}{m}$ by the previous condition on $\alpha$. Having chosen $c_t$, we are prepared to produce a learning rate parameter $\eta$ to capture all conditions on $\alpha$ in one, as given in Lem. 14.

**Lemma 14.** *For all $t \geq 1$, we have the following.*

- *Setting $c_t = a_t/[16\alpha(n/b - 1)^2]$ implies that $a_t \leq 2\alpha C_t$.*

- *Using the update scheme*

$$a_2 = 4\eta a_1, \text{ and } a_t = (1 + \eta)\, a_{t-1} \text{ for } t > 2,$$

  *when*

$$\eta = \frac{1}{4}\min\left\{\frac{b}{32n}, \frac{\mu}{24eL\kappa_Q}, \frac{b}{n}\sqrt{\frac{nG^2}{\mu\nu}}\right\}.$$

  *we have that (28) holds.*

*Proof.* Noting that $m = 16\alpha(n/b - 1)^2$ we confirm that

$$C_t \geq A_{t-1}/2$$
$$\iff A_t - \frac{1}{16\alpha(n/b - 1)}a_t \geq A_{t-1}/2$$
$$\iff \left(1 - \frac{1}{16\alpha(n/b - 1)}\right)a_t \geq -A_{t-1}/2$$
$$\iff 2\left(\frac{1}{16\alpha(n/b - 1)} - 1\right)a_t \leq A_{t-1}.$$

This condition is satisfied when $\alpha \leq \frac{1}{32(n/b-1)}$, so we incorporate $\alpha \leq \frac{b}{32n}$ into the rate, implying that $a_t \leq 2\alpha C_t$. This concludes the proof of the first bullet point.

Next, we show that $a_t \leq (1+\alpha/4)a_{t-1}$ will imply that $a_t \leq \alpha A_{t-1}$, which is the second part of (28). First, we define $a_2 = \alpha a_1$ as an initialization (which also satisfies $a_2 \leq (1+\alpha/4)a_1$ when $\alpha \leq 4/3$), and show inductively that if $a_{t-1} \leq \alpha A_{t-2}$, then $a_t \leq (1 + \alpha/4)a_{t-1} \implies a_t \leq \alpha A_{t-1}$. In the base case, $A_1 = a_1$, so the condition $a_2 = \alpha a_1$ satisfies $a_2 \leq \alpha A_1$. Next, fixing $t$ and assuming that 1) $a_{t-1} \leq \alpha A_{t-2}$ and 2) that $a_t \leq (1 + \alpha/4)a_{t-1}$, we have that

$$\alpha A_{t-1} = \alpha(a_{t-1} + A_{t-2}) \geq (\alpha + 1)a_{t-1} \geq a_t,$$

the desired result. Finally, we consider the condition $a_t \leq 4\alpha(n/b-1)^2 C_{t-1}$, the third part of (28). We wish to show that the following inequality holds:

$$a_t \leq 4\alpha(n/b-1)^2 C_{t-1} = 4\alpha(n/b-1)^2 \left( A_{t-1} - \frac{1}{4\alpha(n/b-1)}a_{t-1} \right)$$

$$= 4\alpha(n/b-1)^2 \left( a_{t-1} + A_{t-2} - \frac{1}{4\alpha(n/b-1)}a_{t-1} \right)$$

which is implied by the inequality

$$a_t \leq 4\alpha(n/b-1)^2 \left( a_{t-1} + (1/\alpha)a_{t-1} - \frac{1}{4\alpha(n/b-1)}a_{t-1} \right)$$

$$= 4(n/b-1)^2 \left( \alpha + 1 - \frac{1}{4(n/b-1)} \right) a_{t-1}.$$

When $n/b \geq 2$, we have that $(1 + \alpha/4) \leq 4(n/b-1)^2 \left( \alpha + 1 - \frac{1}{4(n/b-1)} \right)$, so we require that $b \leq n/2$. Thus, our final updates are given by

$$a_2 = 4\eta a_1 \leq (1 + \eta) a_1 \text{ and } a_t = (1 + \eta) a_{t-1} \text{ for } t > 2.$$

Because each condition was satisfied when using $a_t = (1+\alpha/4)a_{t-1}$, we define $\eta = \alpha/4$ to achieve the claimed result. $\qquad\square$

### C.3.4 Bound on Sum of Successive Gaps

Lem. 15 is an upper estimate for the expected sum of the gap function over $T$ iterates. Recall that $\mathbb{E}_1[\cdot]$ the full expectation over $\{(I_t, J_t)\}_{t=1}^T$. The green term is a quantity that remain as an initialization term, whereas the blue terms have to be bounded from above. The terms directly below the blue terms account for all of the "negative $\pi(t-1, i)$" terms are not yet used up by the telescoping in lines (39), (40), and (41), and there are in fact between 1 and $n$ copies of those terms in each iteration, even though we will use only 1.

**Lemma 15** (Progress Bound). *Assume that*

$$\alpha \leq \min \left\{ \frac{b}{32n}, \frac{\mu}{24eL\kappa_Q}, \frac{b}{36e^2n} \sqrt{\frac{\mu\nu}{nG^2}} \right\}.$$

*For any $T \geq 1$, we have that*

$$\mathbb{E}_0 \left[ \sum_{t=1}^{T} \gamma_t \right] \leq \frac{nG^2}{\nu\mu^2} \left\| \nabla\ell(w_0)^\top q_0 \right\|_2^2$$

$$+ a_T \mathbb{E}_0 \left[ (q_\star - q_T)^\top (\ell(w_T) - \hat{\ell}_T) \right] \tag{55}$$

$$- a_T \mathbb{E}_0 \left[ \langle \nabla\ell(w_T)^\top q_T - \hat{g}_{T-1}^\top \hat{q}_{T-1}, w_\star - w_T \rangle \right] \tag{56}$$

$$- \sum_{i=1}^{n} (n - T + \pi(T-1,i)) \frac{a_{\pi(T-1,i)}}{4L} \mathbb{E}_0 \left[ q_{\pi(T-1,i)} \left\| \nabla\ell_i(w_{\pi(T-1,i)}) - \nabla\ell_i(w_\star) \right\|_2^2 \right] \tag{57}$$

$$+ \frac{6nG^2\alpha}{\mu} \mathbb{E}_0 \sum_{t=1}^{T} a_{t-1} \left\| q_{t-1} - \hat{q}_{t-2} \right\|_2^2 - \sum_{t=1}^{T} \frac{A_{t-1}\nu}{2} \mathbb{E}_0 \left[ D(q_t \| q_{t-1}) \right] \tag{58}$$

$$- \frac{a_T}{2L} \sum_{i=1}^{n} \mathbb{E}_0 \left[ q_{T,i} \left\| \nabla\ell_i(w_T) - \nabla\ell_i(w_\star) \right\|_2^2 \right]$$

$$- \frac{\mu A_T}{2} \mathbb{E}_0 \left[ \left\| w_\star - w_T \right\|_2^2 \right] \tag{59}$$

$$- \frac{\mu a_{T-1}}{2[16\alpha(n/b-1)]} \sum_{\tau=T-\lceil n/b \rceil}^{T-2} \mathbb{E}_0 \left[ \left\| w_T - w_{\tau\vee 0} \right\|_2^2 \right] \tag{60}$$

$$- A_T \nu \mathbb{E}_0 \left[ D(q_\star \| q_T) \right] \tag{61}$$

$$- \sum_{t=1}^{T} \frac{a_t \mu}{4} \mathbb{E} \left\| w_t - w_\star \right\|_2^2 - \sum_{t=1}^{T} \frac{a_t \nu}{2} \mathbb{E}_0 \left\| q_t - q_\star \right\|_2^2. \tag{62}$$

*Proof.* We proceed by first deriving an upper bound on $\gamma_1$, the gap function for $t = 1$. Note that $w_1$ is non-random, as $a_0 = 0$ implies that $v_0 = \nabla\ell(w_0)$. Using that $w_1$ is the optimum for the proximal operator that defines it, the upper bound can be written as

$$a_1 \mathcal{L}(w_1, q_\star) \leq a_1 q_\star^\top (\ell(w_1) - \hat{\ell}_1) + \frac{a_1\mu}{2} \left\| w_1 \right\|_2^2 + a_1 q_1^\top \hat{\ell}_1 - a_1 \nu D(q_1 \| q_0) - A_1 \nu \Delta_D(q_\star, q_1),$$

where we use that $\tilde{\ell}_1 = \hat{\ell}_1$. For the lower bound, use a similar argument to Lem. 11 to achieve

$$a_1 \mathcal{L}(w_\star, q_1) \geq a_1 q_1^\top \ell(w_1) + a_1 \langle \nabla\ell(w_1)^\top q_1 - \nabla\ell(w_0)^\top q_0, w_\star - w_1 \rangle + \frac{a_1\mu}{2} \left\| w_1 \right\|_2^2$$

$$- a_1 \nu D(q_1 \| q_0) + \frac{a_1}{2L} \sum_{i=1}^{n} q_{1,i} \left\| \nabla\ell_i(w_1) - \nabla\ell_i(w_\star) \right\|_2^2 + \frac{\mu A_1}{2} \left\| w_\star - w_1 \right\|_2^2,$$

where we use that $v_0^P = \nabla\ell(w_0)^\top q_0$. We combine them to get

$$\gamma_1 \leq a_1 (q_\star - q_1)^\top (\ell(w_1) - \hat{\ell}_1) - a_1 \langle \nabla\ell(w_1)^\top q_1 - \nabla\ell(w_0)^\top q_0, w_\star - w_1 \rangle$$

$$- \frac{a_1}{2L} \sum_{i=1}^{n} q_{1,i} \left\| \nabla\ell_i(w_1) - \nabla\ell_i(w_\star) \right\|_2^2 - \frac{\mu A_1}{2} \left\| w_\star - w_1 \right\|_2^2 - A_1 \nu \Delta_D(q_\star, q_1)$$

$$- \frac{a_1\mu}{2} \left\| w_1 - w_\star \right\|_2^2 - \frac{a_1\nu}{2} \left\| q_1 - q_\star \right\|_2^2,$$

where the last two terms are the result of the additional quadratic slack terms in $\gamma_1$. All of the terms from the display above will be telescoped. Thus, we apply Lem. 13 and collect the unmatched terms from the $t \geq 2$ one-step bound (using that $A_0 = 0$). The term (57) can be viewed as counting the remainder of (40) after it has telescoped some but not all terms $\frac{a_{\pi(T-1,i)}}{4L} \mathbb{E}_0 \left[ q_{\pi(T-1,i)} \left\| \nabla\ell_i(w_{\pi(T-1,i)}) - \nabla\ell_i(w_\star) \right\|_2^2 \right]$ across iterations. $\square$

### C.3.5 Completing the Proof

We use similar techniques as before to bound the remaining terms from the $T$-step progress bound given in Lem. 15. We may now prove the main result.

**Theorem 2.** *For a constant $\alpha > 0$, define the sequence*

$$a_1 = 1, a_2 = 4\alpha, \text{ and } a_t = (1 + \alpha) a_{t-1} \text{ for } t > 2,$$

*along with its partial sum $A_t = \sum_{\tau=1}^{t} a_\tau$. Under Asm. 1, there is an absolute constant $C$ such that using the parameter*

$$\alpha = C \min \left\{ \frac{b}{n}, \frac{\mu}{L\kappa_{\mathcal{Q}}}, \frac{b}{n} \sqrt{\frac{\mu\nu}{nG^2}} \right\},$$

*the iterates of Algorithm 1 satisfy:*

$$\sum_{t=1}^{T} a_t \mathbb{E}_1[\gamma_t] + \frac{A_T \mu}{4} \mathbb{E}_1 \|w_T - w_\star\|_2^2 + \frac{A_T \nu}{4} \mathbb{E}_1 \|q_T - q_\star\|_2^2 \leq \frac{nG^2}{\nu} \|w_0 - w_1\|_2^2.$$

*We can compute a point $(w_T, q_T)$ achieving an expected gap no more than $\varepsilon$ with big-O complexity*

$$(n + bd) \cdot \left( \frac{n}{b} + \frac{L\kappa_{\mathcal{Q}}}{\mu} + \frac{n}{b} \sqrt{\frac{nG^2}{\mu\nu}} \right) \cdot \ln \left( \frac{1}{\varepsilon} \right). \tag{10}$$

*Proof.* We first apply Lem. 15, and proceed to bound the inner product terms (55) and (56). Apply Young's inequality with parameter $\nu A_{T-1}/2$ to get

$$a_T \mathbb{E}_0 \left[ (q_\star - q_T)^\top (\ell(w_T) - \hat{\ell}_T) \right] \leq \frac{\nu A_{T-1}}{4} \mathbb{E}_0 \|q_\star - q_T\|_2^2 + \frac{a_T^2}{\nu A_{T-1}} \mathbb{E}_0 \|\ell(w_T) - \hat{\ell}_T\|_2^2$$

$$\leq \frac{\nu A_{T-1}}{4} \mathbb{E}_0 \|q_\star - q_T\|_2^2 + \frac{a_T^2 G^2}{\nu A_{T-1}} \sum_{\tau=T-n/b}^{T-2} \mathbb{E}_0 \|w_T - w_{\tau \vee 0}\|_2^2$$

$$\leq \frac{\nu A_{T-1}}{4} \mathbb{E}_0 \|q_\star - q_T\|_2^2 + \frac{a_T \alpha G^2}{\nu} \sum_{\tau=T-n/b}^{T-2} \mathbb{E}_0 \|w_T - w_{\tau \vee 0}\|_2^2.$$

The left-hand term will be canceled by (61) by applying strong concavity (leaving behind $-\frac{\nu A_{T-1}}{4} \mathbb{E}_0 \|q_\star - q_T\|_2^2$) and the right-hand term (because of the condition $\alpha \leq \frac{\sqrt{\mu\nu}}{4n^{3/2}G}$) will be canceled by (60). Next, consider (56). By Young's inequality with parameter $\mu A_{T-1}/2$, we have

$$- a_T \mathbb{E}_0 \left[ \langle \nabla \ell(w_T)^\top q_T - \hat{g}_{T-1}^\top \hat{q}_{T-1}, w_\star - w_T \rangle \right]$$

$$\leq \frac{a_T^2}{\mu A_{T-1}} \mathbb{E}_0 \left\| \nabla \ell(w_T)^\top q_T - \hat{g}_{T-1}^\top \hat{q}_{T-1} \right\|_2^2 + \frac{\mu A_{T-1}}{4} \mathbb{E}_0 \|w_\star - w_T\|_2^2$$

$$\leq \frac{a_T \alpha}{\mu} \mathbb{E}_0 \left\| \nabla \ell(w_T)^\top q_T - \hat{g}_{T-1}^\top \hat{q}_{T-1} \right\|_2^2 + \frac{\mu A_{T-1}}{4} \mathbb{E}_0 \|w_\star - w_T\|_2^2,$$

where the second term will be canceled by (59). For the remaining term,

$$\mathbb{E}_0 \left\| \nabla \ell(w_T)^\top q_T - \hat{g}_{T-1}^\top \hat{q}_{T-1} \right\|_2^2$$

$$\leq \mathbb{E}_0 \left\| (\nabla \ell(w_T) - \ell(w_\star))^\top q_T + \nabla \ell(w_\star)^\top (q_T - \hat{q}_{T-1}) + (\nabla \ell(w_\star) - \hat{g}_{T-1})^\top \hat{q}_{T-1} \right\|_2^2$$

$$\leq 3\mathbb{E}_0 \left\| (\nabla \ell(w_T) - \nabla \ell(w_\star))^\top q_T \right\|_2^2 + 3\mathbb{E}_0 \left\| \nabla \ell(w_\star)^\top (q_T - \hat{q}_{T-1}) \right\|_2^2 + 3\mathbb{E}_0 \left\| (\nabla \ell(w_\star) - \hat{g}_{T-1})^\top \hat{q}_{T-1} \right\|_2^2$$

$$\leq 3\sigma n \sum_{i=1}^{n} \mathbb{E}_0 \left[ q_{T,i} \|\nabla \ell_i(w_T) - \nabla \ell_i(w_\star)\|_2^2 \right] + 3nG^2 \mathbb{E}_0 \|q_T - \hat{q}_{T-1}\|_2^2 + 3\sigma n \sum_{i=1}^{n} \mathbb{E}_0 \left[ \hat{q}_{T-1,i} \|\nabla \ell_i(w_\star) - \hat{g}_{T-1,i}\|_2^2 \right]$$

We may add the middle term above to (58), so that the remaining term to bound is

$$\frac{6nG^2\alpha}{\mu} \sum_{t=1}^{T+1} a_{t-1}\mathbb{E}_0 \left\| q_{t-1} - \hat{q}_{t-2} \right\|_2^2 - \sum_{t=1}^{T} \frac{A_{t-1}\nu}{2}\mathbb{E}_0\left[\Delta_D(q_t, q_{t-1})\right].$$

To show that this quantity is non-negative, we use that $a_0 = 0$ and Lem. 10 (recalling that $M = n/b$ to see that

$$\frac{6nG^2\alpha}{\mu}\mathbb{E}_0 \sum_{t=1}^{T+1} a_t \left\| q_t - \hat{q}_{t-1} \right\|_2^2 \leq \frac{18e^2n^3G^2\alpha}{b^2\mu}\mathbb{E}_0 \sum_{t=1}^{T+1} a_{t-1} \left\| q_t - q_{t-1} \right\|_2^2 \leq \frac{18e^2n^3G^2\alpha^2}{b^2\mu}\mathbb{E}_0 \sum_{t=1}^{T} A_{t-1} \left\| q_t - q_{t-1} \right\|_2^2,$$

which will cancel with the rightmost term in (58) provided that $\alpha \leq \frac{b}{36e^2n}\sqrt{\frac{\mu\nu}{nG^2}}$. Thus, plugging the previous displays into Lem. 15, we have that

$$\mathbb{E}_0\left[\sum_{t=1}^{T} \gamma_t\right] \leq \frac{n^2G^2}{\nu\mu^2}\left\| \nabla\ell(w_0)^\top q_0 \right\|_2^2$$

$$+ \frac{3a_T\sigma_n\alpha}{2\mu}\sum_{i=1}^{n}\mathbb{E}_0\left[q_{T,i}\left\|\nabla\ell_i(w_T) - \nabla\ell_i(w_\star)\right\|_2^2\right] - \frac{a_T}{2L}\sum_{i=1}^{n}\mathbb{E}_0\left[q_{T,i}\left\|\nabla\ell_i(w_T) - \nabla\ell_i(w_\star)\right\|_2^2\right]$$

$$+ \frac{3\sigma_na_T\alpha}{2\mu}\sum_{i=1}^{n}\mathbb{E}_0\left[q_{\pi(T-1,i),i}\left\|\nabla\ell_i(w_\star) - \nabla\ell_i(w_{\pi(T-1,i)})\right\|_2^2\right]$$

$$- \sum_{i=1}^{n}(n - T + \pi(T-1,i))\frac{a_{\pi(T-1,i)}}{4L}\mathbb{E}_0\left[q_{\pi(T-1,i)}\left\|\nabla\ell_i(w_{\pi(T-1,i)}) - \nabla\ell_i(w_\star)\right\|_2^2\right]$$

$$- \sum_{t=1}^{T}\frac{a_t\mu}{4}\mathbb{E}_0\left\|w_t - w_\star\right\|_2^2 - \sum_{t=1}^{T}\frac{a_t\nu}{4}\mathbb{E}_0\left\|q_t - q_\star\right\|_2^2$$

$$- \frac{A_{T-1}\mu}{4}\mathbb{E}_0\left\|w_T - w_\star\right\|_2^2 - \frac{A_{T-1}\nu}{4}\mathbb{E}_0\left\|q_T - q_\star\right\|_2^2.$$

The black lines will cancel given our conditions on $\alpha$. Substituting the definition of $\gamma_t$ and moving the final non-positive terms on the last line, that is, $\frac{(A_{T-1}+a_T)\mu}{4}\mathbb{E}_0\left\|w_T - w_\star\right\|_2^2$ and $\frac{(A_{T-1}+a_T)\nu}{4}\mathbb{E}_0\left\|q_T - q_\star\right\|_2^2$ to the left-hand side achieves the claim. $\qquad\square$

### C.4 Modification for Unregularized Objectives

For completeness, we describe a modification of DRAGO for unregularized objectives, or (2) when $\mu \geq 0$ and $\nu \geq 0$. The analysis follows similarly to the previous subsections (regarding the $\mu, \nu > 0$ case), and we highlight the steps that differ in this subsection by presenting a slightly different upper bound on the gap criterion based on the modified primal and dual updates. This will result a different update for the sequence $(a_t)_{t\geq 1}$, subsequently affecting the rate.

#### C.4.1 Overview

The modified algorithm is nearly identical to Algorithm 1, except that the dual and primal updates can be written as

$$q_t := \arg\max_{q\in\mathcal{Q}} a_t\left\langle v_t^{\mathrm{D}}, q\right\rangle - a_t\nu D(q\|q_0) - (\nu A_{t-1} + \nu_1)\Delta_D(q, q_{t-1}) \qquad (63)$$

and

$$w_t := \arg\min_{w\in\mathcal{W}} a_t\left\langle v_t^{\mathrm{P}}, w\right\rangle + \frac{a_t\mu}{2}\left\|w\right\|_2^2 + \frac{C_{t-1}\mu + \mu_1}{2}\left\|w - w_{t-1}\right\|_2^2 + \frac{c_{t-1}\mu + \mu_2}{2}\sum_{s=t-n/b}^{t-2}\left\|w - w_{s\vee 0}\right\|_2^2, \qquad (64)$$

respectively, and $\mu_1, \mu_2, \nu_1 \geq 0$ are to-be-set hyperparameters. When $\nu > 0$, we may set $\nu_1 = 0$, and when $\mu > 0$, we may set $\mu_1 = \mu_2 = 0$, which recover the Algorithm 1 updates exactly. While

we may set $\nu_1 = 1$ when it is positive (and similarly for $\mu_1$ and $\mu_2$), they may be set to different values in order to balance the terms appearing in the rate below. As in Appx. C.1, we wish to upper bound the expectation of the quantity

$$\gamma_t = a_t \left( \mathcal{L}(w_t, q_\star) - \mathcal{L}(w_\star, q_t) - \frac{\mu}{2} \|w_t - w_\star\|_2^2 - \frac{\nu}{2} \|q_t - q_\star\|_2^2 \right) \tag{65}$$

which is still non-negative in the case of $\mu = 0$ or $\nu = 0$. By using an appropriate averaging sequence $(a_t)_{t \geq 1}$ and defining $A_T = \sum_{t=1}^{T} a_t$, we upper bound $\sum_{t=1}^{T} a_t \mathbb{E}_0[\gamma_t]$ (see Thm. 2) by a constant value independent of $T$. Recall that the batch size is denoted by $b$. As we derive in Appx. C.4.3, our final update on the $(a_t)$ sequence is

$$a_t = \min \left\{ \frac{C_{t-1}\mu + \mu_1}{12enq_{\max}L}, \left(1 + \frac{b}{n}\right) a_{t-1}, \frac{b}{32n} \frac{\sqrt{(A_{t-1}\nu + \nu_1)\min\{C_{t-1}\mu + \mu_1, c_{t-1}\mu + \mu_2\}}}{\sqrt{n}G} \right\}.$$

Observe that when $\mu = 0$, we set $a_t = \frac{\mu_1}{12enq_{\max}L}$ to achieve a $O(1/t)$ rate. We omit proofs in this subsection as they follow with the exact same steps as the corresponding lemmas in the strongly convex-strongly concave setting (which we point to for each result).

### C.4.2 Upper Bound on Gap Criterion

Following the steps of Appx. C.3.1 and Appx. C.3.2, we will first derive lower and upper bounds on $\mathbb{E}_t[a_t\mathcal{L}(w_\star, q_t)]$ and $a_t\mathcal{L}(w_t, q_\star)$ and combine them to upper bound $a_t\mathbb{E}_t[\gamma_t]$. Recalling that $\mathbb{E}_t[\cdot]$ denotes the condition expectation given $(w_{t-1}, q_{t-1})$, and we can then take the marginal expectation to upper bound $a_t\mathbb{E}_0[\gamma_t]$. The following lower bound is analogous to Lem. 11 and follows the exact same proof technique.

**Lemma 16.** *For $t \geq 2$, assuming that $a_t \leq \frac{C_{t-1}\mu + \mu_1}{12enq_{\max}L}$ and $a_t \leq (1 + b/n)a_{t-1}$, we have that:*

$$-\mathbb{E}_t[a_t\mathcal{L}(w_\star, q_t)]$$

$$\leq -a_t\mathbb{E}_t[q_t^\top \ell(w_t)] - \frac{a_t}{2L}\sum_{i=1}^{n} q_{t,i}\mathbb{E}_t\|\nabla\ell_i(w_t) - \nabla\ell_i(w_\star)\|_2^2 + \frac{a_t\mu}{2}\|w_t\|_2^2$$

$$+ \frac{a_{t-1}}{4L}\sum_{i=1}^{n} q_{t-1,i}\|\nabla\ell_i(w_{t-1}) - \nabla\ell_i(w_\star)\|_2^2 + \sum_{i=1}^{n} \frac{a_{\pi(t-2,i)}}{4L}q_{\pi(t-2,i),i}\|\nabla\ell_i(w_{\pi(t-2,i)} - \nabla\ell_i(w_\star))\|_2^2$$

$$+ \frac{3nG^2a_{t-1}^2}{C_{t-1}\mu + \mu_1}\|q_{t-1} - \hat{q}_{t-2}\|_2^2$$

$$- a_t\mathbb{E}_t\left[\langle \nabla\ell(w_t)^\top q_t - \hat{g}_{t-1}^\top \hat{q}_{t-1}, w_\star - w_t\rangle\right] + a_{t-1}\langle \nabla\ell(w_{t-1})^\top q_{t-1} - \hat{g}_{t-2}^\top \hat{q}_{t-2}, w_\star - w_{t-1}\rangle$$

$$+ a_t\nu\mathbb{E}_t[D(q_t\|q_0)] - \frac{A_t\mu + \mu_1 + \mu_2}{2}\mathbb{E}_t\|w_\star - w_t\|_2^2$$

$$- \frac{C_{t-1}\mu + \mu_1}{4}\|w_t - w_{t-1}\|_2^2 + \frac{C_{t-1}\mu + \mu_1}{2}\|w_\star - w_{t-1}\|_2^2$$

$$- \frac{c_{t-1}\mu + \mu_2}{2}\sum_{\tau=t-n/b}^{t-2} \mathbb{E}_t\|w_t - w_{\tau\vee 0}\|_2^2 + \frac{c_{t-1}\mu + \mu_2}{2}\sum_{\tau=t-n/b}^{t-2} \|w_\star - w_{\tau\vee 0}\|_2^2.$$

Similarly, following the same steps as Lem. 12, one can derive the following upper bound.

**Lemma 17.** *For $t \geq 2$, we have that:*

$$a_t\mathcal{L}(w_t, q_\star) \leq a_t q_\star^\top (\ell(w_t) - v_t^{\mathrm{D}}) + (A_{t-1}\nu + \nu_1)\Delta_D(q_\star, q_{t-1})$$

$$+ a_t q_t^\top v_t^{\mathrm{D}} - a_t\nu D(q_t\|q_0) - (A_{t-1}\nu + \nu_1)\Delta_D(q_t, q_{t-1}) - (A_t\nu + \nu_1)\Delta_D(q_\star, q_t).$$

By combining Lem. 17 and Lem. 16, we can upper bound the quantity $\mathbb{E}_t[a_t(\mathcal{L}(w_t, q_\star) - \mathcal{L}(w_\star, q_t))]$. Consequently, the following result follows the same steps as Lem. 13. As before, we identify telescoping terms in blue, non-positive terms in red, where green term is bounded after aggregation across time $t$.

**Lemma 18.** *Assume that $a_{t-1} \leq \frac{C_{t-1}\mu + \mu_1}{12eLnq_{\max}}$ and $a_t \leq (1+b/n)a_{t-1}$. For $t > 2$, we have that:*

$$\mathbb{E}_t\left[\gamma_t\right] = \mathbb{E}_t\left[a_t(\mathcal{L}(w_t, q_\star) - \mathcal{L}(w_\star, q_t)) - \frac{a_t\mu}{2}\|w_t - w_\star\|_2^2 - \frac{a_t\nu}{2}\|q_t - q_\star\|_2^2\right]$$

$$\leq a_t\mathbb{E}_t\left[(q_\star - q_t)^\top(\ell(w_t) - \hat{\ell}_t)\right] - a_{t-1}(q_\star - q_{t-1})^\top(\ell(w_{t-1}) - \hat{\ell}_{t-1}) \tag{66}$$

$$+(A_{t-1}\nu + \nu_1)\Delta_D(q_\star, q_{t-1}) - (A_t\nu + \nu_1)\mathbb{E}_t\left[\Delta_D(q_\star, q_t)\right] \tag{67}$$

$$-\frac{a_t}{2L}\mathbb{E}_t\left[\sum_{i=1}^n q_{t,i}\|\nabla\ell_i(w_t) - \nabla\ell_i(w_\star)\|_2^2\right] \tag{68}$$

$$+\frac{a_{t-1}}{4L}\sum_{i=1}^n q_{t-1,i}\|\nabla\ell_i(w_{t-1}) - \nabla\ell_i(w_\star)\|_2^2 + \sum_{i=1}^n \frac{a_{\pi(t-2,i)}}{4L}q_{\pi(t-2,i),i}\|\nabla\ell_i(w_{\pi(t-2,i)}) - \nabla\ell_i(w_\star))\|_2^2 \tag{69}$$

$$-a_t\mathbb{E}_t\left[\left\langle \nabla\ell(w_t)^\top q_t - \hat{g}_{t-1}^\top\hat{q}_{t-1}, w_\star - w_t\right\rangle\right] + a_{t-1}\left\langle \nabla\ell(w_{t-1})^\top q_{t-1} - \hat{g}_{t-2}^\top\hat{q}_{t-2}, w_\star - w_{t-1}\right\rangle \tag{70}$$

$$+\frac{3nG^2a_{t-1}^2}{C_{t-1}\mu + \mu_1}\|q_{t-1} - \hat{q}_{t-2}\|_2^2 \tag{71}$$

$$-\frac{A_t\mu + \mu_1 + \mu_2(n/b-1)}{2}\mathbb{E}_t\|w_\star - w_t\|_2^2 + \frac{\mu_1}{2}\|w_\star - w_{t-1}\|_2^2 + \frac{c_{t-1}\mu + \mu_2}{2}\sum_{\tau=t-n/b}^{t-2}\|w_\star - w_{\tau\vee 0}\|_2^2 \tag{72}$$

$$+\frac{na_{t-1}^2G^2}{A_{t-2}\nu + \nu_1}\sum_{\tau=t-n/b}^{t-3}\|w_{t-1} - w_{\tau\vee 0}\|_2^2 - \frac{c_{t-1}\mu + \mu_2}{2}\sum_{\tau=t-n/b}^{t-2}\mathbb{E}_t\|w_t - w_{\tau\vee 0}\|_2^2 \tag{73}$$

$$+\frac{na_{t-1}^2G^2}{A_{t-2}\nu + \nu_1}\|w_{t-1} - w_{t-2}\|_2^2 - \frac{C_{t-1}\mu + \mu_1}{4}\mathbb{E}_t\|w_t - w_{t-1}\|_2^2 \tag{74}$$

$$-\frac{a_t\mu}{2}\|w_t - w_\star\|_2^2 - \frac{a_t\nu}{2}\mathbb{E}_t\|q_t - q_\star\|_2^2 - \frac{A_{t-1}\nu + \nu_1}{2}\mathbb{E}_t\left[\Delta_D(q_t, q_{t-1})\right]. \tag{75}$$

*For $t = 2$, the above holds with the addition of the term $\frac{nG^2}{\nu+\nu_1}\|w_1 - w_0\|_2^2$ and without any term including $A_{t-2}$ in the denominator.*

We then select constants to achieve the desired telescoping in each line of Lem. 18.

### C.4.3 Determining Constants

We now select the constants $(\mu_1, \mu_2, \nu_1)$, and the sequences $(a_t)$ and $(c_t)$ to complete the main part of the analysis.

In the statement of Lem. 18, the lines above (71) will telescope without additional conditions. For (72), we set $c_t = a_t/m$ for some parameter $m$ (just as in Appx. C.3.3). Note that this condition does not need to be checked when $n/b < 1$, as the additional sum term over $\tau$ will not be included in the update. Counting all the terms that will appear when matched on the index $t - 1$, we have the condition that

$$\mu\underbrace{\left[-\frac{a_{t-1}}{4} - A_{t-1} + C_{t-1} + \sum_{s=t+1}^{t+n/b-1} a_s/m\right]}_{\leq 0} + \underbrace{-(\mu_1 + \mu_2(n/b-1)) + \mu_1 + (n/b-1)\mu_2}_{=0} \leq 0,$$

where the first underbrace is non-positive based on the choice of $m$ selected in Appx. C, and the equality is satisfied for all values of $\mu_1$ and $\mu_2$. Lines (73) and (74) yield the conditions

$$\frac{na_t^2G^2}{A_{t-1}\nu + \nu_1} \leq \frac{C_{t-1}\mu + \mu_1}{4} \quad \text{and} \quad \frac{na_t^2G^2}{A_{t-1}\nu + \nu_1} \leq \frac{c_{t-1}\mu + \mu_2}{2}. \tag{76}$$

for the telescoping to be achieved, which can equivalently be rewritten as

$$a_t \leq \sqrt{\frac{(A_{t-1}\nu + \nu_1)(C_{t-1}\mu + \mu_1)}{4nG^2}} \text{ and } a_t \leq \sqrt{\frac{(A_{t-1}\nu + \nu_1)(c_{t-1}\mu + \mu_2)}{2nG^2}} \tag{77}$$

These can be accomplished by setting

$$a_t \leq \frac{\sqrt{(A_{t-1}\nu + \nu_1)} \min\{C_{t-1}\mu + \mu_1, c_{t-1}\mu + \mu_2\}}{2\sqrt{n}G}.$$

We must also handle the term $\frac{3nG^2 a_{t-1}^2}{C_{t-1}\mu + \mu_1} \|q_{t-1} - \hat{q}_{t-2}\|_2^2$. By the argument of Lem. 10 using $a_t^2$ instead of $a_t$, we have that when summing over $t$, we have that

$$\sum_{t=1}^{T} \frac{3nG^2 a_t^2}{\mu_t} \|q_t - \hat{q}_{t-1}\|_2^2 \leq \frac{3nG^2}{C_{t-1}\mu + \mu_1} \cdot 3e^4 M^2 \sum_{t=1}^{T} a_t^2 \|q_t - q_{t-1}\|_2^2$$

$$\leq \frac{512nG^2 M^2}{C_{t-1}\mu + \mu_1} \sum_{t=1}^{T} a_t^2 \|q_t - q_{t-1}\|_2^2$$

where $M = n/b$ is the number of blocks, and the $e^4$ term appears from the squared constants (as opposed to $e^2$). Using that we will sum the non-positive terms $\sum_{t=1}^{T} A_{t-1}\nu \mathbb{E}_t[\Delta_D(q_t, q_{t-1},])$, so that the final condition needed to cancel this term is

$$\frac{512nG^2 M^2}{C_{t-1}\mu + \mu_1} a_t^2 \leq \frac{A_{t-1}\nu + \nu_1}{2}$$

which can also be rewritten as

$$a_t \leq \frac{1}{32M} \sqrt{\frac{(A_{t-1}\nu + \nu_1)(C_{t-1}\mu + \mu_1)}{nG^2}}$$

Thus, combining the previous conditions, our final update on the $(a_t)$ sequence is

$$a_t = \min\left\{\frac{C_{t-1}\mu + \mu_1}{12enq_{\max}L}, \left(1 + \frac{b}{n}\right)a_{t-1}, \frac{b}{32n}\frac{\sqrt{(A_{t-1}\nu + \nu_1)}\min\{C_{t-1}\mu + \mu_1, c_{t-1}\mu + \mu_2\}}{\sqrt{n}G}\right\},$$

as alluded to in Appx. C.4.1.

# D   Implementation Details

In this section, we provide additional background on implementing DRAGO in practice. This involves a description of the algorithm amenable for direct translation into code and procedures for computing the dual proximal mapping for common uncertainty sets and penalties. We assume in this section that $\mathcal{W} = \mathbb{R}^d$ and provide multiple options for the uncertainty set $\mathcal{Q}$.

## D.1   Algorithm Description

The full algorithm is given in Algorithm 2. We first describe the notation. Recall that $M = n/b$, or the number of blocks. We partition $[n]$ into $(B_1, \dots, B_M)$, where each $B_K$ denotes a $b$-length list of contiguous indices. For any matrix $u \in \mathbb{R}^{n \times m}$ (including $m = 1$), we denote by $u[B_K] \in \mathbb{R}^{b \times m}$ the rows of $u$ corresponding to the indices in $B_K$. Finally, for a vector $s \in \mathbb{R}^b$, we denote by $se_{B_K}$ the vector that contains $s_k$ in indices $k \in B_K$, and has zeros elsewhere. Next, we comment on particular aspects regarding the implementation version as compared to Algorithm 1 (Sec. 2).

- We store two versions of each table, specifically $\hat{\ell}, \hat{\ell}_1 \in \mathbb{R}^n$, $\hat{g}_1, \hat{g}_2 \in \mathbb{R}^{n \times d}$, and $\hat{q}_1, \hat{q}_2 \in \mathbb{R}^n$. For any iterate $t$, these variables are meant to store $\hat{\ell}_t, \hat{\ell}_{t-1} \in \mathbb{R}^n$, $\hat{g}_{t-1}, \hat{g}_{t-2} \in \mathbb{R}^{n \times d}$, and $\hat{q}_{t-1}, \hat{q}_{t-2} \in \mathbb{R}^n$.

- The quantities $\hat{g}_{\text{agg}} \in \mathbb{R}^d$ and $\hat{w}_{\text{agg}} \in \mathbb{R}^d$ are introduced as to not recompute the sums in the primal update on each iteration (which would cost $O(nd)$ operations). Instead, these aggregates are updated using $O(bd)$ operations.

- The loss and gradient tables are not updated immediately after the primal update. However, the values that fill the tables are computed, and the update occurs at the end of the loop. This is because $\hat{\ell}[B_{K_t}]$ is used to fill $\hat{\ell}_1[B_{K_t}]$ at the end of the loop, we we must maintain knowledge of $\hat{\ell}[B_{K_t}]$ temporarily.

- While the proximal operator is specified for the primal in the case of $\mathcal{W} = \mathbb{R}^d$, the proximal operator for the dual os computed by a subroutine DualProx, which we describe in the next subsection.

## D.2   Solving the Maximization Problem

As discussed in Appx. B, the primary examples of DRO uncertainty sets $\mathcal{Q}$ are balls in $f$-divergence (specifically, KL and $\chi^2$) and spectral risk measure sets. For the penalty $D$, it is also common to use $f$-divergences. We review these concepts in this section and provide recipes for computing the maximization problem.

$f$-**Divergences**   We first recall the definition of $f$-divergences used throughout this section.

**Definition 19.** Let $f : [0, \infty) \mapsto \mathbb{R} \cup \{+\infty\}$ be a convex function such that $f(1) = 0$, $f(x)$ is finite for $x > 0$, and $\lim_{x \to 0^+} f(x) = 0$. Let $q$ and $\bar{q}$ be two probability mass functions defined on $n$ atoms. The $f$-*divergence* from $q$ to $\bar{q}$ generated by this function $f$ is given by

$$D_f(q \| \bar{q}) := \sum_{i=1}^{n} f\left(\frac{q_i}{\bar{q}_i}\right) \bar{q}_i,$$

where we define $0f(0/0) := 0$. For any $i$ such that $\bar{q}_i = 0$ but $q_i > 0$, we define $D_f(q \| \bar{q}) =: +\infty$.

The two running examples we use are the $\chi^2$-divergence generated by $f_{\chi^2}(x) = x^2 - 1$ and the KL divergence generated by $f_{\text{KL}}(x) = x \ln x$ on $(0, \infty)$ and define $0 \ln 0 = 0$. For any convex set $\mathcal{X} \subseteq \mathbb{R}^k$, we also introduce the convex indicator function

$$\iota_{\mathcal{X}}(x) := \begin{cases} 0 & \text{if } x \in \mathcal{X} \\ 1 & \text{otherwise} \end{cases}.$$

In either of the two cases below, we select the penalty $D(q \| \mathbf{1}/n) = D_f(q \| \mathbf{1}/n)$ to be an $f$-divergence. Denote in addition $f^*$ as the Fenchel conjugate of $f$.

---

**Algorithm 2** DRAGO: Implementation Version

---

**Input:** Learning rate parameter $\alpha > 0$, batch size $b \in \{1, \dots, n\}$, number of iterations $T$.

**Initialization:**

$w \leftarrow 0_d$ and $q \leftarrow \mathbf{1}/n$

$\hat{\ell} \leftarrow \ell(w), \hat{\ell}_1 \leftarrow \ell(w), \hat{g}_1 \leftarrow \nabla\ell(w), \hat{g}_2 \leftarrow \nabla\ell(w), \hat{q}_1 \leftarrow q$ and $\hat{q}_2 \leftarrow q$

$\hat{w}_K \leftarrow w$ for $K \in \{1, \dots, M\}$ for $M = n/b$

$\hat{g}_{\mathrm{agg}} \leftarrow \hat{g}_1^\top \hat{q}_1$ and $\hat{w}_{\mathrm{agg}} \leftarrow \sum_{K=1}^M \hat{w}_K$

$\bar{\beta} = 1/[16\alpha(1+\alpha)(n/b - 1)^2]$ if $n/b > 1$ and $0$ otherwise

**for** $t = 1$ **to** $T$ **do**

    Sample blocks $I_t$ and $J_t$ uniformly on $[n/b]$ and compute $K_t = t \mod (n/b) + 1$

    $\beta_t \leftarrow (1 - (1+\alpha)^{1-t})/(\alpha(1+\alpha))$

    **Primal Update:**

    $g \leftarrow [\nabla\ell_i(w)]_{i \in B_{I_t}} \in \mathbb{R}^{b \times d}$ and $v^{\mathrm{P}} \leftarrow \hat{g}_{\mathrm{agg}} + \frac{1}{1+\alpha}\delta^{\mathrm{P}}$.

    $w \leftarrow \frac{1}{(1+\beta_t)}\left((\beta_t - \bar{\beta}(M-1))w + \bar{\beta}(\hat{w}_{\mathrm{agg}} - \hat{w}_{K_t}) - v^{\mathrm{P}}/\mu\right)$

    $\hat{w}_{\mathrm{agg}} \leftarrow \hat{w}_{\mathrm{agg}} + w - \hat{w}_{K_t}$ and $\hat{w}_{K_t} \leftarrow w$

    **Compute Loss and Gradient Table Updates:**

    $(l_t, g_t) \leftarrow [\ell_k(w), \nabla\ell_k(w)]_{k \in B_{K_t}} \in \mathbb{R}^b \times \mathbb{R}^{b \times d}$

    **Dual Update:**

    $l \leftarrow [\ell_j(w)]_{j \in B_{J_t}} \in \mathbb{R}^b$

    $\delta^{\mathrm{D}} \leftarrow M(l - \hat{\ell}_1[J_t])$ and $v^{\mathrm{D}} \leftarrow \hat{\ell} + (l - \hat{\ell}[B_{K_t}])e_{B_{K_t}} + \frac{1}{1+\alpha}\delta^{\mathrm{D}}e_{B_{J_t}}$

    $q \leftarrow \mathrm{DualProx}(q, v^{\mathrm{D}}, \beta_t) = \arg\max_{\bar{q} \in \mathcal{Q}}\left\{\langle v^{\mathrm{D}}, \bar{q}\rangle - \nu D(\bar{q}\|\mathbf{1}_n/n) - \beta_t \nu \Delta_D(\bar{q}, q)\right\}$

    **Update All Tables:**

    $\hat{g}_2[B_{K_t}] \leftarrow \hat{g}_1[B_{K_t}]$ and $\hat{g}_1[B_{K_t}] \leftarrow g_t$

    $\hat{\ell}_1[B_{K_t}] \leftarrow \hat{\ell}[B_{K_t}]$ and $\hat{\ell}[B_{K_t}] \leftarrow l_t$

    $\hat{q}_2[B_{K_t}] \leftarrow \hat{q}_1[B_{K_t}]$ and $\hat{q}_1[B_{K_t}] \leftarrow q[B_{K_t}]$

    $\hat{g}_{\mathrm{agg}} \leftarrow \hat{g}_{\mathrm{agg}} + \hat{g}_1[B_{K_t}]^\top \hat{q}_1[B_{K_t}] - \hat{g}_2[B_{K_t}]^\top \hat{q}_2[B_{K_t}]$

**end for**

**return** $(w, q)$.

---

### D.2.1 Spectral Risk Measure Uncertainty Sets

As in Appx. B, the spectral risk measure uncertainty set is defined by a set of non-decreasing, non-negative weights $\sigma = (\sigma_1, \dots, \sigma_n)$ that sum to one. Our uncertainty set is given by

$$\mathcal{Q} = \mathcal{Q}(\sigma) := \mathrm{conv}\left(\{\text{permutations of } \sigma\}\right),$$

and we use $D_f$ has the penalty for either $f_{\chi^2}$ or $f_{\mathrm{KL}}$. The set $\mathcal{Q}(\sigma)$ is referred to the *permutahedron* on $\sigma$. In this case, the maximization problem can be dualized and solved via the following result.

**Proposition 20.** *[Mehta et al., 2024, Proposition 3] Let $l \in \mathbb{R}^n$ be a vector and $\pi$ be a permutation that sorts its entries in non-decreasing order, i.e., $l_{\pi(1)} \leq \dots \leq l_{\pi(n)}$. Consider a function $f$ strictly convex with strictly convex conjugate defining a divergence $D_f$. Then, the maximization over the permutahedron subject to the shift penalty can be expressed as*

$$\max_{q \in \mathcal{Q}(\sigma)}\left\{q^\top l - \nu D_f(q\|\mathbf{1}_n/n)\right\} = \min_{\substack{z \in \mathbb{R}^n \\ z_1 \leq \dots \leq z_n}} \sum_{i=1}^n g_i(z; l), \tag{78}$$

*where we define $g_i(z; l) := \sigma_i z + \frac{\nu}{n} f^*\left((l_{\pi(i)} - z)/\nu\right)$. The optima of both problems, denoted*

$$z^{\mathrm{opt}}(l) = \arg\min_{\substack{z \in \mathbb{R}^n \\ z_1 \leq \dots \leq z_n}} \sum_{i=1}^n g_i(z; l), \ q^{\mathrm{opt}} = \arg\max_{q \in \mathcal{Q}(\sigma)} q^\top l - \nu D_f(q\|\mathbf{1}_n/n),$$

*are related as $q^{\mathrm{opt}}(l) = \nabla(\nu D_f(\cdot\|\mathbf{1}_n/n))^*(l - z^{\mathrm{opt}}_{\pi^{-1}}(l))$, that is,*

$$q_i^{\mathrm{opt}}(l) = \frac{1}{n}[f^*]'\left(\frac{1}{\nu}(l_i - z^{\mathrm{opt}}_{\pi^{-1}(i)}(l))\right). \tag{79}$$

As described in Mehta et al. [2024, Appendix C], the minimization problem (78) is an exact instance of isotonic regression and can be solved efficiently with the pool adjacent violators (PAV) algorithm.

### D.2.2 Divergence-Ball Uncertainty Sets

Another common uncertainty set format is a ball in $f$-divergence, or

$$\mathcal{Q} = \mathcal{Q}(\rho) := \left\{q \in \mathbb{R}^n : D_f(q\|\mathbf{1}/n) \le \rho, q \ge 0, \text{ and } \mathbf{1}^\top q = 1\right\}.$$

We describe the case of the rescaled $\chi^2$-divergence in particular, in which the feasible set is an $\ell_2$-ball intersected with the probability simplex. Given a vector $l \in \mathbb{R}^n$, we aim to compute the mapping

$$l \mapsto \underset{\substack{q \in \mathcal{P}_n \\ \frac{1}{2}\|q - \mathbf{1}/n\|_2^2 \le \rho}}{\arg\max} \quad \langle l, q \rangle - \frac{\nu}{2}\|q - \mathbf{1}/n\|_2^2, \tag{80}$$

where $\mathcal{P}_n := \left\{q \in \mathbb{R}^n : q \ge 0, \mathbf{1}^\top q = 1\right\}$ denotes the $n$-dimensional probability simplex. We apply a similar approach to Namkoong and Duchi [2017], in which we take a partial dual of the problem above. Indeed, note first that for any $q \in \mathcal{P}_n$, we have that $\frac{1}{2}\|q - \mathbf{1}_n/n\|_2^2 = \frac{1}{2}\|q\|_2^2 - \frac{1}{2n}$. Thus, the optimal solution to (80) can be computed by solving

$$\max_{q \in \mathcal{P}_n} \min_{\lambda \ge 0} \langle l, q \rangle - \frac{\nu}{2}\|q\|_2^2 - \lambda\left(\frac{1}{2}\|q\|_2^2 - \rho - \frac{1}{2n}\right),$$

or equivalently, by strong duality via Slater's condition, solving

$$\max_{\lambda \ge 0}\left[f(\lambda) := (\nu + \lambda)\min_{q \in \mathcal{P}_n}\frac{1}{2}\|q - l/(\nu + \lambda)\|_2^2 - \lambda\left(\rho + \frac{1}{2n}\right) - \frac{1}{2(\nu + \lambda)}\|l\|_2^2\right].$$

Notice that evaluation of the outer objective itself requires Euclidean projection onto the probability simplex as a subroutine, after which the maximization problem can be computed via the bisection method, as it is a univariate concave maximization problem over a convex set. In order to determine which half to remove in the bisection search, we also compute the derivative of $\lambda \mapsto f(\lambda)$, which is given by

$$f'(\lambda) := \frac{1}{2}\left\|q^{\mathrm{opt}}(\lambda)\right\|_2^2 - \rho - \frac{1}{2n},$$

where $q^{\mathrm{opt}}(\lambda)$ achieves the minimum in $\min_{q \in \mathcal{P}_n}\frac{1}{2}\|q - l/(\nu + \lambda)\|_2^2$ for a fixed $\lambda \ge 0$. For projection onto the probability simplex, we apply Algorithm 1 from Condat [2016], which is a solution relying on sorting the projected vector. The overall method consists of three steps.

1. **Sorting:** Projection onto the simplex relies on sorting the vector $l/(\nu + \lambda)$ on each evaluation. However, because $l/(\nu + \lambda)$ varies from evaluation to evaluation simply by multiplying by a positive scalar, we may pre-sort $l$ and use the same sorted indices on each evaluation of $(f(\lambda), f'(\lambda))$ listed below.

2. **Two-Pointer Search:** We find the upper and lower limits for $\lambda$ by initializing $\lambda_{\min} = 0$ and $\lambda_{\max} = 1$, and repeatedly making the replacement $(\lambda_{\min}, \lambda_{\max}) \leftarrow (\lambda_{\max}, 2\lambda_{\max})$ until $f'(\lambda_{\max}) < -\varepsilon$ for some tolerance $\varepsilon > 0$. This, along with $f'(\lambda_{\min}) > \varepsilon$ indicates that the optimal value of $\lambda$ lies within $(\lambda_{\min}, \lambda_{\max})$. For any $\lambda$ with $|f'(\lambda)| < \varepsilon$, we return the associated $q^{\mathrm{opt}}(\lambda)$ as the solution.

3. **Binary Search:** Finally, we repeatedly evaluate $f'(\lambda)$ for $\lambda = (\lambda_{\min} + \lambda_{\max})/2$. If $f'(\lambda) > \varepsilon$, we set $\lambda_{\min} \leftarrow \lambda$, whereas if $f'(\lambda) < -\varepsilon$, then we set $\lambda_{\max} \leftarrow \lambda$. We terminate when $\lambda_{\max} - \lambda_{\min} < \varepsilon$ or $|f'(\lambda)| < \varepsilon$.

The parameter $\varepsilon$ is set to $10^{-10}$ in our experiments. Note that the same procedure can be used to compute the dual proximal operator in Algorithm 1. In particular, when $\Delta_D((,q), q_{t-1}) = \frac{1}{2} \|q - q_t\|_2^2$, which is true when $D$ is the $\chi^2$-divergence, then

$$q_t = \underset{\substack{q \in \mathcal{P}_n \\ \frac{1}{2}\|q-\mathbf{1}/n\|_2^2 \le \frac{\rho}{n}}}{\arg\max} \ \langle l + \nu\beta_t q_t, q \rangle - \frac{\nu(1+\beta_t)}{2} \|q - \mathbf{1}/n\|_2^2,$$

which is a particular case of (80), and hence can be solved using the exact same procedure. The runtime of this subroutine is $O(n \log n + n \log(1/\varepsilon))$, accounting for both the initial sorting at $O(n \log n)$ cost, and the $O(\log(1/\varepsilon))$ iterations of the exponential and binary searches. Each iteration requires a linear scan of $n$ elements at cost $O(n)$.

**Hardware Acceleration** Finally, note that the computations in Appx. D.2 and Appx. D.2.2 involve primitives such as sorting, linear scanning through vectors, and binary search. Due to their serial nature (as opposed to algorithms that rely on highly parallelizable operations such as matrix multiplication), we also utilize just-in-time compilation on the CPU via the Numba package for increased efficiency.

| Dataset | $d$ | $n$ | Task | Source |
|---------|-----|-----|------|--------|
| yacht | 6 | 244 | Regression | UCI |
| energy | 8 | 614 | Regression | UCI |
| concrete | 8 | 824 | Regression | UCI |
| kin8nm | 8 | 6,553 | Regression | OpenML |
| power | 4 | 7,654 | Regression | UCI |
| acsincome | 202 | 4,000 | Regression | Fairlearn |
| emotion | 270 | 8,000 | Multiclass Classification | Hugging Face |

Table 5: Dataset attributes such as sample size $n$, parameter dimension $d$, and sources.

## E    Experimental Details

We describe details of the experimental setup, including datasets, compute environment, and hyper-paramater tuning. We largely maintain the benchmarks of Mehta et al. [2023].

### E.1    Datasets

The sample sizes, dimensions, and source of the datasets are summarized in Tab. 5. The tasks associated with each dataset are listed below.

(a) *yacht*: predicting the residuary resistance of a sailing yacht based on its physical attributes Tsanas and Xifara [2012].
(b) *energy*: predicting the cooling load of a building based on its physical attributes Baressi Segota et al. [2020].
(c) *concrete*: predicting the compressive strength of a concrete type based on its physical and chemical attributes Yeh [2006].
(d) *kin8nm*: predicting the distance of an 8 link all-revolute robot arm to a spatial endpoint [Akujuobi and Zhang, 2017].
(e) *power*: predicting net hourly electrical energy output of a power plant given environmental factors [Tüfekci, 2014].
(f) *acsincome*: predicting income of US adults given features compiled from the American Community Survey (ACS) Public Use Microdata Sample (PUMS) [Ding et al., 2021].
(g) *emotion*: predicting the sentiment of sentence in the form of six emotions. Each input is a segment of text and we use a BERT neural network Devlin et al. [2019] as an initial feature map. This representation is fine-tuned using 2 epochs on a random half (8,000 examples) of the original emotion dataset, and then applied to the remaining half. We then apply principle components analysis (PCA) to reduce the dimension of each vector to 45.

### E.2    Hyperparameter Selection

We fix a minibatch size of 64 SGD and an epoch length of $N = n$ for LSVRG. In practice, the regularization parameter $\mu$ and shift cost $\nu$ are tuned by a statistical metric, i.e. generalization error as measured on a validation set.

For the tuned hyperparameters, we use the following method. Let $k \in \{1, \dots, K\}$ be a seed that determines algorithmic randomness. This corresponds to sampling a minibatch without replacement for SGD and SRDA and a single sampled index for LSVRG. Letting $\mathcal{L}_k(\eta)$ denote the average value of the training loss of the last ten passes using learning rate $\eta$ and seed $k$, the quantity $\mathcal{L}(\eta) = \frac{1}{K}\sum_{k=1}^{K}\mathcal{L}_k(\eta)$ was minimized to select $\eta$. The learning rate $\eta$ is chosen in the set $\{1 \times 10^{-4}, 3 \times 10^{-4}, 1 \times 10^{-3}, 3 \times 10^{-3}, 1 \times 10^{-2}, 3 \times 10^{-2}, 1 \times 10^{-1}, 3 \times 10^{-1}, 1 \times 10^{0}, 3 \times 10^{0}\}$, with two orders of magnitude lower numbers used in `acsincome` due to its sparsity. We discard any learning rates that cause the optimizer to diverge for any seed.

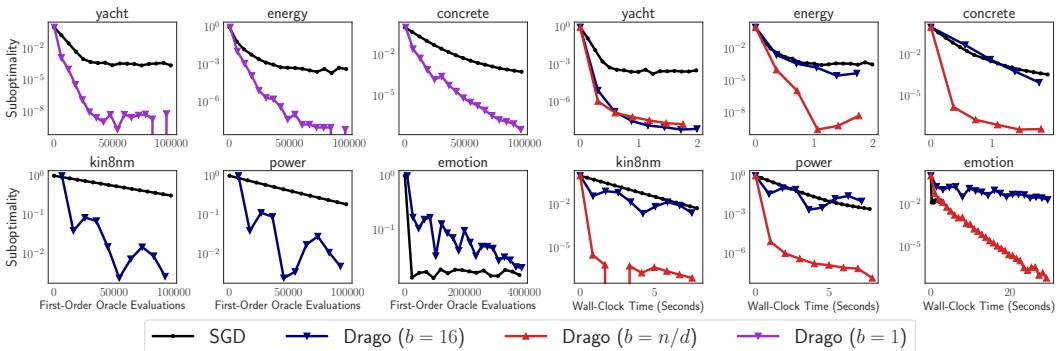

Figure 4: **Benchmarks on the $\chi^2$ Uncertainty Set.** In both panels, the $y$-axis measure the primal suboptimality gap, defined in (15). Individual plots correspond to particular datasets. **Left:** The $x$-axis displays the number of individual first-order oracle queries to $\{(\ell_i, \nabla \ell_i)\}_{i=1}^n$. **Right:** The $x$-axis displays wall-clock time.

## E.3 Compute Environment

Experiments were run on a CPU workstation with an Intel i9 processor, a clock speed of 2.80GHz, 32 virtual cores, and 126G of memory. The code used in this project was written in Python 3 using the Numba packages for just-in-time compilation. Run-time experiments were conducted without CPU parallelism. The algorithms are primarily written in PyTorch and support automatic differentiation.

## E.4 Additional Experiments

We explore the sensitivity of the results to alterations of the objective and algorithm hyperparameters.

**Sensitivity to Uncertainty Set Choice** In Sec. 4, we mainly show performance on spectral risk-based uncertainty sets, in particular the conditional value-at-risk (CVaR). In this section, we also consider $f$-divergence ball-based uncertainty sets, with the procedure described in Appx. D.2.2. As in Namkoong and Duchi [2017], we use a radius that is inversely proportional to the sample size, namely $\rho = \frac{1}{n}$, and the strong convexity-strong concavity parameter $\mu = \nu = 1$. In Fig. 4, we demonstrate the performance of DRAGO with $b = 1$, $b = 16$ (as chosen heuristically), and $b = n/d$. We compare against the biased stochastic gradient descent, which can be defined using oracle to compute the optimal dual variables given a vector of losses; however, note that LSVRG is designed only for spectral risk measures, so the method does not apply in the divergence ball setting. We observe that the optimization performance across both regression and multi-class classification tasks are qualitatively similar to that seen in Fig. 2 nad Fig. 3. The $b = 1$ variant performs well on smaller datasets ($n \leq 1,000$), whereas the $b = 16$ heuristic generally does not dominate in terms of gradient evaluations or wall time. While the number of gradient evaluations is significantly larger for the $b = n/d$ variant, implementation techniques such as just-in-time complication (see Appx. D) allow for efficient computation, resulting in better overall optimization performance as a function of wall time.

**Sensitivity to Batch Size** In Fig. 5, we consider the datasets with the largest ratio of $n$ to $d$ (hence the largest theoretically prescribed batch size) and assess the performance of DRAGO with smaller batch sizes. For both datasets, we have that in magnitude, $n/d \approx 1000$. Intuitively, the smaller batch size methods would perform better in terms of oracle queries but the large batch methods would be more performant in terms of wall time. With only a batch size of $b = 64$, this variant of DRAGO generally matches the best-performing setting when viewed from either oracle calls or direct wall time. This is approximately $16\times$ smaller than the $n/d$ benchmark, indicating that tuning the batch size can significantly reduce the memory overhead of the algorithm while increasing speed.

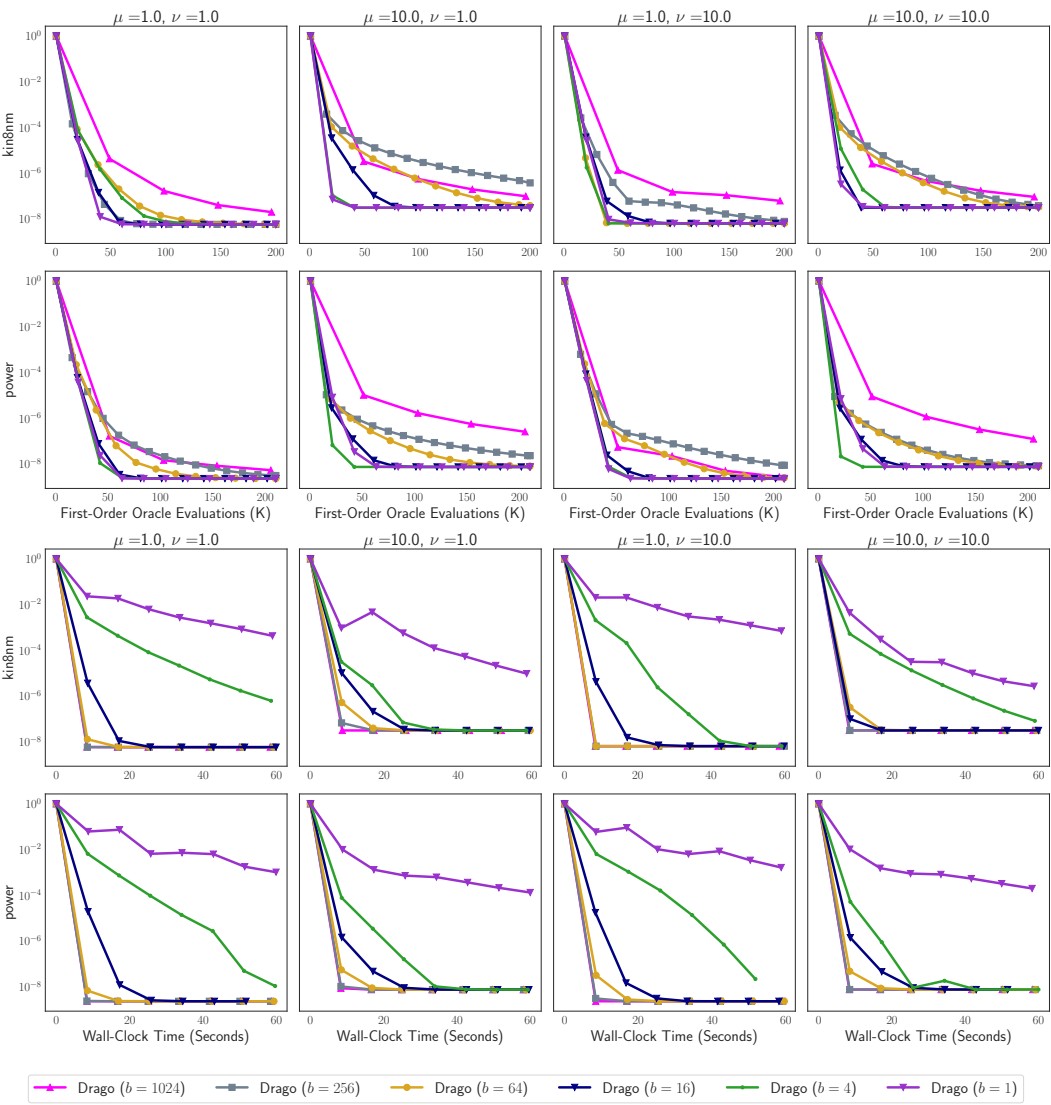

Figure 5: **DRAGO on varying batch sizes and strong convexity parameters.** Each row indicates a dataset, where as each column denotes the CVaR objective with the given regularization parameters. **Top Rows:** The $x$-axis displays the number of individual first-order oracle queries to $\{(\ell_i, \nabla \ell_i)\}_{i=1}^n$. **Bottom Rows:** The $x$-axis displays wall-clock time.

