# OpenReview forum: "Drago: Primal-Dual Coupled Variance Reduction for Faster Distributionally Robust Optimization"
_NeurIPS.cc/2024/Conference — NeurIPS 2024 poster_

### Official Review · Reviewer_WLa8 · 2024-07-14

**Soundness:** 3
**Presentation:** 3
**Contribution:** 2
**Rating:** 6
**Confidence:** 3

**Summary:**

To tackle the optimization problem where the objective is a weighted sum of losses from multiple domains, an improved optimization algorithm is designed by incorporating past history and appropriate regularization. Merits of the proposed algorithm are demonstrated by a rigorous theoretical analysis of convergence rate  and numerical analysis by synthetic and text classification data.

**Strengths:**

Theoretical analysis seems solid.

**Weaknesses:**

The proposed method seems to be only relevant for problems with a low dimensionality $d$. Otherwise, the fact that the algorithm requires to keep track of historical weights would cause a high space complexity.

**Questions:**

1. Regarding the remark on global complexity on page 7, I do not follow where the $bd$ term comes from in the per iteration complexity. Is it possible to provide some details? My understanding may be inaccurate, but if each elementwise update of $v_t^D$ is of low computational cost, and we update all elements in a distributed fashion, can we improve the per iteration cost to be $O(n)$?
2. The dimensionality considered in the empirical analysis is pretty low. Can additional empirical analysis be shown where $d$ is, e.g. of a similar magnitude as $n$, if not greater than $n$? I understand that there may be a challenge in updating $q$.

**Limitations:**

No direct societal impact as the paper is focused on theoretical aspects of the out of distribution inference problem.

---

> ### Author Rebuttal · Authors · 2024-08-06
>
> Thank you for your concrete comments. We address them below.
>
> >**I do not follow where the $bd$ term comes from in the per iteration complexity. Is it possible to provide some details?**
>
> The quantity comes from the fact that $b$ evaluations of $(\ell_i, \nabla \ell_i)$ in the computation of $v^{P}_t$ (not from $v^{D}_t$), and we assume each evaluation costs $O(d)$.
>
> >**The proposed method seems to be only relevant for problems with a low dimensionality 𝑑. Otherwise, the fact that the algorithm requires to keep track of historical weights would cause a high space complexity.**
>
> Thank you for identifying this point. We have updated the manuscript to include a discussion on space complexity, which in many cases can be reduced far below $O(nd)$ for two reasons. Firstly, this is a known limitation for SAG/SAGA-type variance reduction methods for empirical risk minimization. That being said, the additional storage cost (beyond the training data) is $O(n)$ in many common cases. This is because convex losses in machine learning can typically be expressed as $\ell_i(w) = h(y_i, x_i^\top w)$ for some differentiable error function $h$ and so the gradient is given by the scalar multiplication $h’(y_i, x_i^\top w) \cdot x_i$, where $h’$ is the derivative with respect to the second argument. Thus, only these scalar derivatives need to be stored.
>
> Secondly, a more subtle advantage of our method is that we use cyclic updates instead of randomized updates for the table of past gradients (as opposed to the randomized updates in SAGA). Conceptually, this means that for very large-scale workloads, a fraction of the gradient table can be paged out of memory until its values are updated again.
>
> >**The dimensionality considered in the empirical analysis is pretty low. Can additional empirical analysis be shown where $d$ is, e.g. of a similar magnitude as $n$, if not greater than $n$?**
>
> Thank you for raising this point. We scaled up the text classification task to $d = n/2 \approx 4000$ in the attached PDF (see Figure 5). This means that the batch size is approximately $2$. Qualitatively, similar trends are observed across varying orders of magnitude for $d$, in that we show improvement over LSVRG baseline in the attached PDF. We hypothesize that the benefit over LSVRG comes from the fact that the dual variable is updated in epochs of length $n$ iterations to balance out the $O(n)$ cost of the update. We incur a higher per-iteration cost but have improved theoretical and empirical performance over the training trajectory.

---

> ### Comment · Reviewer_WLa8 · 2024-08-11
> **Thank you for answering my questions**
>
> Given that my questions have been satisfactorily addressed, I'd like to raise my score.

---

### Official Review · Reviewer_ccWo · 2024-07-15

**Soundness:** 3
**Presentation:** 3
**Contribution:** 3
**Rating:** 5
**Confidence:** 2

**Summary:**

The authors provide a state-of-the-art optimization algorithm with convergence guarantees for f-divergence DRO. The numerical study is also extensive

**Strengths:**

1. Techincal results are strong and to the best of my knowledge, I haven't find mistakes
2. Numerical study is extensive

**Weaknesses:**

As the author has commented, they only solve for convex optimization formulation. The extension to non-convex setup also seems interesting and important.

**Questions:**

N/A

---

> ### Author Rebuttal · Authors · 2024-08-06
>
> Thank you for your comments. We address the main concern below.
>
> >**As the author has commented, they only solve for convex optimization formulation. The extension to non-convex setup also seems interesting and important.**
>
> Thank you for raising this point. Broadly, the study of optimization approaches for the non-convex setting (e.g., deep learning) relies on completely different technical tools and evaluations. Convexity is considered to make precise statements about global optimality, namely the linear convergence of the algorithm’s iterates to a solution in terms of problem constants. In the non-convex setting, one seeks convergence to a first-order stationary point under a gradient-domination or Polyak-Łojasiewicz condition or for large deep learning models with Gaussian random weights or other strong assumptions. Our focus here is on sound theoretical convergence results, for convex optimization algorithms, one of the subject areas of NeurIPS.

---

> > ### Comment · Reviewer_ccWo · 2024-08-13
> >
> > I thank the authors for answering my questions. I will keep my score.

---

### Official Review · Reviewer_VCV6 · 2024-07-16

**Soundness:** 3
**Presentation:** 2
**Contribution:** 3
**Rating:** 7
**Confidence:** 3

**Summary:**

This paper proposes DRAGO, a variance-reduced optimization method for
distributionally robust optimization (DRO). In particular, the authors consider
the penalized DRO framework and view DRO as min-max optimization with a simple
finite-sum structure. Their algorithm maintains explicit histories for the
individual loss functions $l_i$, loss function gradients $\nabla l_i$, and dual
parameters $q_i$ (i.e. like SAG/SAGA); these estimates are used to control the
variance in both the primal and dual updates, which the authors prove leads to
a fast linear rate of convergence.  Unlike existing methods, DRAGO supports any
penalty parameter $\nu \geq 0$ and its convergence rate on depends explicitly
on the different problem constants.

**Strengths:**

- DRAGO obtains a fast linear rate and the analysis shows its dependence on
    batch size, dataset size, and the other problem constants.

- DRAGO performs well in practice, particularly when the batch-size is tuned.

- The analysis permits any non-negative penalization strength $\nu$ for the
    dual (i.e. uncertainty) parameters.

**Weaknesses:**

- The paper organization makes it quite difficult to understand the different
    algorithmic components used in DRAGO.

- The assumptions necessary for convergence are somewhat unrealistic, although
    these appear to be standard in the literature.

- The experimental evaluation is small and limited to relatively low-dimensional
    problems.

### Detailed Comments

This is an interesting paper which tries to speed up DRO by combining variance reduction with proximal gradient updates for a structured min-max problem. While I'm not an expert in DRO, the paper seems to make a solid contribution, particularly by relaxing previous requirements on the dual regularization strength $\nu$. I also appreciate that the convergence rate shows the role of each problem constant and tuning parameter, including batch-size. Please note that I did not check the appendix for correctness.

**Paper Organization**:
I found this paper quite confusing to read. Instead of introducing DRAGO and
then explaining it's various algorithmic components, the authors first try to
explain the intuition behind the updates. This is a good idea in theory, but in
practice there are too many moving parts to DRAGO and I quickly got lost in the
(mostly unimportant) details. The actually key parts of DRAGO --- proximal-gradient
updates with SAG/SAGA style variance reduction --- are barely mentioned and this
makes the algorithm much harder to understand.

My suggestion is to introduce Algorithm 1 first and then explain why
additional proximal terms are needed in the proximal-gradient update in order
to control the variance. At least this way the reader has a reference point
for understanding the discussion.

**Assumptions**:
I think that assuming each $l_i$ is both Lipschitz continuous and Lipschitz
smooth is quite unrealistic. Typically only one or the other of these
conditions holds. For instance, Lipschitz continuity implies that $l_i$ is
bounded above by a linear function, while Lipschitz smoothness implies that
$l_i$ is bounded above by a quadratic. While both can be true, the quadratic
bound will generally be vacuous. However, I understand from Table 1 that these
assumptions are typical for the literature, so I don't think this is too much
of an issue.

**Experimental Evaluation**:
I would have liked to see experiments on a higher dimensional dataset.  Since
the entire Jacobian $\nabla l(w)$ must be maintained in the history, SAG-type
methods usually have high memory requirements as $d$ increases. The authors
partially handle this by using a large batch-size, but this is only feasible if
the mini-batches can still be fit on the GPU. When $d$ is sufficiently large,
mini-batch gradients will be inefficient to compute and I am curious if the
relative performance of DRAGO and LSVRG ill switch. This is particularly
worth considering because it is only in the large-batch setting that DRAGO
seems to outperform LSVRG

### Minor Comments

- Line 52: This is a super-linear complexity, but a sub-linear convergence rate.
    Contrast with Line 62, which shows a linear rate, but a logarithmic complexity.

- Line 65: Should this read "but do not"?

- Line 24/71: It seems like you are only considering $\nu$ in the probability simplex.
    It would be nice if you stated this explicitly somewhere.

- Line 80: It's generally unrealistic for both $l_i$ and $\nabla l_i$ to be
    Lipschitz continuous over an unbounded set.

- Line 86: You should mention that strong convexity comes only from the regularizer,
    as otherwise it seems to conflict with Lipschitzness of $l_i$.

- Line 102: Does $\nabla l(w_{t-1})$ denote the Jacobian of the vector-valued map
    $l$ such that $[l(w)]_i = l_i(w)$? It would be nice if you clearly defined
    this notation.

- Table 1: The font-size is much too small to read when printed.

- Algorithm Description: The purposes of $\bar{\beta}$ and $\beta_t$ and their
    settings are not well described.

- Display after Line 164: You seem to be using $D(q \| 1/n)$ is
    1-strongly-convex for this result, but this assumption is only introduced
    in Assumption 1. Maybe move Assumption 1 to the start of the discussion
    of convergence to fix this?

- Figure 3: The font-size is much too small to be readable without zooming.

**Questions:**

- Line 491: Isn't $n q_{\text{max}} \leq n$ trivially satisfied since $q$
    is constrained to the probability simplex?

- Figure 2: Why is $b = 1$ only show for three datasets and only in
    terms of oracle complexity?

- Display after Line 124: It looks you are using the equality,

$$\\mathbb{E}[(l_{j_t}(w_{t+1}) - \hat l_{t, j_t}) (l_{j_t}(w_t) - \hat l_{t-1, j_t})] = \\mathbb{E}[(l(w_{t+1}) - \hat l_{t}) (l_{j_t}(w_t) - \hat l_{t-1, j_t})]$$

but I don't think this is true because the terms in the product are correlated.
Can you please explain this?

**Limitations:**

Limitations appear to be appropriate addressed in the paper.

---

> ### Author Rebuttal · Authors · 2024-08-06
>
> Thank you for your insightful comments. We address them below.
>
> >**SAG-type methods usually have high memory requirements as $d$ increases.**
>
> Thank you for identifying this important point. We have updated the manuscript to include a discussion on space complexity, which in many cases can be reduced far below $O(nd)$ for two reasons. First, this is a known limitation for SAG/SAGA-type variance reduction methods for empirical risk minimization. That being said, the additional storage cost (beyond the training data) is $O(n)$ in many common cases. This is because convex losses in machine learning can typically be expressed as $\ell_i(w) = h(y_i, x_i^\top w)$ for some differentiable error function $h$ and so the gradient is given by the scalar multiplication $h’(y_i, x_i^\top w) \cdot x_i$, where $h’$ is the derivative with respect to the second argument. Thus, only these scalar derivatives need to be stored. Second, there is a more subtle difference between our method and SAG-type methods. We use cyclic updates instead of randomized updates for the table of past gradients (as opposed to the randomized updates in SAGA). Conceptually, this means that for very large-scale workloads, a fraction of the gradient table can be paged out of memory until its values are updated again.
>
> >**The experimental evaluation is small and limited to relatively low-dimensional problems.**
>
> To illustrate performance, we scaled up the text classification task to $d = n/2 \approx 4000$ and show the improvement over LSVRG in the attached PDF (see Figure 5). This means that the batch size is approximately $2$. We hypothesize that the benefit over LSVRG comes from the fact that the dual variable is updated in epochs of length $n$ iterations to balance out the $O(n)$ cost of the update. We incur a higher per-iteration cost but have improved theoretical and empirical performance over the training trajectory.
>
> >**The paper organization makes it quite difficult to understand the different algorithmic components used in Drago.**
>
> Thank you for the helpful suggestions on paper organization. In the revised version, we have introduced the algorithm early on and described it using the reference points you have mentioned. We have dedicated more of the main text toward the comparisons noted in Appendix B.
>
> >**The assumptions necessary for convergence are somewhat unrealistic, although these appear to be standard in the literature. I think that assuming each $\ell_i$ is both Lipschitz continuous and Lipschitz smooth is quite unrealistic. Typically only one or the other of these conditions holds.**
>
> Because the losses exist within the statistical learning context, assuming both Lipschitz continuity and smoothness is valid in common scenarios. A canonical loss in robust statistics is the Huber loss, which is both Lipschitz and smooth. Similarly, any smooth loss will be Lipschitz continuous if $\mathcal{W}$ is compact. Compactness is a common assumption, as statistical guarantees (uniform convergence, minimax rate optimality, etc.) rely on it.
>
> >**Isn't $nq_{\max{}} \leq n$ trivially satisfied since 𝑞 is constrained to the probability simplex?**
>
> We have added to Appendix B a precise claim regarding the fact that $n q_{\max{}}$ is constant. In summary, we mean that $nq_{\max{}}$ is bounded by a constant independent of $n$. The two canonical examples of $\mathcal{Q}$ are divergence-ball based and spectral risk measure-based feasible sets. It is true that $\mathcal{Q}$ is always a subset of the probability simplex so $q_{\max{}} \leq 1$. However, this bound is only tight when  $\mathcal{Q}$ is the entire probability simplex, regardless of the choice of $n$. Because this feasible set is often defined using statistical properties of the weights $q$, each element can be upper bounded as $q_i \leq c/n$ for some $c \geq 1$. For spectral risk measures, this can be seen by the argument in Section 2 of [Mehta (2023)](https://proceedings.mlr.press/v206/mehta23b.html) - every weight will be the integral of a bounded function over an interval of length $1/n$. For the $\theta$-CVAR (see our Appendix B.2$, this bound is $1/(\theta n)$.  For divergence balls, in [Namkoong and Duchi (2017)](https://papers.nips.cc/paper_files/paper/2017/hash/5a142a55461d5fef016acfb927fee0bd-Abstract.html) Eq (4), the divergence ball ambiguity set is defined using radius $\rho/n$, which enforces the condition $\frac{n}{2} \Vert q - \mathbf{1}/n \Vert _2^2 \leq \frac{\rho}{n}$ on every element of $\mathcal{Q}$ in the case of $\chi^2$-divergence.
>
> >**Why is $b=1$ only show for three datasets and only in terms of oracle complexity?**
>
> As mentioned toward the end of section 4.1, the $b=1$ is only competitive with baselines for small datasets. As the dataset size grows, the batch size needs to be increased to maintain performance. While the theoretically optimal batch size $n/d$ is a good default value, $b$ can be tuned experimentally to achieve the best performance (with respect to $b$) for a particular empirical setting.
>
> >**Display after Line 124: It looks you are using the equality ... but I don't think this is true because the terms in the product are correlated.**
>
> This equality holds because the expression in the expectation is the inner product between two vectors $\ell(w\_{t+1}) - \hat{\ell}\_t$ and a “one-hot” vector $(\ell\_{t, j\_t} - \hat{\ell}\_{t-1, j\_t})e\_{j\_t}$ (see Eq (8)). Thus, the only terms that are multiplied are in the $j_t$ coordinate.

---

> > ### Comment · Reviewer_VCV6 · 2024-08-08
> >
> > > SAG-type methods usually have high memory requirements as $d$ increases.
> >
> > Yes, I'm very familiar with the least-squares-type arguments for reducing the memory requirements for SAG. Using cyclic updates makes your method somewhat more similar to the incremental aggregated gradient method (IAG) [1]. This algorithm is the conceptual predecessor of SAG, so it's probably worth referencing in your paper.
> >
> > > The experimental evaluation is small and limited to relatively low-dimensional problems.
> >
> > Thanks for including these new experiments. It's nice to see that Drago works well when the batch-size is quite small for at least one problem, but I think I was talking more about the regression experiments in Figure 2. Here, Drago only seems to work well in the large-batch setting (which your comment about $b = 1$ seems to confirm). Thus, I was hoping to see an ablation study on the role of $b$ in the UCI experiments.
> >
> > > The paper organization makes it quite difficult to understand the different algorithmic components used in Drago.
> >
> > Sounds good.
> >
> > > The assumptions necessary for convergence are somewhat unrealistic, although these appear to be standard in the literature. I think that assuming each $\ell_i$ is both Lipschitz continuous and Lipschitz smooth is quite unrealistic. Typically only one or the other of these conditions holds.
> >
> > I don't like compactness assumptions for general machine learning problems, but this is worth adding as a comment to the paper. The Huber loss example is nice, but not particularly used in modern ML, although maybe it remains popular in robust statistics --- I'm not sure.
> >
> > Thanks for the clarifications on the remaining points.
> >
> > [1] Blatt, Doron, Alfred O. Hero, and Hillel Gauchman. "A convergent incremental gradient method with a constant step size." SIAM Journal on Optimization 18.1 (2007): 29-51.

---

> > > ### Author Response · Authors · 2024-08-09
> > > **Response to Reviewer VCV6**
> > >
> > > Thank you for these comments and your thorough review. Please let us know if we have addressed them fully.
> > >
> > > >**Using cyclic updates makes your method somewhat more similar to the incremental aggregated gradient method (IAG) [1].**
> > >
> > > Thank you for this highly relevant reference. We have updated the discussion of related work to include it and give context to our method.
> > >
> > > >**I don't like compactness assumptions for general machine learning problems, but this is worth adding as a comment to the paper.**
> > >
> > > We have added discussion on cases in which Lipschitzness and smoothness of each $\ell_i$ are valid to be clear about the limitations of our assumptions.
> > >
> > > >**I was hoping to see an ablation study on the role of $b=1$ in the UCI experiments.**
> > >
> > > We may have misunderstood your original comment on experimental evaluations when pursuing the "large $d$" experiments&mdash;thank you for the clarification. When running $b=1$ on the UCI datasets, we see that it is only competitive with baselines and other Drago variants when looking at suboptimality against gradient evaluations for the small $n$ datasets (yacht, energy, concrete), which is shown in Figure 2 of the manuscript. In terms of suboptimalty against wall time, the $b=1$ variant generally performs worse than other Drago variants and LSVRG. We collect some of these results in the table below for concreteness. The first table is from yacht, which is generally representative of the three smaller UCI datasets. The second table is from kin8nm, which is generally representative of the larger UCI datasets and ascincome (which is sourced from Fairlearn).
> > >
> > > **Performance against Wall Time (yacht)**
> > > |  Algorithm | Suboptimality after 2 seconds |
> > > | ------- | ------- |
> > > | LSVRG  | $10^{-8.30}$   |
> > > | Drago ($b=1$) | $10^{-4.13}$   |
> > > | Drago ($b=16$)    | $10^{-7.96}$   |
> > > | Drago ($b=n/d$)    | $10^{-8.21}$   |
> > >
> > > **Performance against Wall Time (kin8nm)**
> > > |  Algorithm | Suboptimality after 5 seconds |
> > > | ------- | ------- |
> > > | LSVRG  | $10^{-8.23}$   |
> > > | Drago ($b=1$) | $10^{-1.66}$    |
> > > | Drago ($b=16$)    | $10^{-4.97}$    |
> > > | Drago ($b=n/d$)    | $10^{-8.23}$    |
> > >
> > > We hypothesize that the key measure of problem size is $n/d$ (which happens to be the theoretically optimal batch size). When both $n$ and $d$ are large (as seen in the experiments on emotion in the supplemental PDF), a small batch size ($b \approx 2$) can perform quite well. When $d$ is small relative to $n$, which is generally true of the UCI datasets, large batch sizes perform drastically better. It is worth mentioning that because $d$ is small in these scenarios, it is usually not computationally prohibitive to use larger batch sizes.
> > >
> > > Thank you again and please let us know if there are more questions we can answer.

---

> > > > ### Comment · Reviewer_VCV6 · 2024-08-12
> > > >
> > > > Thanks for providing these additional experimental details. It seems clear that the batch-size is a critical tuning parameter for Drago and I think including a larger-scale ablation study would definitely strengthen the submission. At the very least, an enhanced discussion of the batch-size (similar to what we've had here) should be included in experimental section of the paper.
> > > >
> > > > Overall, I think this paper is a solid submission that should be accepted. I will maintain my score for now and reevaluate during the reviewer discussion period if necessary.

---

> > > > > ### Author Response · Authors · 2024-08-12
> > > > > **Response to Reviewer VCV6**
> > > > >
> > > > > Thank you once again for your review and timely replies! We will reflect this discussion in the final version and include the associated experiments and figures for varying batch size.

---

> ### Author Response · Authors · 2024-08-06
> **Response to Reviewer VCV6 (continued)**
>
> >**Minor Comments: ...**
>
> Thank you for the additional “Minor Comments”. We have incorporated them into the manuscript. To address the questions:
> - Line 102: This is the full Jacobian - thank you for the suggestion.
> - Algorithm Description: These constants are defined by simplifying the algorithm description used in the analysis. We have made the algorithm description consistent with the analysis so that these constants can be avoided.

---

### Official Review · Reviewer_em1q · 2024-07-29

**Soundness:** 4
**Presentation:** 3
**Contribution:** 3
**Rating:** 8
**Confidence:** 3

**Summary:**

The paper considers the DRO problem with a dual penalization term and primal regularization term. Their core contribution is a new primal-dual algorithm for this problem:
- It achieves a new best first-order oracle complexity in certain parameter regimes. In particular, it beats previous algorithms when the dual penalization parameter $\nu$ is small.
- It achieves fine-grained dependence on the primal and dual condition numbers (i.e., not having a max over min term).
- The uncertainty set $\mathcal{Q}$ can be any closed and convex subset of the simplex. This is in contrast to previous works which specialize to more structured uncertainty sets.
- The algorithm adapts ideas in a novel way - in particular, they show that coordinate-style updates can be applied even though the dual feasible set is non-separable. This might lead to applications in other minimax problems. It also only has a single hyperparameter $\eta$ to tune, and comes with a batch-size parameter $b$ which can be used to trade-off between iteration count and per-iteration complexity.
- Strong experimental results are shown (albeit, I should say that while they look very encouraging, I don't have the empirical background necessary to evaluate the methods used/experimental setup carefully).

**Strengths:**

I think this is a strong contribution to the DRO literature. The algorithm combines several ideas (variance reduction, minibatching, and cyclic coordinate-style updates) in a careful way. Of these, the application of cyclic coordinate-style updates to this setting (with a non-separable dual feasible set) seems the most novel and interesting to me. It is also very nice that it handles general uncertainty sets unlike previous work. The presentation is generally solid, although I think it could be improved in a few ways (see below).

**Weaknesses:**

Regarding presentation:
1. I like Table 1 a lot, but I wish the comparison between your first-order complexity bound and previous complexity bounds was a bit more "in your face" and expansive. Currently, the comparisons are kind of spread out throughout the paper, requiring the reader to do more work than I think they should have to to understand where your method improves. For example, I like the discussion of Lines 471-479, but I think this should be in the main body. Also, I would compare your method directly to the sub-gradient method in the same place and carefully state what regimes you are winning in.
2. I would formally put your first-order oracle complexity (and probably runtime too) into a theorem which also collects all of your assumptions together (or mentions Assumption 1). Currently, this is just in the body of the paper after Theorem 2. I think putting it into a theorem makes it more clear and easier to cite.
3. The cases with $\mu = 0$ or $\nu = 0$ are handled in Appendix C.4, but not in a very formal way. E.g., a $O(1 / t)$ rate is mentioned, but I think the performance of your algorithm in these settings should be both stated carefully in a theorem and compared to prior work in this setting (e.g., the sub-gradient method as well as Levy et al. 2020, Carmon and Hausler 2022). Indeed, the $\mu = 0$ and $\nu = 0$ settings are important enough that entire papers have been written about them (and in fact, in some sense it is the original setting), so I think it is important that these are not an afterthought, and that the reader doesn't have to do any work to understand how the bounds you get in these settings compares to prior work. This is actually the single biggest weakness in my opinion.

Minor/typos:
1. Line 71 is missing a subscript. $\ell(w)$ should be $\ell_i(w)$.
2. In the two inline lines above Line 431, there shouldn't be a square on the norm. I.e., it should just be $|| w_1 - w_2||_2$.
3. Although it is clear what it is from context, I don't the the map $\ell(w)$ is ever formally defined before you start using it in Line 97.
4. "witha" in Line 7.
5. It might be good to just mention that when you say "smooth" in the paper, you always mean it in the sense that the gradient is Lipschitz (and not in the sense of a $C^\infty$ function).
6. I would include the fact that $\mathcal{Q}$ is any closed, convex, and nonempty set in Assumption 1. As far as I can tell, the only place you formally state this is in the text under Table 1.
7. Maybe just use the term "runtime" instead of, e.g., "global complexity." (Unless there is a reason you aren't using the term "runtime" that I'm missing?)
8. It might be nice to mention some prior work which doesn't just have a single hyperparameter (or really, state which algorithms do and don't), just to make it more clear how significant of a contribution that is.

**Questions:**

1. Just to make sure I understand correctly, it is always best to choose $b = 1$ to get the best first-order oracle complexity, but your runtime may be better with a different choice of $b$?
2. I'm confused by the $n q_{max} \le n$ in Line 491. Isn't this always true? Similarly, Line 81 says that $n q_{max}$ is upper bounded by "a constant in $n$," but isn't it just bounded by $n$ because $q_{max} \le 1$? I thought $\mathcal{Q}$ still needed to be a subset of the simplex per the text under Table 1. If not, I would fix that text and also include exactly what $\mathcal{Q}$ is in Assumption 1.
3. Regarding the case where $\mu = 0$ or $\nu = 0$, in Appendix C.4 you modify the entire algorithm/analysis to handle this case. It is good to know this works, but a classic trick to recover a non-strongly convex rate from the strongly-convex case is to just set something like, e.g., $\mu = \epsilon / D^2$ where $D$ is a bound on the diameter of $\mathcal{W}$. Would this recover the same rate, and/or is there a reason you don't do this? Just curious.
4. Just to make sure I understand correctly, is this the first work you are aware of which applies these cyclic coordinate-style updates to a non-separable dual feasible set, or is there any precedent for this?
5. I'm a bit confused by Lines 199 to 204 and how they connect to Appendix B.3 and in particular the bound after Line 556. It is also claimed in Line 202 that $n q_{max}$ is constant - why is this? (Even after clarifying this to me, I would recommend connecting it more closely to Appendix B, i.e., cite the exact equation in Appendix B you are referencing as opposed to all of Appendix B.)

**Limitations:**

Yes.

---

> ### Author Rebuttal · Authors · 2024-08-06
>
> Thank you for your detailed suggestions&mdash;we have incorporated the comments about comparisons, formal statements of complexity results, and minor typos into the manuscript.
>
> >**The cases with  $\mu=0$ or $\nu=0$  are handled in Appendix C.4, but not in a very formal way.**
>
> Thank you for raising this point. We have changed the analysis to account for the settings of $\mu = 0$, $\nu = 0$, and $\mu = \nu = 0$ formally. Because achieving an unconditional linear convergence rate that improves upon gradient descent was the main motivation of the work, we focused on the strongly convex-strongly concave setting. In the final version, we will provide a unified analysis for all settings.
>
> >**...it is always best to choose  $b=1$ to get the best first-order oracle complexity, but your runtime may be better with a different choice of $b$?**
>
> That is correct. If we assume that the cost of querying the oracle is $O(d)$ and $n \geq d$, then taking into account the $O(n)$ cost of updating the dual variable in each iteration, a batch size of $b = n/d$ balances the costs of the two updates. This affords us a $O(n^{3/2})$ dependence on $n$ in the overall runtime.
>
> >**Line 81 says that $n q_{\max{}}$ is upper bounded by "a constant in 𝑛," but isn't it just bounded by $n$ because $ q_{\max{}} \leq 1$?**
>
> We have added to Appendix B a precise claim regarding the fact that $n q_{\max{}}$ is constant. In summary, we mean that $nq_{\max{}}$ is bounded by a constant independent of $n$. The two canonical examples of $\mathcal{Q}$ are divergence-ball based and spectral risk measure-based feasible sets. It is true that $\mathcal{Q}$ is always a subset of the probability simplex so $q_{\max{}} \leq 1$. However, this bound is only tight when  $\mathcal{Q}$ is the entire probability simplex, regardless of the choice of $n$. Because this feasible set is often defined using statistical properties of the weights $q$, each element can be upper bounded as $q_i \leq c/n$ for some $c \geq 1$. For spectral risk measures, this can be seen by the argument in Section 2 of [Mehta (2023)](https://proceedings.mlr.press/v206/mehta23b.html)&mdash;every weight will be the integral of a bounded function over an interval of length $1/n$. For the $\theta$-CVAR (see our Appendix B.2), this bound is $1/(\theta n)$.  For f-divergence balls, in [Namkoong and Duchi (2017)](https://papers.nips.cc/paper_files/paper/2017/hash/5a142a55461d5fef016acfb927fee0bd-Abstract.html) Eq (4), the divergence ball ambiguity set is defined using radius $\rho/n$, which enforces the condition $\frac{n}{2} \Vert q - \mathbf{1}/n \Vert _2^2 \leq \frac{\rho}{n}$ on every element of $\mathcal{Q}$ in the case of $\chi^2$-divergence.
>
> >**Is this the first work you are aware of which applies these cyclic coordinate-style updates to a non-separable dual feasible set?**
>
> We are not aware of any previous work that does so in a cyclic fashion. The key element at play is that $\hat{q}_t$ (see Algorithm 1) does not need to be in the feasible set to be used in the gradient estimate in the primal update. Thus, we may apply cyclic updates to this vector even if we cannot do the same for $q_t$.
>
> >**...a classic trick to recover a non-strongly convex rate from the strongly-convex case is to just set something like, e.g., $\mu = \frac{\epsilon}{2D}$ where $D$ is a bound on the diameter of $\mathcal{W}$.**
>
> Thank you for identifying this trick. This would be one way to approach this, but it relies on knowing $D$, the diameter or initial distance-to-optimum in an unbounded setting. Furthermore, it leads to an extra log factor on the complexity. We chose to provide analysis for a direct method, which does not require knowledge of $D$ and leads to the improved convergence rates.

---

> ### Comment · Reviewer_em1q · 2024-08-08
>
> Thank you for these clarifications. The one answer I still don't fully follow is the one related to $n q_{max}$. I now realize Line 81 is connected to Line 468 in Appendix B. Line 468 reads: "$n q_{max}$ is upper bounded by a constant independent of $n$ in common cases" and then goes on to cite $\theta$-CVaR and $\chi^2$-DRO with radius $\rho$ as examples. I don't agree with this though in general. For $\theta$-CVaR for example, the constraint is $q_{max} \le \frac{1}{\theta n}$ as you wrote in your response. Then it is true that $n q_{max}$ is a constant if you restrict to $\theta = \Omega(1)$. But to my knowledge, the $\theta$-CVaR DRO problem encompasses all $\theta \in [1 / n, 1]$, in which case it is not correct to claim that $n q_{max}$ is always a constant for $\theta$-CVaR. (Indeed, the entire range $\theta \in [1 / n, 1]$ is considered in Levy et al. 2020. And it is even pointed out later in your Appendix B - Line 489 - that previous methods in the literature don't beat gradient descent when $\theta = 1 / n$.)
>
> It doesn't seem to me that the strength of your contribution relies on $n q_{max}$ being a constant as opposed to being as large as $n$ per the rates in Table 1. However, I would avoid claiming $n q_{max}$ is a constant in general for $\theta$-CVaR, $\chi^2$-DRO, etc. It seems to me like the point you want to make (?) is that if $\theta = \Omega(1)$, then your rate improves, but then you must make the $\theta = \Omega(1)$ restriction clear (and similarly for $\chi^2$-DRO, etc.).
>
> Please let me know if I'm still misunderstanding something however.
>
> (Also just to add to my original question - I would suggest avoiding the phrase "a constant in $n$" and just say "a constant" or "a constant independent of $n$." I understand it now, but when I first read "a constant in $n$," I thought it meant $cn$ for some constant $c$.)

---

> > ### Author Response · Authors · 2024-08-09
> > **Response to Reviewer em1q**
> >
> > Thank you for this recommendation and for improving the clarity of our work&mdash;we have added the assumption that the risk parameters (such as $\theta$) will be of constant order to make this claim.
> >
> > To give some background on this assumption, both spectral risk measures and $f$-divergence balls can be described using statistical functionals applied to the empirical distribution of the data. The dependence on $n$ typically comes only from the empirical distribution, and not the functional itself (although we will make this explicit in an assumption as you correctly pointed out).
> >
> > For any real-valued random variable $Z$ with cumulative distribution function (CDF) $F$ and finite first moment, the (population) $\theta$-CVaR of $Z$ is defined as $\mathbb{E}[Z | Z \geq F^{-1}(1-\theta)]$, where $F^{-1}$ is the quantile function or generalized inverse of $F$. In other words, it is the conditional mean of $Z$ given that $Z$ is above its $(1-\theta)$-quantile. For i.i.d. data $Z_1, \ldots, Z_n$ with empirical CDF $F_n$, we may define the (sample) $\theta$-CVaR as $\mathbb{E}[Z | Z \geq F_n^{-1}(1-\theta)]$ (which is roughly the mean of the top $n\theta$ order statistics). When applied to the empirical distribution of training losses, this recovers the description of CVaR given in Appendix B.2. When $\theta = \Theta(1)$, statistical arguments yield the convergence of the sample quantity to the population quantity, which is why it is typically assumed.
> >
> > Thank you again for this discussion and please let us know if we have addressed your questions fully.

---

> > > ### Comment · Reviewer_em1q · 2024-08-10
> > >
> > > This addresses my question - thank you! (I definitely think adding this explanation or something similar to the paper would be good - especially people who come from a more purely optimization as opposed to statistical background may be confused otherwise.)

---

### Official Review · Reviewer_P67z · 2024-07-30

**Soundness:** 3
**Presentation:** 2
**Contribution:** 3
**Rating:** 7
**Confidence:** 4

**Summary:**

The paper studies stochastic variance-reduced algorithms for regularized distribution robust optimization problems, formulated as a minimax problem. The paper proposed a novel primal-dual algorithm, and analyzed its iteration complexity. Under Lipschitz and smoothness assumptions, the algorithm achieves a linear convergence rate, with $O(n + d)$ per-iteration complexity, and total oracle calls of order $n + n q_{\max} L/\mu + n^{3/2} \sqrt{G^2 / (\mu \nu)}$.

**Strengths:**

Distribution-robust estimation is an important question, and the minimax ERM formulation leads to a challenging optimization problem. This paper made solid progress in this direction. The algorithm idea is natural, and the result improves over several exisitng works along this line of research. Importantly, the result imposes no additional structural assumptions on the uncertainty set $\mathcal{Q}$, except for convexity and an upper bound on the $\ell^\infty$ norm bounds for its elements.

**Weaknesses:**

When seeing $G$, $\mu$ and $\nu$ as constants, the oracle complexity scales as $n^{3/2}$ in terms of $n$. Although we get logarithmic dependence on $\varepsilon$ and the result is already state-of-the-art, if we only track the dependence on $n$, it's worse than the vanilla subgradient methods, as listed in Table 1.

Additionally, the analysis relies on the Lipschitz smooth convex + strongly convex regularization structure. Is it possible to extend the analysis to non-strongly convex settings, or the case with convex individual functions but strongly convex summation?

Another weakness is the space complexity. Storing the gradient estimates $\widehat{g}$ requires $O(n d)$ space. In many cases, this costs the same complexity as storing all the training data. So the advantage of stochastic algorithm is not very clear.

**Questions:**

- In order to minimize the total computational cost (assuming that the individual gradients can be computed in $O (d)$ time), what is the optimal choice of the minibatch size?
- On a related note, if we choose the parameters optimally, how does the total computational cost compare to other algorithms. In particular, as the per-iteration cost depends on $n$ as well, it is important to compare with the full-batch gradient-based algorithms.

**Limitations:**

The paper addressed limitations adequetly.

---

> ### Author Rebuttal · Authors · 2024-08-06
>
> Thank you for your thorough comments. We address them below.
>
> > **Is it possible to extend the analysis to non-strongly convex settings, or the case with convex individual functions but strongly convex summation?**
>
> Thank you for raising this important point. We discuss in Appendix C.4 modifications to the analysis which would handle if either or both of the strongly convex (strongly concave) parts of the objective were changed to convex (concave). In summary, we would not achieve linear convergence of course, but $O(1/t^2)$ if either $\mu = 0$ or $\nu = 0$ or $O(1/t)$ if $\mu = \nu = 0$. Because achieving an unconditional linear convergence rate that improves upon gradient descent was the main motivation of the work, we focused on the strongly convex-strongly concave setting. In the final version, we will provide a unified analysis for all settings.
>
> The “strongly convex summation” could be handled with a similar overall structure, but different inequalities to produce the upper and lower bounds on the primal-dual objective function. For the lower bound, rather than applying the smoothness of $\ell_i$ and then the definition of the proximal operator, we may apply an interpolation inequality that combines smoothness and strong convexity. One challenge is that we require $\ell_i$ to be Lipschitz continuous, so we would need to assume a compact domain for it to also be strongly convex.
>
> >**...if we only track the dependence on $n$, it's worse than the vanilla subgradient methods.**
>
> The vanilla subgradient method has an $n^2$ dependence due to the second term, whereas our method achieves an $n^{3/2}$ dependence.
>
> >**In many cases, this costs the same complexity as storing all the training data.**
>
> Thank you for identifying this point. We have updated the manuscript to include a discussion on space complexity, which in many cases can be reduced far below $O(nd)$ for two reasons. Firstly, this is a known limitation for SAG/SAGA-type variance reduction methods for empirical risk minimization. That being said, the additional storage cost (beyond the training data) is $O(n)$ in many common cases. This is because convex losses in machine learning can typically be expressed as $\ell_i(w) = h(y_i, x_i^\top w)$ for some differentiable error function $h$ and so the gradient is given by the scalar multiplication $h’(y_i, x_i^\top w) \cdot x_i$, where $h’$ is the derivative with respect to the second argument. Thus, only these scalar derivatives need to be stored.
>
> Secondly, a more subtle advantage of our method is that we use cyclic updates instead of randomized updates for the table of past gradients (as opposed to the randomized updates in SAGA). Conceptually, this means that for very large-scale workloads, a fraction of the gradient table can be paged out of memory until its values are updated again.
>
> >**In order to minimize the total computational cost... what is the optimal choice of the minibatch size?**
>
> The optimal batch size is $b = n/d$, which is used in the red lines in the experiments. As mentioned in the “Global Complexity” paragraph on page 7, because the dual variables must normalize to 1, we have an $O(n)$ computational cost per iteration regardless. Thus, under the assumption that $n \geq d$, we can compute $n/d$ gradients at $O(d)$ cost without changing the per-iteration complexity. This allows for the $n^{3/2}$ dependence in the sample size in Table 1, as opposed to $n^2$ for gradient descent.
>
> >**If we choose the parameters optimally, how does the total computational cost compare to other algorithms? In particular, as the per-iteration cost depends on $n$ as well, it is important to compare with the full-batch gradient-based algorithms.**
>
> Thank you for the suggestion; we have added a table for this quantity as well for clarity (see the supplemental PDF Table 6). Among the linearly convergent methods of the manuscript, this can be measured by multiplying the complexity of gradient descent, LSVRG, and Drago by $d$ and multiplying the complexity of LSVRG/Prospect by $n+d$. Thus, under $b = n/d$, Drago achieves the best complexity without conditions on $\nu$. A comparison to general-purpose primal-dual min-max algorithms is also given in Table 4 of the manuscript, with runtime.

---

> > ### Comment · Reviewer_P67z · 2024-08-13
> > **Response**
> >
> > Thanks for the response. I appreciate the detailed explanation about storage costs and comparison to other methods, and I request the authors to add those discussion in the revised version. I will keep my score.

---

### Author Rebuttal · Authors · 2024-08-06

We thank the reviewers for their hard work reviewing our paper and providing insightful and concrete comments! We have revised the manuscript to incorporate these suggestions. A summary of the main points of concern is listed below, and all remaining comments are addressed in individual responses.

**Space Complexity:** Reviewers P67z, VCV6, and WLa8 raised the major point that the space complexity of the proposed method appears to be $O(nd)$, which hinders applicability in high-dimensional scenarios. We describe in the individual responses that the true space complexity of the method can in fact be much lower (up to $O(n)$). One reason is due to a standard improvement of SAG/SAGA-type variance-reduced algorithms, and a second reason is particularly due to the unique cyclic coordinate-wise updates in our algorithm. We have also added additional experiments in a higher dimensional setting to understand performance in the $n \approx d$ regime.

**Theoretical assumptions and dependences:** First, Reviewers em1q and VCV6 inquired why we stated that quantity $n q_{\max{}}$ is upper bounded by a constant, given that it is trivially upper bounded by $n$. We clarify that the constant is independent of $n$, so it does not affect the overall complexity in terms of $n$. Second, Reviewers P67z and VCV6 mentioned the assumption that the losses and their gradients are both Lipschitz continuous may be unrealistic in common settings. In the comments, we justify these assumptions in various practical scenarios.

**The convex-concave setting:** Finally, Reviewers em1q and P67z brought up the importance of the non-strongly convex/non-strongly concave settings, i.e. when either or both $\mu = 0$ and $\nu = 0$. We have revised the manuscript to include the analysis formally; the dependence on $n$ remains competitive.

Please see the individual responses for more details. Thank you!

---

### Author Response · Authors · 2024-08-09
**Thank you for your feedback / Addressing additional concerns**

Dear reviewers,

As the discussion period draws to a close on August 13 (2 business days from now) we kindly request that you take a moment to review and acknowledge our responses to your questions and comments.

In response to the comments raised by Reviewer WLa8, P67z, VCV6 we have added additional experiments that demonstrate the performance of Drago for higher-dimensional problems with small batch sizes. We have better clarified our theoretical assumptions, such as those on the losses $\ell_i$ and the size of the uncertainty set $\mathcal{Q}$. Please see the responses for details.

If any further concerns or questions arise that we could address, please do not hesitate to reach out. We appreciate your time and feedback.

Sincerely,

The authors

---

### Decision · Program_Chairs · 2024-09-25

**Decision:**

Accept (poster)

**Comment:**

The paper received positive reviews, with a consensus that this is a solid contribution to the literature on optimization algorithms for distributionally robust optimization. Consequently, I recommend acceptance of this paper. When preparing the camera-ready revision, please pay close attention to the reviewers’ feedback and include the additional explanations provided during the rebuttal period. In addition, in Table 1, the complexities listed for Levy et al. do not agree with the best results obtained in that paper (and shown in Table 1 of Levy et al.) - please correct this in the revision.